



# Improvements in the Canadian Earth System Model (CanESM) through systematic model analysis: CanESM5.0 and CanESM5.1

Michael Sigmond[1], James Anstey[1], Vivek Arora[1], Ruth Digby[1,2], Nathan Gillett[1], Viatcheslav Kharin[1], William Merryfield[1], Catherine Reader[1], John Scinocca[1], Neil Swart[1], John Virgin[1,3], Carsten Abraham[1], Jason Cole[1], Nicolas Lambert[1], Woo-Sung Lee[1], Yongxiao Liang[2], Elizaveta Malinina[1], Landon Rieger[4], Knut von Salzen[1], Christian Seiler[5], Clint Seinen[1], Andrew Shao[6], Reinel Sospedra-Alfonso[1], Libo Wang[7], and Duo Yang[1]

[1]Canadian Centre for Climate Modelling and Analysis, Climate Research Division, Environment and Climate Change Canada, Victoria, BC, Canada
[2]School of Earth and Ocean Sciences, University of Victoria, Victoria, British Columbia, Canada
[3]Department of Geography and Environmental Management, University of Waterloo, Waterloo, Ontario, Canada
[4]Institute of Space and Atmospheric Studies, University of Saskatchewan, Saskatoon, Saskatchewan, Canada
[5]School of Environmental Studies, Queen's University, Kingston, Ontario, Canada
[6]Hewlett Packard Enterprise Canada, At-Scale Engineering, Victoria, BC, Canada
[7]Climate Processes Section, Climate Research Division, Environment and Climate Change Canada, Toronto, ON, Canada

**Correspondence:** Michael Sigmond (michael.sigmond@ec.gc.ca)

**Abstract.** The Canadian Earth System Model version 5.0 (CanESM5.0), the most recent major version of the global climate model developed at the Canadian Centre for Climate Modelling and Analysis (CCCma) at Environment and Climate Change Canada (ECCC), has been used extensively in climate research and for providing future climate projections in the context of climate services. Previous studies have shown that CanESM5.0 performs well compared to other models and have revealed several model biases. To address these biases, CCCma has recently initiated the 'Analysis for Development' (A4D) activity, a coordinated analysis activity in support of CanESM development. Here we describe the goals and organization of this effort and introduce two variants ("p1" and "p2") of a new CanESM version, CanESM5.1, which features substantial improvements as a result of the A4D activity. These improvements include the elimination of spurious stratospheric temperature spikes and an improved simulation of tropospheric dust. Other climate aspects of the p1 variant of CanESM5.1 are similar to those of CanESM5.0, while the p2 variant of CanESM5.1 features reduced equilibrium climate sensitivity and improved ENSO variability as a result of intentional tuning of the atmospheric component. The A4D activity has also led to the improved understanding of other notable CanESM5.0/5.1 biases, including the overestimation of North Atlantic sea ice, a cold bias over sea ice, biases in the stratospheric circulation and a cold bias over the Himalayas. It provides a potential framework for the broader climate community to contribute to CanESM development, which will facilitate further model improvements and ultimately lead to improved climate change information.





# 1   Introduction

Effective efforts to adapt to and mitigate future climate change rely on reliable climate change projections, which are provided by climate models. It is therefore important to develop high quality climate models that provide credible and user relevant output. The latest major version of the global climate model developed at the Canadian Centre for Climate Modelling and

Analysis (CCCma) in the Climate Research Division (CRD) of Environment and Climate Change Canada (ECCC) is the Canadian Earth System Model version 5.0 (Swart et al., 2019b). Since its inception in 2018, simulations of CanESM5.0 have contributed to 21 model intercomparion projects (MIPS), 18 of which were done in the context of the World Climate Research Programme (WCRP) Coupled Model Intercomparison Project Phase 6 (CMIP6). Participation in CMIP6 resulted in about 154,000 simulation years ($\sim$ 300 TB) of published CanESM5 output on the Earth System Grid Federation (ESGF)

encompassing almost 600 different physical variables. Through participation in a broad range of MIPs and its large ensembles (Figure 1), CanESM5 simulations were particularly valuable in underpinning many parts of the Working Group I Contribution to the IPCC Sixth Assessment Report. For example, CanESM5's 50-member ensemble, as the largest CMIP6 ensemble, was used to estimate the internal variability in projected warming in the assessment (Lee et al., 2021). CanESM5 was also used as one of five models to illustrate a high warming storyline (Lee et al., 2021), as one of four ESMs to illustrate the response to

carbon dioxide removal (Canadell et al., 2021) and was the only model to provide multiple ensemble members of idealized zero-emission scenarios providing valuable understanding on the recovery of the Atlantic Meridional Overturning Circulation (AMOC) under stabilized warming (Lee et al., 2021; Sigmond et al., 2020). In addition to CMIP6, CanESM5 simulations have been contributed to a number of other MIPs including CovidMIP (Jones et al., 2021), the Southern Ocean Freshwater Initiative (SOFIA), and the Stratospheric Nudging And Predictable Surface Impacts (SNAPSI) project (Hitchcock et al., 2022), providing

input to other key science questions.

While CanESM5.0's atmospheric climatology has been found to be particularly good compared to other models (Eyring et al., 2021), several studies have revealed some model deficiencies and biases, such as the occurrence of spurious stratospheric warming spikes (Santer et al., 2021) and an overestimation of the aerosol optical depth variability (Jones et al., 2021). A notable characteristic of CanESM5.0 is its high equilibrium climate sensitivity, which at 5.65K is the highest among all CMIP6 models

(Zelinka et al., 2020). To support model development and help eliminate or reduce biases in future versions of CanESM, CCCma has recently initiated the 'Analysis for Development' (A4D) activity. This activity has established a process through which CanESM output is analyzed in a systematic and ongoing manner. This process and the goals of the A4D activity are described in Section 2. One important result of the A4D activity is a new CanESM version, CanESM5.1, which includes several improvements over CanESM5.0 as described in Section 3. A basic comparison between the characteristics of CanESM5.0

and two variants of CanESM5.1 ("p1", the default variant and "p2", with an alternate atmospheric model tuning) will be provided in Section 4. Finally, Section 5 provides a more in-depth analysis of specific CanESM5.0 and CanESM5.1 biases and characteristics.





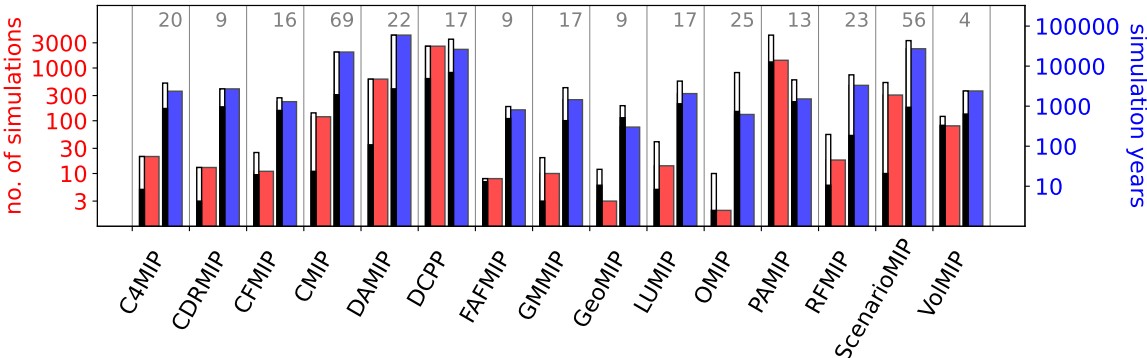

**Figure 1.** CanESM5.0 number of simulations (red) and total simulation years (blue) contributed to experiments defined by CMIP6-endorsed MIPs. For each MIP, thinner inset bars show the median (black) and maximum (white) number of contributions by all models participating in the MIP, with the number of contributing models indicated at top. Determined by a search of the ESGF archive on 22 Oct 2022 for monthly-mean atmospheric near-surface temperature (tas), precipitation (pr), ocean surface temperature (tos) and surface salinity (sos).

## 2 Analysis in support of model development

Model development focuses primarily on systematic errors, or "issues", with the properties and behaviour of previous and existing model versions. Such model issues can usefully be categorized into three types: *version-specific* (i.e issues particular to a specific model version) that are due to the changes made in a single model version; *model-systemic* (i.e. issues shared across multiple versions of a model) that are due to underlying properties of the model, which are relatively insensitive to recent development efforts; and *community-systemic* (i.e. issues shared across multiple diverse CMIP ESMs), which are typically due to a community-wide issue related to the absence, or manner of treatment, of one or more physical processes. While model issues are generally first identified as a simple bias in some quantity relative to observations, ideally they would ultimately be connected to, and understood in terms of, the specific representation of physical processes or dynamical mechanisms in the model.

Ideally, model development would involve an iterative process in which each new candidate version would undergo a comprehensive evaluation exercise to gauge the success of development efforts and to inform the utility of that version for its intended applications. However, given the complex nature of model development and tight deadlines associated with large projects such as CMIP, there is typically insufficient time and resources to iteratively undertake the breadth of experiments and subsequent diagnostic analysis that would be required to derive such insight. Each modeling centre meets this challenge in some undocumented manner in the finalization of production versions of their ESM.

At the CCCma, we have recently put a focus on this challenge and initiated a new activity, Analysis for Development (A4D), to critically review existing procedures and to develop strategies to evaluate new versions of our earth system model, CanESM. There are three primary objectives of A4D:





– To better document, and make more systematic, our efforts to resolve model issues - including both successes and failures. A regular A4D meeting was established and two lead scientists appointed to coordinate and oversee its activities. Through such meetings, issues with CanESM are reviewed and their potential sources discussed. This information is then kept track of, and permanently documented on an issue tracker of the same Gitlab version control system used to manage all of the CanESM code base. More detailed, "deep-dive", scrutiny of some issues are undertaken by striking working groups to independently investigate specific issues through additional analysis and sensitivity tests. Since these working groups need not be limited to CCCma staff, they can also help to enable input from the wider scientific community into CanESM development.

– To initiate a more automatic diagnostic analysis to identify and monitor issues. Systematic and automated evaluation of climate model versions has been highlighted by the community as an important tool for advancing model development (e.g., Gleckler et al., 2016; Eyring et al., 2016, 2020). All working groups are required to propose metrics that are salient to the issue they are tasked with investigating (even if that issue was eventually resolved). The collection of such CanESM specific metrics, along with more standard diagnostics, are brought together to form a suite of diagnostics that can be run automatically. The advantages of such an automatic package are that it can be applied to the model at any time during development, provide early detection of new issues, be used to monitor the occurrence and severity of known issues across model versions, and remove the burden from individual scientists to reproduce their analysis on future versions long after their working group responsibilities have ended. The output of this package is organized in standardized diagnostic reports and posted on the public CanESM5 gitlab page for key model versions (https://gitlab.com/cccma/canesm/-/wikis/home).

– To engage CCCma's full resources to better identify and maintain focus on model issues. While a large portion of a climate laboratory's scientists are engaged in model development, typically a much smaller subset are responsible for bringing the major components together to obtain a new ESM version with "acceptable" properties and behaviour (e.g. biases relative to observations and responses to external forcings such as climate sensitivity). With the A4D initiative, we have chosen to make the acceptance of new ESM versions a group-wide responsibility. The value of this approach is that it engages all CCCma staff – particularly those with extensive analysis expertise who might not typically engage in model development activities - and it provides a process to engage the expertise of external colleagues in the academic community.

The rest of the paper documents the results of the first phase of the A4D activity, which includes an new and improved version of CanESM, CanESM5.1, and a number of deep-dive analyses that provide insight on the origin and possible elimination of systematic model biases in CanESM5.



## 3  Models

### 3.1  CanESM5.0

CCCma use a three digit naming convention for our models: M.m.p, where **M** is the major version, **m** is the minor version and
**p** is the patch version. Changes in the patch version represent technical changes which generally do not alter the bit pattern
(and are tested not to alter the climate) and are not advertised externally; changes in the minor version number represent the
incorporation (or activation) of new physics and technical changes within the existing model components; and changes in
the major version number represent wholesale replacement or major changes to the model basic components. Following this
convention, CanESM5.0 represented a major change compared to its predecessor CanESM2, CCCma's Earth system model
that contributed to most experiments of CMIP5. Compared to CanESM2, CanESM5.0 includes completely new models for
the ocean, sea ice and marine ecosystems, and a new coupler, while the atmosphere, land surface and terrestrial ecosystem
models improved incrementally. The resolution of CanESM5.0 is similar to that of CanESM2 and at the lower end of CMIP6
models, allowing for more simulation years (and larger ensemble sizes) to be achieved than other CMIP6 models with higher
resolution. Two CanESM5.0 model variants, labelled "p1" and "p2", have been released[1]. Compared to the p1 variant, the p2
variant featured improved remapping of windstress fields passed from the atmosphere to the ocean, and the removal of a bug
that led to cold spots over Antarctica. It should be noted though that for most variables differences between the simulated mean
climate and its response to forcing are generally small and physically insignificant. Here we use the p2 version, which will be
referred to in this paper as "CanESM5.0"[2]. A 40-member ensemble of the p2 version with standard CMIP6 historical forcings
(i.e., the CMIP6 "historical" experiment) was run[3] and is used for the analysis in this paper. Note that we use all ensemble
members, except when explicitly mentioned. In such cases fewer ensemble members were needed to obtain robust conclusions.
For further details on CanESM5.0 we refer the reader to Swart et al. (2019b).

### 3.2  CanESM5.1

Here we present a new version of CanESM, CanESM5.1, which includes the following main improvements compared to
CanESM5.0:

– A retuning of parameters associated with the hybridization of advective tracers, including dust and moisture. In particular,
  the dust hybridization parameters were poorly tuned in CanESM5.0, which led to unrealistic spikes in dust burdens.
  These improvements and impacts on the simulation of stratospheric temperatures and tropospheric dust will be described
  in Section 5.1.

---

[1]The "p" label is the "physics_index" from the CMIP6 data reference syntax (available at https://pcmdi.llnl.gov/CMIP6/Guide/dataUsers.html). It is an
official label for a publicly released model version, and is unrelated to the patch version label, **p**, described above.

[2]CanESM5.0 is usually referred to as "CanESM5" in Swart et al. (2019b) and elsewhere. Here we retain the minor version label to distinguish this model
from CanESM5.1.

[3]Note that a 25-member historical ensemble of the p1 CanESM5.0 version is also available on the ESGF.





- An improved remapping of atmospheric heat fluxes that are passed to the ocean grid within the coupler, from the Earth
System Modeling Framework (ESMF) *conservative* routine in the p2 version of CanESM5.0, to the *conservative2* option
        in CanESM5.1[4] . The major effect of this change is the preservation of a smooth derivative across grid cells, which helped
        to reduce the nonphysical "blocky" pattern in the heat fluxes on the ocean grid (see Figure A1).

- Very significant technical changes were implemented in the structure of the source code of CanAM, the atmospheric
        general circulation model component of CanESM, with the syntax updated from pre-Fortran 77 to free-form Fortran
(F90+), and more significantly, a major reorganization of array structures within the model to improve efficiency and
        flexibility. An important objective behind these changes was to facilitate the interfacing of the CanAM physics with
        different dynamical cores (and specifically GEM; Qaddouri and Lee, 2011), and to provide quality of life improvements
        for developers, such as making configurable parameters available in namelists rather than hard-coded. We note that the
        syntax changes preserved the bit pattern of the model, and that the changes to array structure were not bit-identical but
had no statistically discernible impact on the model climate.

    As for CanESM5.0, p1 and p2 model variants of CanESM5.1 are provided, but it is important to note that the differences
between the CanESM5.1 p1 and p2 variants are completely independent from, and unrelated to, the differences between the
CanESM5.0 p1 and p2 variants. As will be shown in Section 4 the physical climate of the p1 variant of CanESM5.1 is virtually
indistinguishable from the p1 and p2 variants of CanESM5.0. The climate characteristics of the p2 version of CanESM5.1, by
contrast, are quite different, as the atmospheric component was retuned to best match historical warming, ENSO amplitude
and ENSO seasonality. The goal of this retuning exercise was to investigate the extent to which an alternative parameter tuning
could 1) alleviate the high bias in historical warming of our CMIP6 model (CanESM5.0, and equivalently CanESM5.1-p1);
and 2) improve the skill of CanESM5.1-p1 seasonal to decadal forecasts, for which the representation of the El Niño-Southern
Oscillation (ENSO) is crucial. This tuning was accomplished by adjusting the 9 free parameters summarized in Table 1 within
their physically plausible ranges. The impacts of this retuning exercise on ENSO variability and climate sensitivity will be
described in Section 5.2. Large ensembles of simulations with standard CMIP6 historical forcings consisting of 47 members
for CanESM5.1-p1 and 25 members for CanESM5.1-p2 have been published on ESFG under the CMIP6 project, and have
been used for the analysis presented in the next Sections. Note that the strong similarity in physical climate between three of
the CanESM versions that are available on the ESGF (CanESM5.0-p1 and p2, and CanESM5.1-p1) supports combining these
into a single large ensemble for those applications that could benefit from this size of historical ensemble (25 members of
5.0-p1, 40 members of 5.0-p2, and 47 members of 5.1-p1, giving 112 members in total).



**Table 1.** Summary of the 9 tuned parameters for the p1 and p2 variants of CanESM5.1.

| Scheme | Parameter | Physical description | p1 | p2 | unit |
|---|---|---|---|---|---|
| Cloud micro-physics | ap_facacc | Mass accretion rate of cloudwater to precipitation due to the collection of cloud droplets by raindrops | 15 | 5.3874 | $s^{-1}$ |
| | ap_facaut | Efficiency coefficient in mass autoconversion rate of cloud-water to precipitation due to the collision-coalescence processes of cloud droplets which determines cloud life cycle | 0.1204 | 0.2355 | – |
| | ap_uicefac | Empirical constant in calculations of the ice crystal fall speed due to the influence of gravity | 4200 | 5688 | $s^{-1}$ |
| Deep convection | ap_alf | Proportionality parameter relating vertically integrated convective kinetic energy with the cloud-base mass flux | $5.00\ 10^8$ | $3.97\ 10^6$ | $m^4\ kg^{-1}$ |
| | ap_taus1 | Dissipation timescale of CAPE in prognostic closure | 21600 | 2332 | s |
| | ap_weight | Evaporation efficiency in downdrafts of deep convection regions | 0.7500 | 0.8165 | – |
| Shallow convection | ap_scale_scmbf | Scaling factor for shallow convection cloud-base mass flux | 0.0300 | 0.0493 | – |
| Surface layer (atmosphere) | ap_ct | Neutral drag coefficient over water controlling the heat ventilation and evaporation rates | 0.0010 | 0.0012 | – |
| Surface module (land) | ap_drngat | Scaling factor for soil drainage at the bottom of the soil levels | 0.1000 | 0.1748 | – |

## 4 Basic comparison between CanESM5.0 and CanESM5.1

### 4.1 Historical mean climate

In this section we provide a high level comparison between the model characteristics of CanESM5.0 and the two variants of
CanESM5.1. As documented in Swart et al. (2019b), CanESM5.0 reproduces the large-scale features of the observed climate,
but suffers from several regional biases. As shown in Figures 2a and 2d, these biases include a cold bias over sea ice covered
regions in winter in both hemispheres (further discussed in section 5.3.2), a cold bias over the Tibetan Plateau (section 5.3.4),
warm biases over the eastern boundary current systems, the Amazon, and North America in summer, and a cold bias in the
North Atlantic associated with a positive sea ice bias (section 5.3.1 and Figure A3). This positive North Atlantic sea ice bias
results in an overestimation of total Arctic sea ice area in winter (Figure 3a); a similar North Atlantic bias is found in one other
CMIP6 model (CAMS-CSM1-0; Figure 4c). By contrast, September Arctic sea ice area only shows a very small (positive) bias,

---

[4]For details we refer to the ESMF reference manual at https://earthsystemmodeling.org/docs/release/latest/ESMF_refdoc.pdf



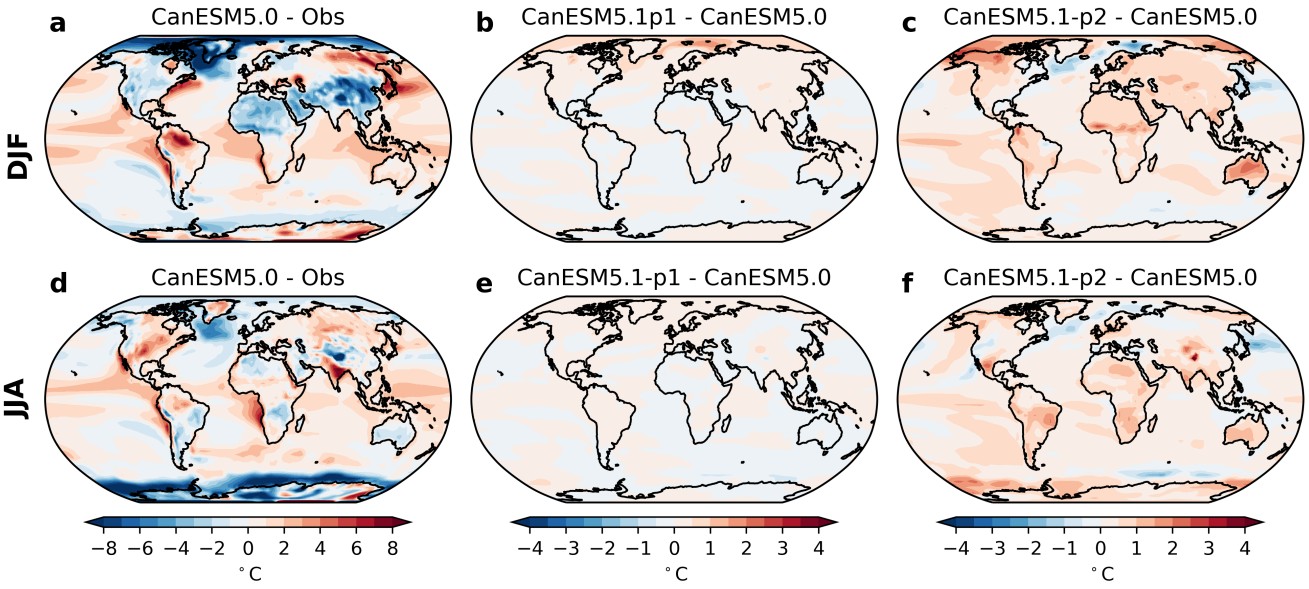

**Figure 2.** Ensemble-mean surface air temperature bias relative to ERA5 (Hersbach et al., 2020) in CanESM5.0 (a,d), and the ensemble-mean difference between CanESM5.1-p1 and CanESM5.0 (b,e), and between CanESM5.1-p2 and CanESM5.0 (c,f), averaged over 1981–2010 for the December–February (DJF; top row) and June–August (JJA; bottom row) seasons. Note the different contour intervals for (a,d) and (b,c,e,f).

which is among the smallest among CMIP6 models as shown by Figure 4a. CanESM5.0 shows a year-round overestimation of Arctic sea ice volume compared to the observation-based product PIOMAS. In the Antarctic, CanESM5.0 generally simulates too much sea ice (Figure 3c), and in austral winter this positive bias is largest among the CMIP6 models considered here
(Figure 4b).

Compared to CanESM5.0, CanESM5.1-p1 has a slightly warmer Arctic winter (Figure 2b). While Arctic sea ice area is slightly smaller in CanESM5.1-p1 (Figure 3a), the main cause of these slightly warmer temperatures is likely the thinner sea ice, as the Arctic sea ice volume is smaller (and closer to PIOMAS) than in CanESM5.0 (Figure 3b). Apart from these small differences, the climates of CanESM5.0 and CanESM5.1-p1 are virtually indistinguishable, as shown in panels (b) and (e) of
Figures A3-A8.

Differences between the historical climates of CanESM5.1-p2 and CanESM5.0 are larger. The pre-industrial climate in CanESM5.1-p2 is 0.5 °C warmer than that of CanESM5.0 and CanESM5.1-p1, with a global mean surface air temperature of 13.8 °C in CanESM5.1-p2 versus 13.3 °C in CanESM5.1-p1 and CanESM5.0 (not shown). While CanESM5.1-p2 warms less over the historical period (as described below), its global mean surface temperature is still slightly higher than that of
CanESM5.0 when averaged over 1981–2010 (Figure 2c,f). Cloud fraction is generally lower than in CanESM5.0/CanESM5.-p1, especially over the ocean (Figure A2). The warmer surface temperatures are consistent with a warmer troposphere (Figure A7c,f), a larger tropical wet bias (Figure A5c,f), and a stronger subtropical jet (Figure A8c,f). In addition, the strong strato-

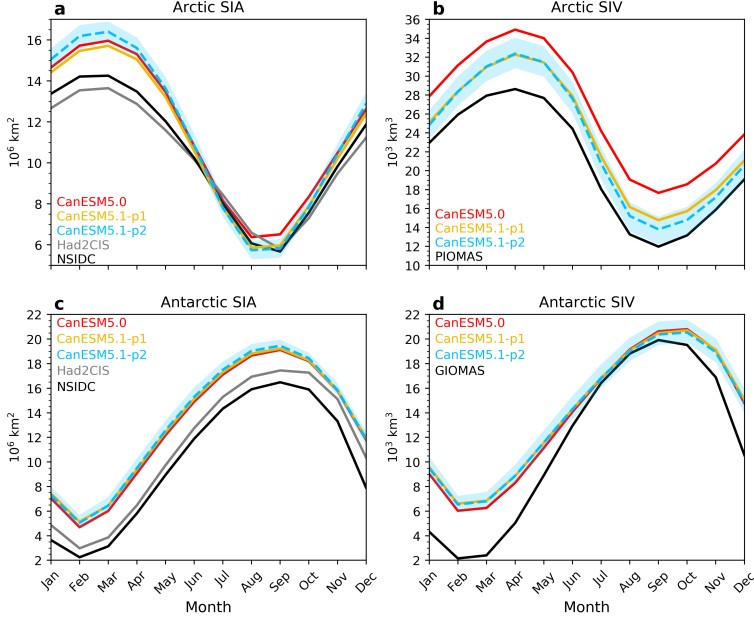

**Figure 3.** Seasonal cycles of sea-ice area (a,c) and volume (b,d) in the Northern Hemisphere (a,b) and the Southern Hemisphere (c,d) averaged over 1981–2010, for CanESM5.0, CanESM5.1p1, CanESM5.1p2, NSIDC (Meier et al., 2021) and Had2CIS (Lin et al., 2020) for sea ice area and the PIOMAS and GIOMAS reanalyses (Zhang and Rothrock, 2003) for sea ice volume. The lines represent the ensemble means and the blue shading the ensemble range in CanESM5.1-p2.

spheric polar vortex bias that was seen in CanESM5.0 is slightly larger in CanESM5.1-p2, which has implications for sudden stratospheric warmings (Section 5.3.3). An exception to the warmer temperatures in CanESM5.1-p2 is the North Atlantic, which experiences a slightly larger cold bias (Figure 2c,f). This is consistent with a slightly larger wintertime high sea ice bias, both in the North Atlantic (Figures 4c and A3c), and over the entire Arctic (Figure 3a). For mean biases in other physical aspects of the historical climate we refer to the reports published at the CanESM gitlab page (https://gitlab.com/cccma/canesm/-/wikis/home).

## 4.2 Climate change

As documented by Swart et al. (2019b) and shown in Figure 5a, CanESM5.0 simulates more historical warming than in observations, while historical Arctic sea ice trends are very close to observations (Figure 4a, vertical axis). This implies that the sea ice decline per degree of global warming is underestimated, which is a common model bias (Notz and SIMIP Community, 2020). Historical Antarctic sea ice trends are negative as in almost all other CMIP6 models (Roach et al., 2020), which appears inconsistent with the positive historical trends in observations (Figure 4b). We note though, that that there is one ensemble member of CanESM5.1-p1 that simulates a positive Antarctic sea ice trend in February as in observations, suggesting that





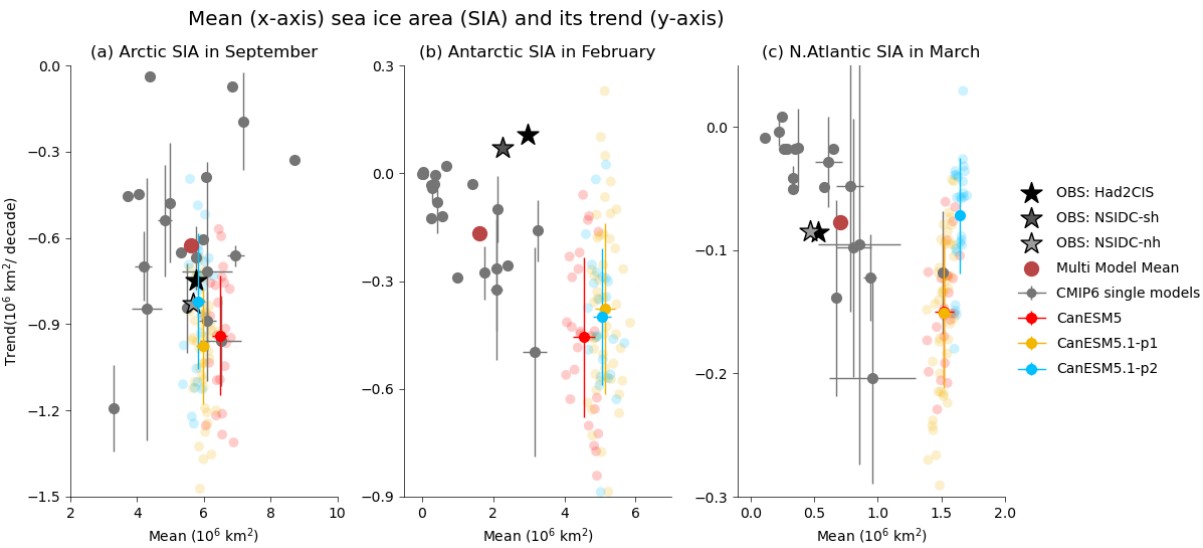

**Figure 4.** 1981–2010 climatological mean (horizontal axis) versus historical trends (vertical axis) in September Arctic SIA (a), February Antarctic SIA (b) and March North Atlantic SIA (c). Stars indicate observations, gray dots ensemble means of CMIP6 models, small colored solid dots the ensemble means of CanESM5.0 and CanESM5.1, the large red dot the CMIP6 ensemble mean, horizontal and vertical bars the across-ensemble standard deviations for models with multiple ensemble members, and the faint coloured dots the individual ensemble members of the CanESM5.0 and CanESM5.1 simulations. The figure is an updated version of Figure 3.20 from IPCC AR6 WGI.

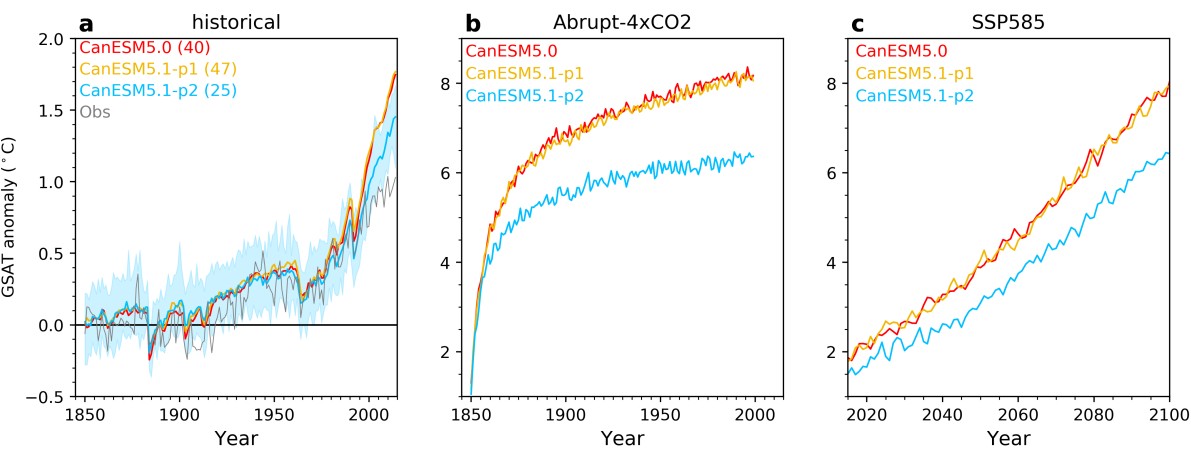

**Figure 5.** Global mean Surface Air Temperature (GSAT) anomaly relative to the mean in the piControl runs in CanESM5.0-p2, CanESM5.1-p1 and CanESM5.1-p2 for a) historical, b) abrupt 4xCO2 and c) SSP585 simulations. In a) the lines represent the ensemble mean, the blue shading the ensemble range in CanESM5.1-p2 the gray line the observed anomaly relative to the 1850 to 1900 mean from HadCRUT5 (Morice et al., 2021) and the number in the legend indicates the ensemble size.





CanESM5.0/5.1 is not inconsistent with observations, and that the observed positive Antarctic sea ice trends could be the result of internal variability.

While historical global warming (Figure 5a) and sea ice trends (Figure 4) in CanESM5.1-p1 are virtually identical to those in CanESM5.0, historical warming in CanESM5.1-p2 is about 20% weaker than in CanESM5.0 (Figure 5a), consistent with the atmospheric model tuning described in Section 3.2. The lower warming rate is also seen in the abrupt 4xCO2 simulations (Figure 5b), used in Section 5.2.2 to calculate the equilibrium climate sensitivity, and in future projections such as that run under the SSP5-8.5 scenario (Figure 5c). We note though that while the historical warming in CanESM5.1-p2 is closer to observations, the lower end of the GSAT ensemble spread (blue shading in Figure 5a) still exceeds observations after 2005, implying that historical warming is still biased high. Finally, we point out that historical sea ice trends in CanESM5.1-p2 are similar to those in CanESM5.0/CanESM5.1-p1, with the exception that historical winter trends in North Atlantic sea ice are somewhat smaller and more consistent with observations (Figure 4).

## 5   Model characteristics and systematic model biases

In this section we provide more detailed analyses of specific model biases and characteristics in CanESM5.0 and CanESM5.1. We start with the improvements in CanESM5.1 compared to CanESM5.0 that are the result of the improved dust tuning (Section 5.1). This is followed by a description of changes in ENSO characteristics and climate sensitivity between the two variants of CanESM5.1 as a result of the retuning of the atmospheric component (Section 5.2). Finally, Section 5.3 describes the analysis of a number of outstanding biases in CanESM5.0/CanESM5.1, which in most cases have led to promising paths to resolving these biases in future versions of CanESM.

### 5.1   Improvements in CanESM5.1 compared to CanESM5.0: dust tuning

Recent analyses have revealed some unrealistic features in CanESM5.0, including the occurrence of spurious stratospheric temperature spikes in certain ensemble members of its historical large ensemble (Santer et al., 2021) and spurious tropospheric dust storms. Subsequent testing and analysis revealed that both of these features were the result of improperly tuned free parameters associated with the "hybridization" of the fine- and coarse-mode mineral dust tracers in CanESM5.0. Hybridization is a transform applied to tracer variables designed to reduce their dynamic range and alleviate artifacts associated with the spectral advection of positive definite tracers (von Salzen et al., 2013). However, the advected tracer mass may not be conserved as a consequence of the hybridization, depending on the values of two free hybridization parameters. Conservation of global tracer mass is ensured in CanESM by correcting the tracer mass following each advection time step (von Salzen et al., 2013), and improper tuning of the hybridization parameters associated with mineral dust resulted in anomalously large mass corrections in some instances. Subsequent retuning of these parameters reduced the magnitude of the mass corrections in CanESM5.1, and eliminated spurious stratospheric warming events and tropospheric dust storms that were present in CanESM5.0, as we document here.




We first describe the spurious lower-stratospheric warming spikes in CanESM5.0 and their relationship to the mineral dust spikes. Figure 6a shows the monthly timeseries of global mean lower stratospheric temperature anomalies in the first ensemble members of CanESM5.0, CanESM5.1-p1 and CanESM5.1-p2. All three model versions show multi-month warming peaks

in response to the major volcanoes and long-term cooling in response to increasing carbon dioxide concentrations, but only CanESM5.0 shows ∼1–2 month warming spikes that cannot be explained by external forcings. To more clearly highlight these warming spikes, we subtract the ensemble mean timeseries (which averages out internal variability and hence represents the forced response), which leaves the part of the temperature anomalies that is due to internal variability and is hereafter referred to as the unforced variability (Figure 6b). It appears that the warming spikes coincide with large peaks in the global coarse

mineral dust burden (Figure 6c). This is further quantified in Figure 6d, which shows the relationship between the dust and temperature spikes in all months of all available ensemble members. It clearly shows a positive correlation between coarse mode mineral dust burden and unforced lower stratospheric temperature. It also shows that elimination of coarse mineral dust spikes in CanESM5.1-p1 and CanESM5.1-p2 is associated with the elimination of lower stratospheric temperature spikes. Lagged scatter plots reveal that there is no correlation between coarse mineral dust spikes and the lower stratospheric temperature in

the prior month (Figure A9a), which is evidence that the temperature spikes are the result and not the cause of the dust spikes. They also show that there is a weak impact of coarse mineral dust spikes on lower stratospheric temperature in the following month (Figure A9b), and that this impact does not last for longer than a month (Figs. A9c,d). Further analysis revealed that the coarse mineral dust spikes (defined as spikes whose global burden exceeds $12 \times 10^9$ kg) occur once every ∼50 years, mainly in boreal summer (Figure A10a), rarely last longer than two months (Figure A10b), and have a maximum impact on zonal mean

temperature in the low-latitude lower stratosphere (Figure A11).

We next document the impact of the improved hybrid mineral dust tracer tuning in CanESM5.1 on the aerosol optical depth (AOD). A more detailed analysis shows that dust AOD in CanESM5.0 is primarily associated with concentrations of fine-mode tropospheric dust and less so with stratospheric coarse-mode dust (not shown). Differences in seasonality of the fine and coarse dust are plausible, given that they have rather different spatial distributions and are therefore subject to different transport

processes, including uplifting and deep convection.

Figure 7 shows total aerosol optical depth from 2007–2014 (a,c) and the mean seasonal cycle over that time period (b,d) in CanESM5.0-p2, CanESM5.1-p1, and CanESM5.1-p2. Also shown are remotely sensed AOD observations from the Moderate Resolution Imaging Spectroradiometer (MODIS, dashed lines; Platnick et al., 2017; King et al., 2013), Multi-angle Spectral Radiometer (MISR, dash-dot; Diner et al., 1998), and the Cloud-Aerosol Lidar with Orthogonal Polarization (CALIOP, solid

lines; Winker et al., 2009). These datasets are described in Appendix B.

The upper panels (a,b) of Figure 7 show near-global-mean AOD (restricted to 60°S–60°N to facilitate comparison with observations), and the lower panels (c,d) show AOD over Eastern and Central Asia. The time period 2007–2014 is chosen because 2007 is the earliest year for which all observational data products are available, and 2014 is the last year of the historical simulation.

In CanESM5.0, events with high AOD were predominantly associated with dust emissions in Eastern and Central Asia, with AOD that exceeded 10 locally (near the emission source), and produced extended plumes with aerosol optical depth



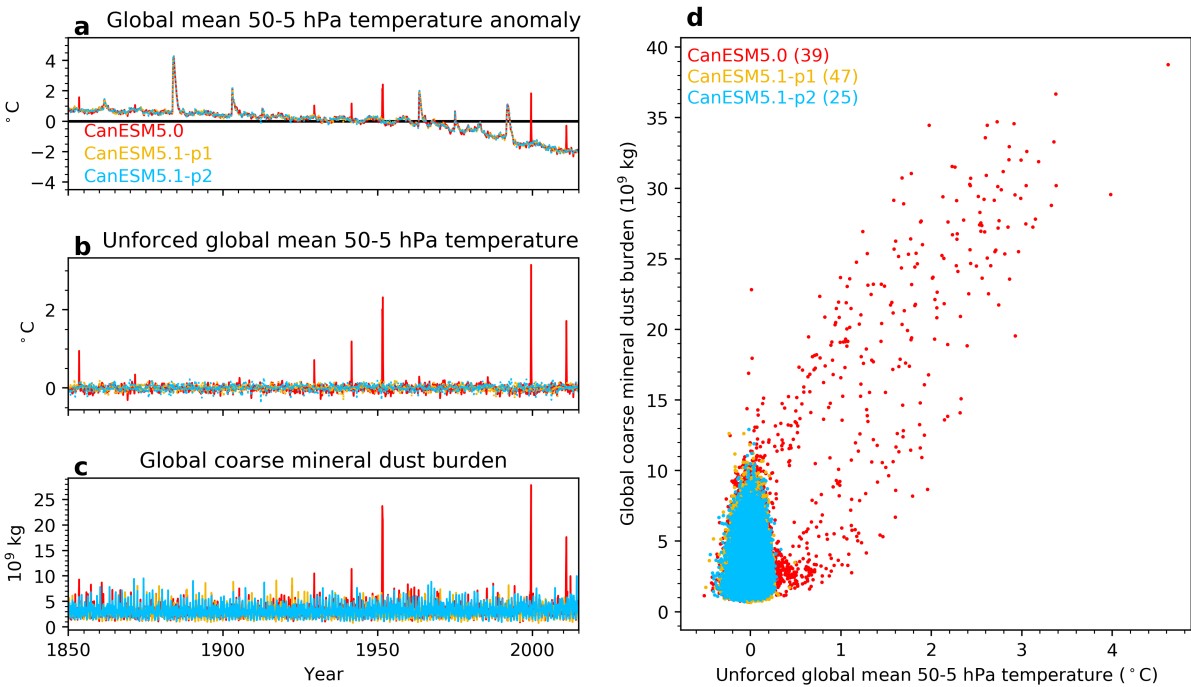

**Figure 6.** Lower stratospheric temperature spikes and their relation to coarse mineral dust. The monthly timeseries of a) global mean lower stratospheric temperature anomalies b) unforced variability in the global mean lower stratospheric temperature (details in text) and c) total global coarse mineral dust burden in the first ensemble member of each model version, and d) a scatter plot of the unforced global mean lower stratospheric temperature versus total global coarse mineral dust burden in all months of all ensemble members.

of order 1 that covered most or all of the northern extratropics. In general, CanESM5.0 was characterized by unphysically high variability in AOD across a range of different time scales, even during time periods that did not produce any notable stratospheric dust spikes (Figure 7), both regionally and globally (see, e.g., Jones et al. (2021)). Furthermore, periods with high

AOD preferentially occurred in boreal spring and autumn, producing a double-peaked seasonal cycle (Figure 7b,d), with an especially large autumn peak in Eastern and Central Asia (Figure 7d). A recent assessment of simulated mineral dust in CMIP6 models (Zhao et al., 2022) identified substantial variability in the regional seasonal cycles simulated by different models, and a number of models exhibit similar double-peaked behaviour over North China, which the authors attribute to excessive surface wind speeds in autumn and winter. However, CanESM5.0 is unique in the magnitude of this second peak. These issues

are substantially improved in CanESM5.1. The re-tuning of the hybridization parameters removes the mechanism by which mineral dust emission spikes were formed, such that they are absent in CanESM5.1. This correction reduces both the AOD variability and time mean, and improves the seasonal cycle relative to observations by substantially suppressing the boreal autumn peak (Figure 7b,d).



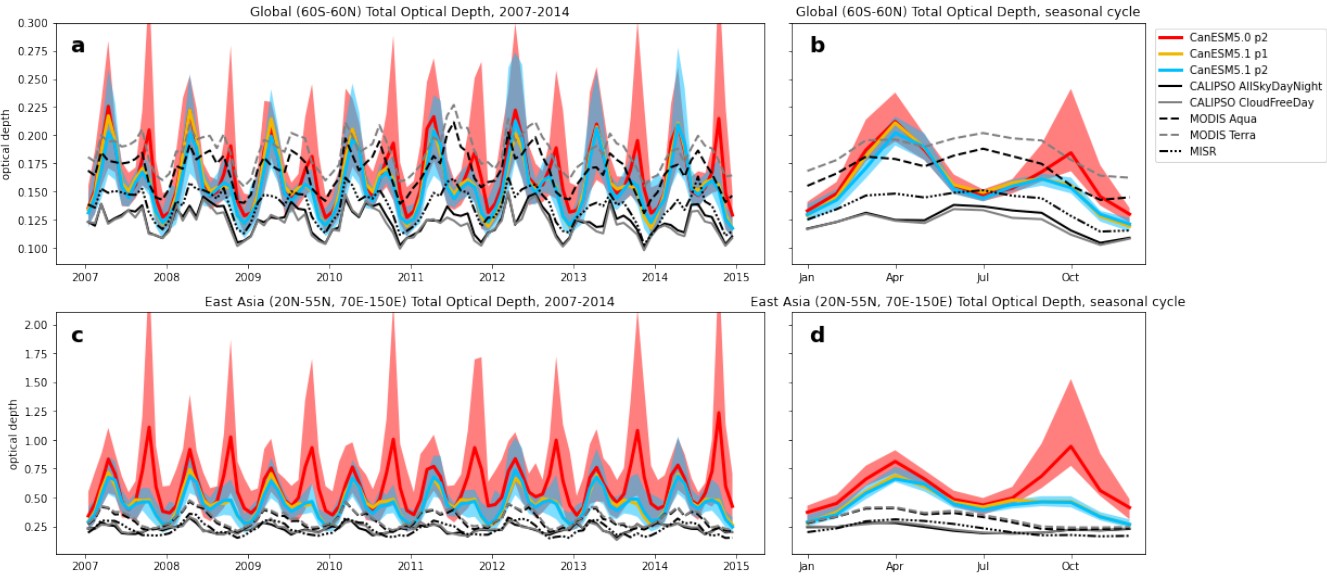

**Figure 7.** Timeseries (a,c) and mean seasonal cycles (b,d) of near-global mean (60°S–60°N, top) and East Asian (20°N–55°N, 70°E–150°E, bottom) total AOD at 550 nm in CanESM5.0-p2, and CanESM5.1-p1 and CanESM5.1-p2, and in remotely-sensed observations (black and grey). For the models, solid lines indicate ensemble medians and shaded envelopes indicate the 5th-to-95th percentile range across ensemble members. The ensemble spread of CanESM5.1 p1 is omitted for visual clarity but is similar to that of CanESM5.1 p2.

Figure 8 demonstrates the improvement in the spatial pattern and spatial variability of mineral dust optical depth from
CanESM5.0 to CanESM5.1, and in comparison to remotely sensed observations and other CMIP6 models. Observations of dust optical depth are more uncertain than those of total optical depth, and the observations used here are therefore less reliable than those shown in Figure 7; these limitations are described in Appendix B.

The first panel of Figure 8 shows the coefficient of determination ($R^2$, which indicates how close the distributions fall to a 1:1 relationship), between the patterns of 2007–2014 averaged dust optical depth fields in various model simulations and
observational datasets, following Figure 13h of Zhao et al. (2022). The CMIP6 models included here were selected on the basis of data availability. Rows correspond to different reference observations. Vertical bars give the $R^2$ of the other observations against each reference observation, and can be considered a benchmark of what performance is reasonable to expect from the models. The second panel of Figure 8 shows a Taylor diagram (Taylor, 2001) of dust optical depths, using as reference the average of the six observational datasets. Note that the "correlation" axis in the Taylor diagram refers to the Pearson
correlation coefficient of the patterns, which quantifies how tight the scatter between two quantities is but does not compare the magnitudes of their values. Given the data limitations described in Appendix B, precise position on this diagram should be considered with skepticism, but relative groupings are robust. The left- and right-hand panels of Figure 8 together demonstrate that while development from CanESM5.0 to CanESM5.1 brought dramatic improvement in the spatial variability of dust



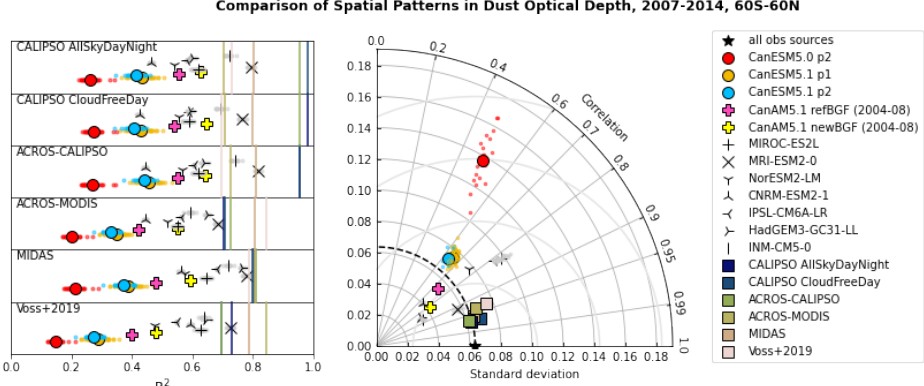

**Figure 8.** Improvements in the 2007–2014 mean spatial pattern and variability of global dust optical depth from CanESM5.0 to CanESM5.1, compared to observational datasets and select CMIP6 models. Left: correlation between simulated datasets (points) and six remotely-sensed observational datasets. Each row corresponds to a different reference observation, and vertical bars show the correlation between that reference observation and the other five. Right: Taylor diagram comparing simulations and observations to the average dust optical depth field across observational products.

(further illustrated in Figure A12), it brought only modest improvement in the mean spatial pattern. The spatial correlation of
both CanESM5.0 and CanESM5.1 with observations remains lower than for other CMIP6 models.

One factor that influences dust is the bare ground fraction, used to determine potential mineral dust sources in the models. An interpolation error has been identified in the bare ground mask used in CanESM5.0 and CanESM5.1 which led to errors in the input emissions. This issue is investigated by comparing two atmosphere-only simulations in which the atmosphere is nudged to reanalysis. An advantage of such runs is they more closely reproduce the observed meteorological conditions, which
have a large impact on dust. This allows for a comparison of dust to observations while controlling for internal variability in the atmospheric winds. Comparison of such atmosphere-only simulations with and without corrected bare ground fraction (denoted by, respectively, the yellow and pink plus sign in Figure 8) suggests that the correction improves the dust simulation, with the Pearson correlation coefficient increasing from 0.73 with the interpolation error to 0.80 without the error (Figure 8b). It is important to note, however, that the hybridization tuning parameters used in this simulation were not updated to
reflect the corrected bare ground fraction. Although the spatial pattern appears improved in this test simulation, the variability is substantially reduced such that the RMSE is approximately unchanged. While these results are suggestive, they are also preliminary and further analysis is required to determine whether corrections to the bare ground fraction are sufficient to make the spatial pattern of dust in CanESM as realistic as that of most other CMIP6 models, as suggested by Figure 8a.

## 5.2 Differences between CanESM5.1-p1 and CanESM5.1-p2

The p2 variant of CanESM5.1 was obtained by systematically retuning the atmospheric component in an attempt to reduce biases in ENSO characteristics and historical warming (section 3.2). This section provides a detailed description of the ENSO

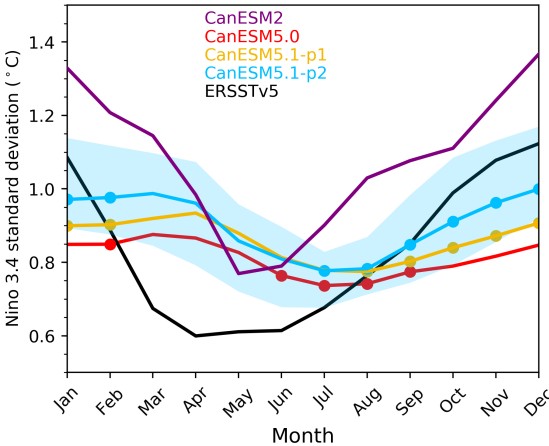

**Figure 9.** Standard deviations of interannual variability of the NINO3.4 index (the average SST anomaly in the region bounded by 5°S, 5°N, 170°W, 120°W) for each calendar month in 1950–2014. For CanESM5.1-p2, the shading represents the range over 10 ensemble members. Dots indicate months for which the 10 ensemble ranges bracket the observations from ERSSTv5 (Huang et al., 2017).

characteristics and climate sensitivity in various CanESM versions, with a focus on the changes that are the result of the atmospheric tuning in the p2 variant of CanESM5.1 compared to the p1 variant.

### 5.2.1 ENSO

The ENSO phenomenon, driven by atmosphere-ocean interactions in the near-equatorial Pacific Ocean, is the strongest and most wide-reaching pattern of interannual climate variability. A long-standing objective has therefore been for coupled climate and Earth system models to simulate ENSO realistically in order to better understand and predict ENSO, its global impacts, and likely changes in a warming climate.

Many metrics have been devised to describe various aspects of modelled ENSO variability and associated physical processes
(e.g., Planton et al., 2021). One common bias is that equatorial Pacific SST anomalies associated with warm (El Niño) and cool (La Niña) episodes often extend significantly farther westward than is observed (Jiang et al., 2021). This is the case for CanESM5.0 (Swart et al., 2019b) and the updated versions considered here, as well as for earlier CCCma model versions (Merryfield et al., 2013). However, other aspects of ENSO variability differ between CanESM5.0 and its immediate predecessors, CanESM2 and CanCM4[5]. Of particular note is that the seasonal cycle of ENSO SST variability is relatively accurate in
CanESM2 and CanCM4, with strong seasonal differences peaking in December as observed (Guilyardi et al., 2012; Merryfield et al., 2013), and overall ENSO amplitude slightly overestimated, as shown for CanESM2 in Figure 9. By comparison, Figure 9 also shows that ENSO in CanESM5.0 is too weak and displays little seasonal variation with a minimum in boreal summer instead of spring and no distinct winter peak, as also reported in Swart et al. (2019b), Planton et al. (2021), and Eyring et al.

---

[5]CanCM4 is almost identical to CanESM2 except that it does not include an interactive carbon cycle. CanCM4 has contributed to the decadal prediction experiments of CMIP5, as well as ECCC's operational seasonal and decadal forecasts.



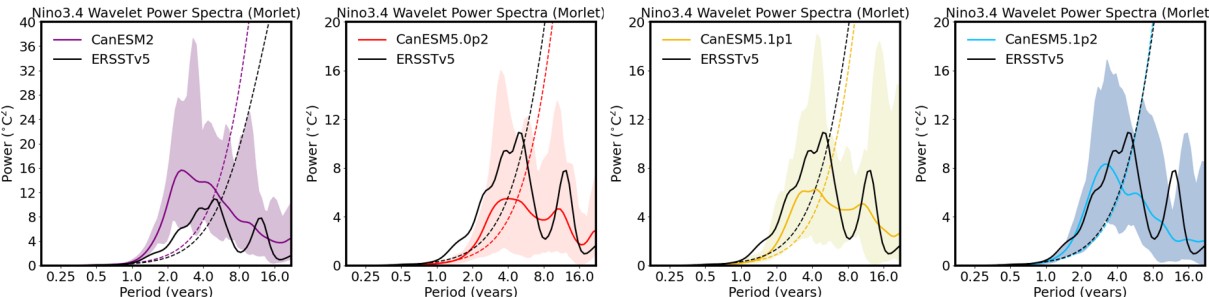

**Figure 10.** Power spectra of detrended monthly NINO 3.4 anomalies during 1950–2014 for (left to right): CanESM2, CanESM5.0, CanESM5.1-p1, CanESM5.1-p2. Black curves represent spectra of the observed time series from ERSSTv5, colored curves the mean of the spectra from 20 ensemble members, and shading the range of those spectra. Spectra are obtained from wavelet transforms based on the Morlet wavelet (Torrence and Compo, 1998), and dashed curves represent the 95% confidence level for an AR(1)-based estimate of statistical significance. Note the different vertical scale for CanESM2.

(2021). This likely bears on why experimental seasonal predictions from CanESM5.0 (not shown) are considerably less skillful

at predicting future ENSO evolution out to 12 months than CanCM4, which has relatively high ENSO prediction skill relative to other operational seasonal prediction models (e.g., Ham et al., 2019). On the other hand, Figure 10 shows that the distribution of ENSO variability across time scales is more realistic in CanESM5.0, with the CanESM2 spectral peak occurring at shorter periods than for the observed spectrum, whereas for CanESM5.0 the distribution of spectral power is closer to that of the observed time series.

While the p1 variant of CanESM5.1 has slightly higher ENSO variability than CanESM5.0, its seasonal cycle peaks in March as in CanESM5.0, which is inconsistent with the December peak in observations (Figure 9). The impact of the atmospheric tuning on ENSO variability can be seen by comparing the p1 and p2 variants of CanESM5.1. Figures 9 and 10 show that in CanESM5.1-p2, the ENSO amplitude is indeed higher than in CanESM5.1-p1, and that the seasonal cycle is somewhat improved with a December peak, although overall the seasonal variation remains too weak.

Some insights into why ENSO seasonality is too weak in these CanESM5 versions can be gained from several recent studies comparing ENSO and climatological SST seasonality in CMIP5 and CMIP6 models. The climatological seasonal cycle (annual mean subtracted) of equatorial Pacific SST from observations and the three CanESM5 versions are compared in Figure 11. All are concentrated in the eastern part of the domain where the thermocline is relatively shallow. Notably, the observed seasonal cycle primarily consists of the first annual harmonic despite the semi-annual cycle of insolation, whereas all CanESM5 versions

display a distinct semi-annual component in the east. Song et al. (2020) evaluated the simulated Eastern equatorial Pacific SST seasonal cycle in CMIP5 and CMIP6 models. They found that the correlation between the simulated and observed seasonal cycle in the region 110°–85°W where the seasonal cycle is strongest is only 0.55 for CanESM5.0, the lowest among 17 CMIP6 models considered, whereas for CanESM2 it is much higher (0.87). This suggests a connection between how realistically the



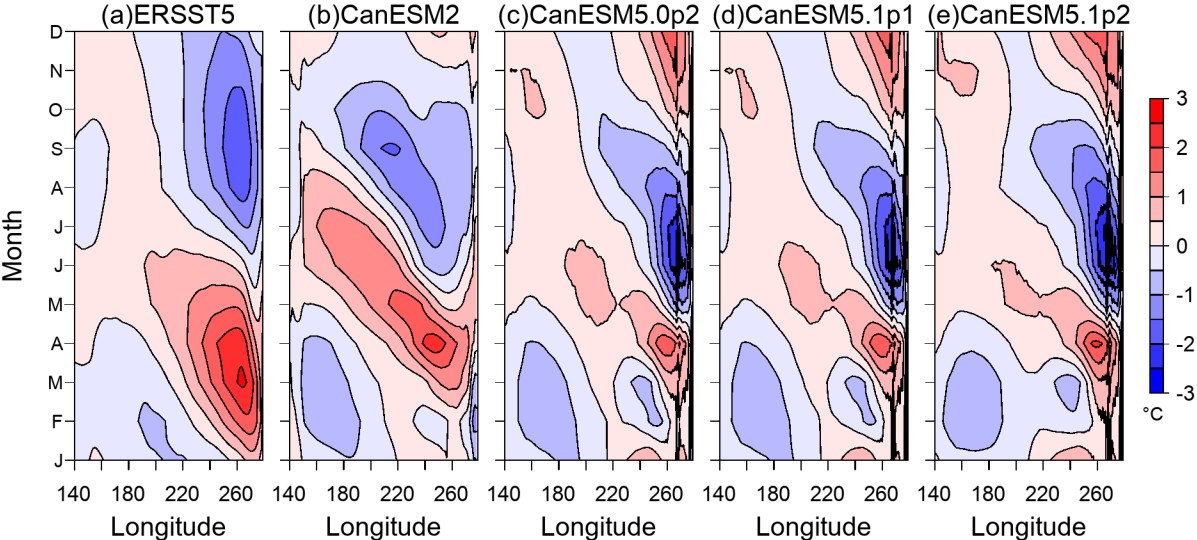

**Figure 11.** Mean seasonal cycle of equatorial Pacific SST during 1950-2014 for a) ERSSTv5. b) CanESM2. c) CanESM5.0p-2. d) CanESM5.1-p1. e) CanESM5.1-p2. Each model version is represented by the mean of 10 ensemble members.

equatorial Pacific SST seasonal cycle and ENSO seasonality are modelled, in accordance with studies indicating that they are
linked (e.g., Neelin et al., 2000).

A pair of additional studies have examined ENSO seasonality in terms of its phase-locking to the mean seasonal cycle across
numerous CMIP5 and CMIP6 models. Liu et al. (2021) showed that the month of peak ENSO variability and the distribution
over different calendar months in which ENSO events peak in CanESM2 are among the most realistic of all models considered
whereas CanESM5 is among the most unrealistic, and argued that the fidelity of the simulated SST seasonal cycle may be
a major factor responsible for such differences. Liao et al. (2021) applied different metrics, but ranked the performances of
CanESM2 and CanESM5 in simulating ENSO seasonality similarly to Liu et al. (2021), and furthermore associated ENSO
seasonality biases with biases in representing the seasonal zonal equatorial SST gradient. They found the temporal correlation
of observed and modeled seasonality of a metric describing the zonal gradient[6] to be among the lowest in CanESM5 of all
models considered. Such a calculation based on the CanESM5 versions considered here and the OISSTv2 dataset (Reynolds
et al., 2007) for 1982–2021 yields the same value of 0.49 for CanESM5.1-p1 as for CanESM5.0. This correlation is modestly
improved to 0.56 in CanESM5.1-p2, in accordance with the similarly modest improvement in ENSO seasonality in Figure 9.

The hypothesis that ENSO seasonality biases across CanESM5 versions are tied to biases in the simulated mean seasonal
cycle, as strongly suggested by the above results, raises the question of how those climatological biases can be reduced.
CanESM5 employs different atmosphere and ocean model components than CanESM2, making it difficult to assess whether
one or the other is primarily responsible for the degradation of climatological and ENSO seasonality between these model

---

[6]Diff_CE, defined as SST averaged over 2°S–2°N, 160°E–160°W minus SST averaged over 2°S–2°N, 120°W–90°W





versions. However, comparisons presented in Merryfield et al. (2013) between CanCM4, whose ENSO-related metrics are nearly the same as for CanESM2 according to Planton et al. (2021), and the earlier model version CanCM3 provide some insights. In particular, the mean seasonal cycle of equatorial Pacific SST is very similar between these two coupled models, which employ the same CanOM4 ocean model but different atmospheric model versions. This strongly suggests that the change

from CanOM4 in CanESM2 to CanNEMO in CanESM5 may be the main underlying reason for the equatorial Pacific seasonal cycle changes between these two models, and the relative insensitivity to atmospheric model differences in the CanESM5 versions described here aligns with this view. Which specific attributes of CanNEMO should be examined in this context is not obvious because the mean seasonal cycle of equatorial Pacific SST is governed by a complex interplay of oceanic physical processes (Wang and McPhaden, 1999), indicating that considerable model experimentation will likely be required.

**5.2.2 Climate sensitivity**

A second metric that was used to tune the p2 version of CanESM5.1 was historical warming, (which was unrealistically high in CanESM5.0 Swart et al. (2019b)) and is about 20% weaker than in the p1 version of CanESM5.1 (section 4.2). Here we quantify the associated change in climate sensitivity, and compare that with the climate sensitivity in other versions of CanESM. The metric that we use is the Effective Climate Sensitivity (EffCS), which is based on the regression method of

Gregory et al. (2004). In addition, we decompose Top-Of-Atmosphere (TOA) radiative feedbacks using radiative kernels. Non-cloud radiative feedbacks were calculated using radiative kernels from the Community Atmosphere Model (CAM3, Shell et al. (2008)). Climate responses for temperature, water vapour, surface albedo, and cloud radiative effects [CRE, longwave (LW) and shortwave (SW)] were calculated as the mean over years 125–150 of an abrupt $4xCO_2$ simulation relative to a 100-year mean from a pre-industrial control simulation. Cloud feedbacks were calculated using the adjusted cloud radiative effect (Soden

et al., 2008), where the CRE is adjusted according to differences between clear and total sky non-cloud feedbacks (Chung and Soden, 2015). The adjusted CRE method assumes that the clear sky direct (or instantaneous) radiative forcing (RF) can be related to the total sky IRF using a globally uniform proportionality constant of 1.16 (Soden et al., 2008).

As described in Swart et al. (2019b) and Virgin et al. (2021) and shown in Figure 12a, CanESM5.0 has an EffCS of 5.65K, which is 54% higher than its predecessor CanESM2 making it the CMIP6 model with the highest climate sensitivity (Zelinka

et al., 2020). Virgin et al. (2021) showed that while no single feedback fully explains this increase, the increase in the shortwave cloud feedback explains over half (1.08 K), as shown in Figure 12b. They further showed that this increase in shortwave cloud feedback comes primarily from a decreased boundary layer cloud fraction over subtropical eastern ocean basins and that this is also reflected in the warming rate asymmetry in the Pacific ocean, where the eastern versus western equatorial Pacific warming difference was larger in CanESM5.0 than in CanESM2 - also known as the pattern effect (Dong et al., 2020).

As expected based on the almost identical tuning settings of their atmospheric components, the EffCS and associated feedbacks in CanESM5.1-p1 are almost identical to those in CanESM5.0. By contrast, Figure 12a shows that the atmospheric tuning in CanESM5.1-p2 reduced the EffCS to 4.09 K, a reduction of 28%. TOA feedback decomposition reveals that this is due to a decrease in the SW cloud feedback, which is substantially weaker relative to CanESM5.1-p1/CanESM5.0 (Figure 12b).



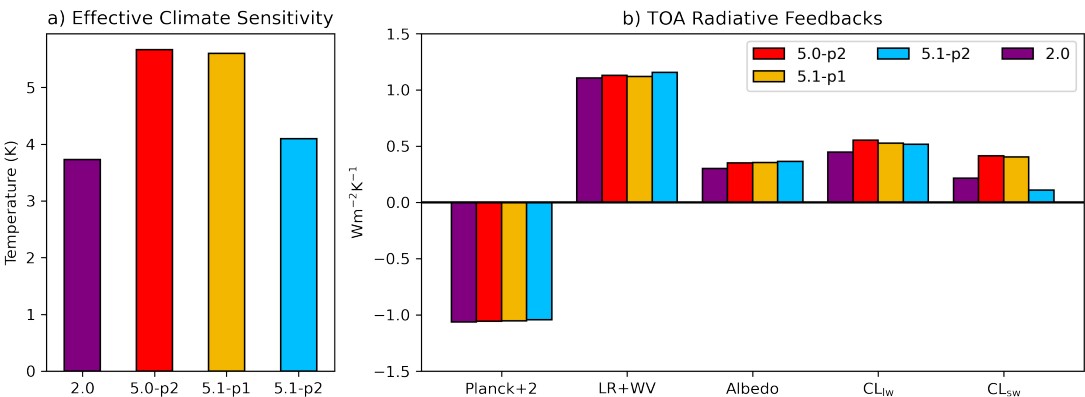

**Figure 12.** a) Effective Climate Sensitivity (EffCS) and b) Global, annual mean TOA radiative feedbacks in Wm$^{-2}$K$^{-1}$. From left to right, feedbacks are listed as: Planck +2 (for display purposes), Lapse Rate + Water Vapour, Surface Albedo, Longwave Cloud, Shortwave Cloud.

Previous studies have shown that SW cloud feedback and climate sensitivity across CMIP5 and CMIP6 models correlate well with the shallowness of low cloud in weakly subsiding tropical regions (Brient et al., 2016; Liang et al., 2022). It appears that CanESM5.0 has the strongest high bias in Brient Cloud Shallowness (BCS;  Brient et al., 2016) among all the CMIP6 models (Figure 13c), contributing to its high climate sensitivity. The BCS of CanESM5.1-p1 is slightly lower than that of CanESM5.0, which may be due to the fact that the retuning of the hybridization parameters described in Section 3.2 also affects moisture.

The atmospheric retuning in CanESM5.1-p2 resulted in an even lower BCS and hence a slightly smaller high bias relative to observations, which contributes modestly to CanESM5.1-p2's lower climate sensitivity. These results suggest that tuning future model versions to further reduce BCS would make such model versions agree better with observations in terms of BCS itself as well as in terms of historical warming (Figure 13a). A similar analysis of the sensitivity of cloud fraction and cloud albedo to SST variations over the tropics, which also correlate with SW cloud feedback (Figure 13b and d), shows only relatively small

changes between CanESM5.1-p1 and CanESM5.1-p2 which explain only a small fraction of the decreased climate sensitivity. Moreover the changes in these metrics make CanESM5.1-p2 less consistent with observations than CanESM5.1-p1. These results suggest that tuning future model versions to reduce the amount by which tropical cloud fraction decreases with SST warming (i.e. to further increase MBLC and BCA metrics) should be avoided.

**5.3   Other CanESM5.0/CanESM5.1 biases and possible avenues to tackle them**

**5.3.1   North Atlantic biases**

The North Atlantic is characterized by prominent biases in CanESM5.0, which occur throughout the year, but are largest towards the end of winter (Figure 14). By March, there is excessive sea-ice covering the Labrador Sea and Denmark Strait, a cold bias in sea surface temperature (SST) and a fresh bias in sea surface salinity (SSS). A corresponding bias exists in surface



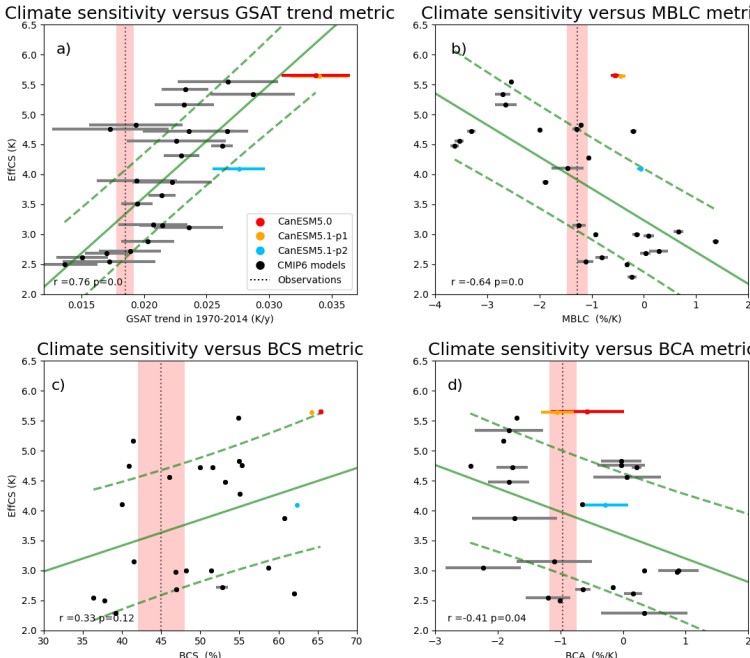

**Figure 13.** Scatterplots showing relationships between EffCS and observable metrics that have previously been shown to correlate with climate sensitivity across multi-model ensembles, in CanESM5 model versions and CMIP6 models: (a) the 1970–2014 global mean near-surface air temperature trend; (b) the sensitivity of marine boundary layer cloud fraction to SST changes in subsiding regions over oceans between $20°$ and $40°$ in both hemispheres (MBLC; Zhai et al., 2015); (c) Brient Cloud Shallowness, defined as the ratio of cloud fraction below 900 hPa to that below 800 hPa over weakly subsiding tropical ocean regions (BCS; Brient et al., 2016); and (d) Brient Cloud Albedo, defined as the sensitivity of the deseasonalized shortwave cloud albedo to SST changes over the tropical oceans (BCA; Brient and Schneider, 2016). Error bars show $\pm 1$ standard deviations across realizations (i.e., initial condition ensembles) for each model. The dashed vertical lines show the mean observed value and the red shading shows its $\pm 1$ standard deviation, defined as follows: for GSAT in (a), from the 200-member HadCRUT5 ensemble; for MBLC in (b), the estimate of internal variability as described in Zhai et al. (2015); for BCS in (c), one seasonal standard deviation as described by Brient et al. (2016); for BCA in (d), an estimate of internal variability obtained by bootstraping random samples of varying time series length (Brient and Schneider, 2016). The correlation coefficients (between constraints and EffCS over all CMIP6 models) and corresponding p-values are reported in the lower left corner of each panel. The green dashed lines in each panel show the 66% confidence interval of the linear regression model. In all four cases a two-sample t-means test indicated that metrics were significantly different between CanESM5.1-p1 and CanESM5.1-p2, at the 0.01 level.





air temperatures, which are too cold over the North Atlantic in both winter and summer (Figure 2). CanESM2 also exhibited
some of these biases, but they were in general far less pronounced (Figure 14). North Atlantic Deep Water formation, one of the
primary drivers of the Atlantic Meridional Overturning Circulation (AMOC), relies on deep convection within the Labrador
and Greenland-Iceland-Nordic Seas (GIN) seas. The presence of unrealistically high sea ice coverage over the Labrador Sea
in CanESM5 inhibits the air-sea interactions that destabilize the surface layer and lead to deep mixing. Within CanESM5 the
AMOC is fairly weak (∼13 Sv in the climatological average) and deep convection is confined exclusively to the shelves of the
GIN seas.

The biases in sea-ice, SST and SSS in CanESM5 are clearly interconnected, but determining cause and effect in the coupled
CanESM5 system is far from trivial. During the development of CanESM5.0, two sensitivity tests with the coupled model were
performed that provided some insights. The first test involved running piControl runs with alternative strengths of vertical dif-
fusivity. These showed very similar biases, suggesting that the biases are not the result of improperly tuned vertical diffusivity.
The second test involved restarting the ocean model from rest after the atmosphere had spun up. In this test run the sea ice,
SST and SSS biases developed within a few decades, which pointed to atmospheric biases as a potential source of the coupled
biases.

To learn more on the nature of the coupled biases, we here present additional sensitivity tests, starting with more constrained,
ocean-only experiments. In the standard OMIP experiment submitted to CMIP6, the same ocean model configuration as used
in CanESM5 is driven by reanalysis-based atmospheric forcing through bulk formulae, with runoff from observations and
relaxation towards observed SSS (Griffies et al., 2016). Under observed atmospheric forcing, no such biases in sea-ice, SST,
and SSS exist (Figure 15a). This indicates that the ocean and sea-ice components of CanESM5 are able to simulate a more
realistic state given realistic surface forcing. Next, we repeat the standard OMIP experiment, but instead of reanalysis forcing,
we provide forcing (including SSS restoring) from a CanESM5 historical simulation. Under CanESM5 forcing, the ocean-only
model reproduces the biases seen in the coupled model (Figure 15b). While these ocean-only experiments are instructive, they
lack coupled feedbacks, and do not provide definitive evidence about the cause of the biases, or their solutions. In the coupled
modelling system, we have explored three hypotheses regarding the origin of the biases:

1. Issues in sea-ice thermodynamics or coupling encourage excessive ice growth and resulting SSS/SST biases;

2. Runoff bias impacts SSS and enhances stratification, reducing convection and enabling ice growth;

3. Atmospheric circulation bias impact ocean tracer transport and surface properties.

As a test of the sensitivity to sea-ice thermodynamics, experiments 1 and 2 from Table 2, that adjust snow density and thermal
conductivity within their uncertainty ranges, were carried into the historical period (and are otherwise identical to CanESM5
historical runs). Using a lower specific density and conductivity for snow does reduce the sea-ice bias, but does not remove it,
or even recover a state comparable to CanESM2 (Figure 15d). Variable snow density and conductivities lead to an even smaller
improvement (Figure 15c).

River runoff in CanESM5 is derived from the CLASS land surface model, wherein the precipitation minus evaporation (P-E)
residual is routed downhill to the coast, and input as runoff where the atmospheric land fraction falls below 50% (Swart et al.,



**Figure 14.** North Atlantic biases during March in CanESM2 and CanESM5, relative to observations [named in brackets] , in sea-ice concentration [NSIDC] (a, b), sea surface temperature [HadISST] (c, d), sea surface salinity [WOA13] (e, f) and sea level pressure [ERA5] (g, h). In all cases, biases are shown as colours, and were computed from historical experiments in CanESM, over the period 1981 to 2010 for the month of March. Black contours show the March climatologies from the observations.







**Figure 15.** North Atlantic sea-ice concentration bias during March, relative to NSIDC observations in various experiments: a) the CMIP6 OMIP experiment, forced by reanalysis; b) an OMIP-like experiment, but forced by CanESM5 historical forcing; c) coupled historical experiment using variable snow density and conductivity, d) coupled experiment using a modified fixed snow density and conductivity; e) coupled experiment using observed runoff and f) coupled experiment, with nudging of atmospheric vorticity and divergence to ERA5, starting in 1980. Black solid contours gives the 15% concentration contour in the experiment, the dash black contour is the 15% contour from the CanESM5 default historical run and the cyan contour is the 15% contour from the NSIDC observations.



2019b). This runoff field is then conservatively remapped in the coupler from the roughly 3°atmospheric physics grid, to the nominally 1°ORCA1 tripolar grid. No further adjustment is undertaken. Due to the resolution difference between the models, this procedure effectively leads to runoff being distributed over several ocean grid-cells away from the coast. In addition, the runoff magnitude can be biased by the P-E field in the atmospheric model. To test whether the amount or distribution of runoff in CanESM5 lead to the sea-ice biases, we did an experiment where the CLASS-based runoff was ignored, and the observationally based runoff from the OMIP experiment was read from file instead. This ultimately had little impact on the sea-ice bias (Figure 15e).

Finally, synoptic-scale atmospheric circulation biases over the North Atlantic could play a role by influencing the subpolar ocean gyre. Climatological sea level pressure in CanESM5.0 shows an Icelandic Low that is displaced northeast relative to observations, and an over-expressed gradient between the Icelandic Low and the Azores High (Figure 14h). Similar biases exist in AMIP mode (not shown), suggesting this is a feature of the atmospheric model, not of the coupled climate. The wind stress associated with these pressure gradients would encourage a geostrophic oceanic circulation comprising a North Atlantic Current that remains coherent over the basin and extends the subpolar gyre too far to the east. This would imply that warm, salty tropical waters that should be delivered to the Labrador Sea instead traverse a much longer distance, losing heat to the atmosphere and dissipating within the broader North Atlantic basin. In turn this would reduce the heat and salt flux into the Labrador Sea, thus making it more susceptible to freezing over and high stratification.

Testing such a hypothesis is a challenge. What we have done is to run coupled historical experiments where vorticity and divergence in the atmospheric model are spectrally nudged towards ERA5 reanalysis, only at larger length scales (T21 and coarser). The results from these experiments show a strong reduction in the Labrador sea-ice bias early on, but this improvement fades with time since the onset of the nudging, and leads to only modest improvement when considered over a 30 year period (Figure 15f). The challenge with these experiments is that the introduction of nudging perturbs the model climate. Therefore, interpretation is hard, because the model is not equilibrated, and over time the wind nudging drives the overall climate colder, making conditions more favorable to the return of the sea-ice bias. In the future, we will derive stationary bias corrections from these nudging runs, and use these to establish equilibrated climate states to test this hypothesis. Similarly, OMIP-like ocean-only experiments that systematically change out only the observed wind-stress for the CanESM5 wind stress are not definitive: the specification of other surface forcing from observations (e.g. surface air temperature and SSS) carries with it the imprint of observed sea-ice distributions, and the lack of coupled feedbacks suppresses the true effect that wind stress bias would have in the coupled model.

No definitive cause, or solution to the CanESM5 North Atlantic bias has been identified so far. As CanESM moves towards more automated parameter tuning, inline bias correction, a new dynamical core, and new ocean and atmospheric grids resolutions, we hope to arrive at versions with smaller circulation biases. Similarly, new approaches to sea-ice thermodynamics and coupling are being considered. Finally, while the ocean-only model in the configuration forced by observations does not show these issues, it is not certain that the ocean configuration does not contribute to the problem in the coupled model. Recent tests suggest that increasing horizontal ocean diffusivity can also help to reduce the bias by increasing the heat and salt flux into the Labrador Sea. Such ocean physics will be another element considered in future investigation of this issue.

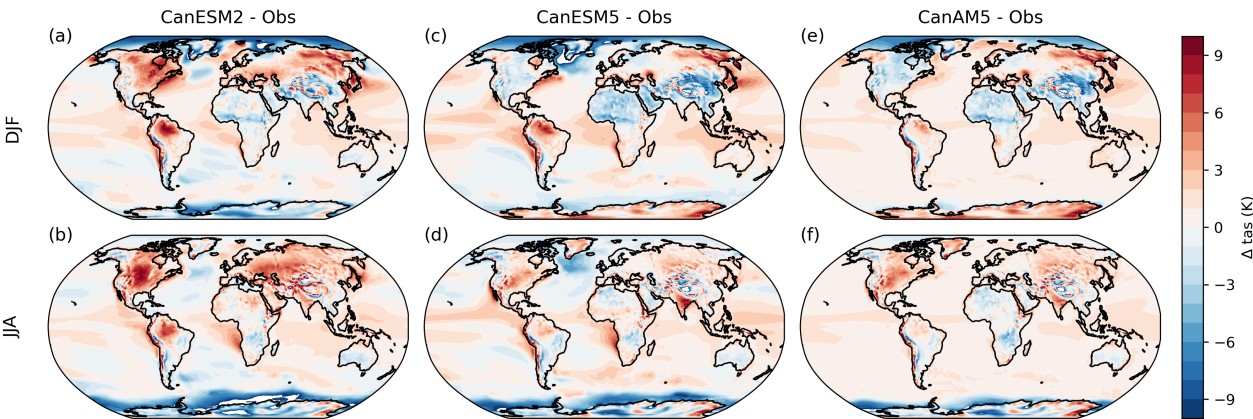

**Figure 16.** Surface air temperature bias over 1981–2010, relative to ERA5, in CanESM2 (a, b), CanESM5.0-p2 (c, d) and CanAM5.0 (e, f), for the DJF (top row) and JJA (bottom row) seasons. Note that CanAM5 is an AMIP simulation with no interactive ocean or ice components.

### 5.3.2 Winter cold bias above sea ice

CanESM5.0 exhibits large cold biases in surface air temperature (SAT) over the region covered by sea-ice during the winter
season, as described in Swart et al. (2019b) and shown in Figure 16c,d. This bias also persists in CanESM5.1 (Figure 2). In contrast, there are not large biases in SST over these regions (see Figure 15 of Swart et al., 2019b). In CanESM5, the surface temperature of sea-ice / snow is computed in CanAM, while the calculation in the sea-ice model, LIM2, is not used prognostically (Swart et al., 2019b). This suggests that the surface temperature computation in CanAM is the source of the issue. This reasoning is supported by the fact that the previous coupled model version, CanESM2, which contained the same
CanAM ground temperature computation, but completely different sea-ice properties (and a different sea-ice model), also exhibited a similar cold bias over the winter sea-ice covered area (Figure 16a,b). In fact, as shown in Figure 7 of Merryfield et al. (2013), a similar bias existed in CanCM3, two full atmospheric model generations before CanESM5 (note they also show the bias for CanCM4, which is a physical-only version of CanESM2). AMIP runs (with specified SST and ice-cover) show the same winter-hemisphere cold SAT bias over sea-ice, demonstrating that this issue arises in CanAM, as opposed to in the ocean
or sea-ice models (Figure 16e,f). Given its persistence across model versions and experiment types, this cold bias above sea-ice is *model systemic*. The bias has been identified in surface air temperature, which is a diagnostic, rather than a prognostic model field. A very similar bias exists if the comparison with observations is done based on ground temperature (not shown).

The ground temperature over sea-ice/snow in CanAM evolves according to:

$$gt_{(t)} = gt_{(t-1)} + (fsg + flg - hsens - hlat + hsea) * delt/cice \tag{1}$$

where $t$ is the time step count, $gt$ is the ground temperature (K), $fsg$ is the net absorbed solar flux, $flg$ is the net long wave flux, $hsens$ is the sensible heat flux, $hlat$ is the latent heat flux, and $hsea$ is the conductive heat flux through ice/snow from





| Experiment | Snow density (kg m$^{-3}$) | Ice Conductivity W m$^{-1}$K$^{-1}$ | Snow conductivity W m$^{-1}$K$^{-1}$ |
|------------|---------------------------|-------------------------------------|--------------------------------------|
| default | 330 | 2.25 | 0.31 |
| 1 | 300 | 2.25 | 0.2 |
| 2 | f(thick) | 2.25 | f(density) |
| 3 | 330 | 2.25 | 0.15 |
| 4 | 330 | 2.25 | 0.46 |
| 5 | 330 | 2.0 | 0.31 |
| 6 | 330 | 2.75 | 0.31 |

**Table 2.** Pre-industrial control experiments with altered sea-ice therodynamics in CanAM. In experiment 2, snow density is a function of snow thickness, and snow conductivity is a empirically based function of snow density (this scheme was used in CanESM2). In other experiments, snow density and ice/snow conductivities are fixed constants. Experiments 1 and 2 were used as initial conditions (restarts) for historical experiments with the same parameter modifications.

the ocean (all Wm$^{-2}$), $delt$ is the time step duration (s) and $cice$ is the heat capacity of the upper 10 cm of the solid surface (J m$^{-2}$ K$^{-1}$).

Given the winter (and high latitude) manifestation of this problem, short wave radiation contributions are unlikely as it occurs during polar night, and we have focused on other parts of the surface energy balance. The conduction of heat through sea-ice and snow that features prominently in the winter energy balance (i.e., the $hsea$ term in Eq. 1), has been investigated. A series of control experiments were conducted, systematically altering the ice and snow conductivities used in CanAM (Table 2).

Of these experiments, the largest effect is seen when decreasing the snow conductivity, which leads to some warming of SATs around the edges of Arctic sea-ice in DJF (Figure A13). However, the tested thermodynamic changes only have a small impact relative to the large size of the existing cold bias, and none of them address the systematic and widespread nature of the cold bias (i.e. over all sea-ice in the winter hemisphere). Possible alternative explanations for the cold bias above sea-ice are issues in the non-solar radiative or turbulent fluxes. Future work will focus on examining these alternative hypotheses.

### 5.3.3 Stratospheric circulation

In this section, we investigate two atmospheric circulation biases in the stratosphere that have been shown to be important for the simulation of boreal winter surface climate variability and climate change. The first metric of interest is the number of sudden stratospheric warmings (SSWs). SSWs are rapid breakdowns of the westerly flow (or polar vortex) in the polar winter stratosphere, which tend to be followed by a persistent anomalous tropospheric circulation pattern and associated surface temperature and precipitation patterns that can last up to two months (e.g., Baldwin et al., 2021). It has been shown that SSWs




are a main source of subseasonal to seasonal predictability for surface climate (e.g., Sigmond et al., 2013) Hence, a realistic simulation of SSWs is important for maximizing the skill of seasonal forecasts, particularly in boreal winter and early spring.

Previous multi-model studies have shown that there is a large spread in the simulated frequency of SSWs between different climate models. So called "low-top" models, often defined as those models whose model lid is lower than 0.1 hPa as in Domeisen et al. (2020), tend to underestimate stratospheric variability (Charlton-Perez et al., 2013), and consequently, the
number of SSWs, while "high-top" models tend to simulate a more realistic SSW frequency. The results of Kim et al. (2017) indicate that CanESM2, a "low-top" model according to previously mentioned criterion) is a noticeable exception showing an overestimation of the SSW frequency. By contrast, Ayarzagüena et al. (2020) indicates that CanESM5.0 underestimates the SSW frequency. However, it should be noted that the results of both Kim et al. (2017) and Ayarzagüena et al. (2020) are based on only on the first ensemble member, and that due to the large internal variability of the coupled stratosphere-
troposphere system there can be a large spread between individual ensemble members (Polvani et al., 2017). We first investigate the robustness of the Kim et al. (2017) and Ayarzagüena et al. (2020) results by calculating the SSW frequency in large ensembles of CanESM2 and CanESM5.0 simulations. Figure 17 shows that averaged over a 50-member initial condition ensemble, CanESM2 simulates about 10 SSWs per decade, confirming the Kim et al. (2017) conclusion (based on the first ensemble member, plotted by the black dot), that CanESM2 overestimates the SSW frequency. Figure 17 also shows that
averaged over a 25-member ensemble, CanESM5.0 simulates too few SSWs, confirming the results of Ayarzagüena et al. (2020), which was based on the first ensemble member. However it should be noted that there is a large spread between ensemble members, and that the if Ayarzagüena et al. (2020) had picked a different ensemble member with a SSW frequency at the upper end of the range, they would not have identified CanESM5.0 as one of the models with too few SSWs. Figure 17 also shows that the SSW frequency in CanESM5.1-p1 very similar to that in CanESM5.0, while that in CanESM5.1-p2 is
slightly less. This is consistent with the stronger polar vortex shown in Figure A8c.

It is striking how different the simulated SSW frequency is in CanESM5.0 compared to CanESM2, given that the model lid height of both models is the same at 1 hPa. One notable difference in the atmospheric component between CanESM2 and CanESM5.0 is in the tuning of the free parameters in the orographic gravity wave drag (OGWD) scheme. The OGWD scheme used in both models is that of Scinocca and McFarlane (2000), which features two internal parameters that were changed
between CanESM2 and CanESM5.0: 1) $G(\nu)$, a multiplicative factor that scales the amount of gravity wave momentum flux produced by the interaction of the circulation with the topography, and 2) $Fr_{crit}$, the inverse critical Froude number, which sets the maximum non-dimensional amplitude that a parameterized wave may attain at launch, before low-level blocking onsets, and aloft before wave breaking begins and the wave begins to transfer its momentum to the background flow. Adjustments to these parameters were made to correct for the bias that CanESM2 simulated a too weak polar vortex (see table 1). The impact of
these OGWD parameter-value changes on the frequency of SSWs is quantified by an ensemble of 5 historical simulations with CanESM5.0 in which the values of $G(\nu)$ and $Fr_{crit}$ were reset back to those used in CanESM2, hereafter referred to as the CanESM5-G simulations. Figure 17 shows that in these simulations, the SSW frequency is very similar to that in CanESM2. This implies that the difference in simulated SSW frequency between CanESM2 and CanESM5.0 can be attributed to the change in OGWD tuning settings.





| Model | $G(\nu)$ | $F_{crit}$ |
|---|---|---|
| CanESM2 | 0.5 | 0.71 |
| CanESM5.0/CanAM5 | 1.0 | 0.22 |
| CanESM5.0-G | 0.5 | 0.71 |
| CanAM5-G1 | 0.33 | 0.71 |
| CanAM5-G2 | 0.4 | 0.71 |
| CanAM5-G3 | 0.26 | 0.71 |

**Table 3.** Settings of the scaling factor $G(\nu)$ and inverse Froude number $Fr_{crit}$ in the orographic gravity Wave paramaterization for various CanESM versions.

Having established the strong sensitivity of the SSW frequency to OGWD parameter values, we next perform a series of sensitivity tests in which the values of $G(\nu)$ and $Fr_{crit}$ are adjusted to determine if the low SSW frequency bias in CanESM5.0 could be eliminated. These tests were performed using multiple 5 member ensembles of atmosphere-only runs. These runs were branched off from the standard CanAM5 historical runs in 1950. Tuning is performed in atmosphere-only rather than fully coupled mode to speed up the process. This choice is motivated by the observation that the frequency of SSWs is very

similar in CanAM5 compared to CanESM5.0, as shown by Figure 17. Our first combination of OGWD tuning settings are those that have been found to perform best for a previous high top version of our model. The model version with these modified settings is labelled "CanAM5-G1". A second and third set of tuning runs is performed with slightly higher (CanAM5-G2) and slightly lower (CanAM5-G3) values of $G(\nu)$ (details in Table 3). Figure 17 shows that the settings in CanAM5-G1 yield an overestimation of the number of SSWs. Increasing the OGWD in CanAM5-G2 leads to increased deposition of

orographic gravity waves, decreasing the strength of the polar vortex and increasing the number of SSWs, further away from the observed frequency. The decreased OGWD in CanAM5-G3 compared to CanAM5-G1, by contrast, leads to decreased deposition of orographic gravity waves, increasing the strength of the polar vortex and decreasing the number of SSWs and eliminating the high SSW frequency bias in CanAM5-G1. In summary, by varying tuning parameters in the OGWD scheme we have determined a combination of parameter settings that has eliminated the SSW frequency bias in CanESM5.0. While

this combination of parameters have not been implemented in CanESM5.1, it is being considered for future CanESM versions as a way to eliminate SSW biases and maximize the skill that SSW provide in seasonal forecasts.

A second aspect of the stratospheric circulation that is of interest to the surface climate is the strength of the zonal-mean zonal winds in the region between the subtropical jet and the polar vortex (here defined as the wind at averaged over 40-60°N and 100-50 hPa, hereafter referred to as the "neck region winds"). Sigmond and Scinocca (2010) showed that the response of

the Northern Annular Mode (NAM) to increasing greenhouse gases (which has large impacts on patterns of regional climate change) is sensitive to biases in the strength of the neck region winds. In particular they found that overly weak neck region winds result in a barrier to equatorward propagation of large scale atmospheric waves. Climate models consistently show that increasing greenhouse gases result in increased vertical wave fluxes, and Sigmond and Scinocca (2010) showed that in a model

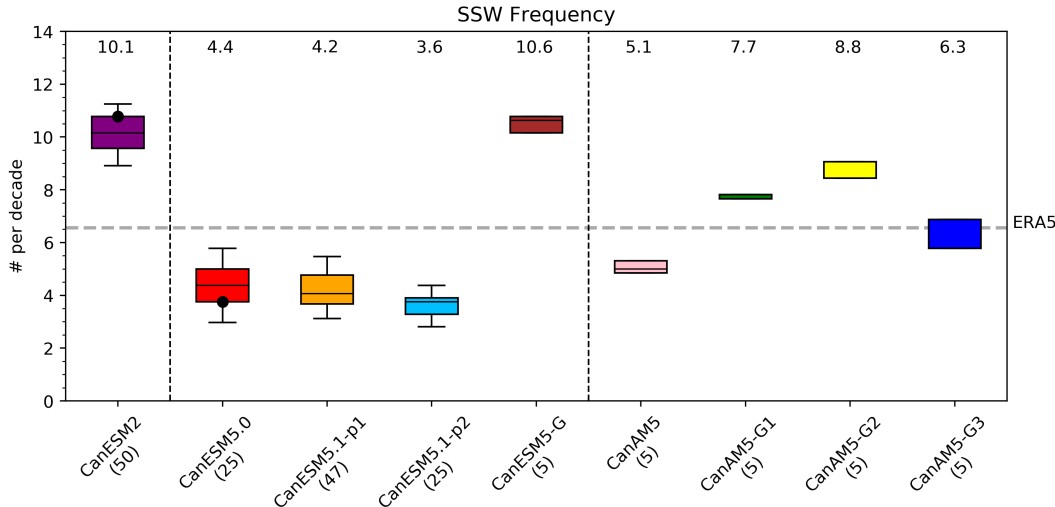

**Figure 17.** The 1950–2014 frequency of SSWs in various CanESM versions (details in text, with the number in parenthesis after the model name indicating the number of ensemble members) and in the ERA5 reanalysis. The boxes extend from the lower to upper quartile of the ensembles. The black lines represents the medians, the whiskers the 5–95% confidence range determined by bootstrapping and the numbers on top the ensemble mean values. The black dots for CanESM2 and CanESM5.0 indicate the first ensemble members, which were used in Kim et al. (2017) and Ayarzagüena et al. (2020), respectively.

version with overly weak neck region winds, these increased wave fluxes tend to be deposited at high latitudes, leading to a
reduction of the westerly winds and hence a negative NAM response. The opposite was true for a model version with relatively
strong neck region winds, which resulted in a positive NAM response to increasing greenhouse gases. This suggests that for
credible projections of future regional climate it is important for the neck region winds to be consistent with observations.

The strength of the neck regions winds in various versions of the CanESM is shown in Figure 18. It shows that while
CanESM2's neck region winds are close to those observed, CanESM5.0's neck region winds are too strong. In response to
increasing greenhouse gases Simpson et al. (2018) reported that the polar vortex response in CanESM2 is close to zero, while
Karpechko et al. (2022) showed that CanESM5.0 has one of the largest positive polar vortex response among the CMIP6
models. Hence, these findings are consistent with the positive correlation between neck region winds and the NAM response
to increasing greenhouse gases as identified by Sigmond and Scinocca (2010). While the neck region winds are similar in
CanESM5.1-p1 compared to CanESM5.0, they are even stronger in CanESM5.1-p2. It appears that the change in OGWD
settings between CanESM2 and CanESM5.0 can explain about 50% of the difference in neck region winds, as the neck region
wind strength in CanESM5.0-G is between that of CanESM2 and CanESM5.0 (Figure 18). The neck region winds in CanAM5
are slightly weaker than CanESM5.0, which implies that tuning efforts that target neck region winds should ideally be done
in the fully coupled model. Finally, the CanAM5 tuning runs show that while the OGWD combination of CanAM5-G3 led
to the most realistic SSW frequency, the bias in the neck region winds is the largest of the three considered OGWD setting



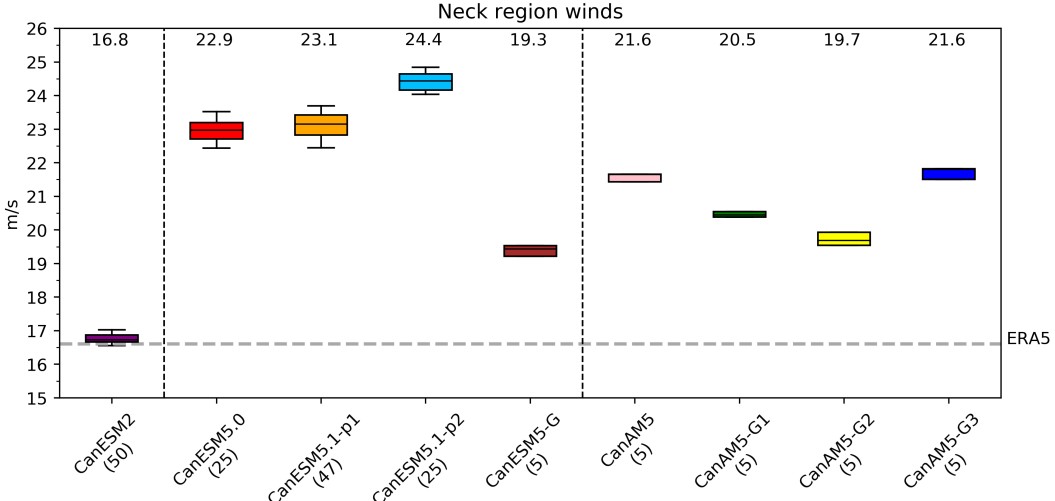

**Figure 18.** The 1950–2014 and December to February mean neck region winds in various versions of the Canadian Earth System Model (details in text, with the number in parenthesis after the model name indicating the number of ensemble members). The boxes extend from the lower to upper quartile of the ensembles. The black lines represents the medians, the whiskers the 5–95% confidence range determined by bootstrapping and the numbers on top the ensemble mean values.

combinations. This highlights intrinsic difficulties often encountered in tuning efforts: the improvement of one aspect of the circulation can often lead to the deterioration of another. Physically, this can indicate the absence, or misrepresentation, of a relevant process in the model. For the OGWD scheme in CanAM, a new orographic lift component is being added following the work of Lott (1999) and Gastineau et al. (2020) to better represent stationary planetary waves in CanAM. It is anticipated that improvement of the planetary waves in CanAM will better allow a simultaneous reduction of biases in both the basic state

(including neck region winds) and SSW biases in future versions of CanESM.

### 5.3.4 Land surface temperatures and albedo

Both CanESM5.0 and CanESM5.1 suffer from a cold bias over the Himalayan region that extends into Eastern China, most prominently in December–February (Figure 2). This bias is likely caused by two factors. First, a positive precipitation bias in the region leads to more snow and therefore also a positive land surface albedo bias in the region. Second, the current

snow fraction parameterization in the land component of CanESM5.0/5.1 does not take into account the effect of topography. Compared to regions with flat terrain, the same amount of snow should cover less area in mountainous regions because snow is not able to accumulate on steep terrain (Swenson and Lawrence, 2012). Since this effect is not yet parameterized in CanESM, snow covers a larger fraction than it realistically should in mountainous regions with steep terrain.

Figure 19a shows the bias in simulated annual-mean surface albedo in a CanESM5.0 historical simulation, compared to

the observation-based product CERES (Kato et al., 2013), for the period 2000–2013 for which the CERES data are available





over the southeastern Asian region. The surface albedo bias is largest over the Himalayan region. Figure 19b shows the bias in mean annual snow water equivalent from a historical CanESM5.0 simulation compared to the ECCC observation-based data (Mudryk, 2020), also for the period 2000–2013, and shows the large bias over the Tibetan plateau. This large amount of snow builds up over time as the model is spun up for conditions corresponding to the pre-industrial state and does not disappear over

the historical period. The grid cells in dark blue in Figure 19b over the Tibetan plateau region that exceed the 12 cm colourbar scale build up snow amounts varying from 70 to 310 cm in snow water equivalent units.

The land component in CanESM5.0/5.1 is represented by the Canadian Land Surface Scheme (CLASS; Verseghy et al., 1993) and the Canadian Terrestrial Ecosystem Model (CTEM; Arora and Boer, 2005). CLASS and CTEM simulate physical and biogeochemical processes, respectively, in the family of CanESM models. CLASS and CTEM will be succeeded in a

subsequent CanESM version by the Canadian Land Surface Scheme including Biogeochemical Cycles (CLASSIC) which couples these two models in a unified framework (Melton et al., 2020). CLASSIC includes a very similar version of physical processes, including snow-related physical processes, based on CLASS as in CanESM5.0/5.1.

Offline simulations of CLASSIC are routinely performed with observation-based meteorological data to evaluate the model's performance for primary water, energy, and $CO_2$ fluxes in an uncoupled mode where land-atmosphere feedbacks are absent.

These simulations have also contributed to the Global Carbon Project (Friedlingstein et al., 2022) since 2016. Figures 19c–h show the bias in these offline simulations in simulated annual-mean land surface albedo and snow water equivalent, compared to CERES and ECCC data, respectively, when CLASSIC is driven with meteorological data from CanESM5.0, CRU-JRA, and GSWP3. The simulated surface albedo and snow water equivalent values are averaged over the period 2000–2013. The CRU-JRA (v2.1.5) meteorological data set, which is the Japanese reanalysis (JRA) with monthly values adjusted to the CRU data,

provides 6-hourly values of seven meteorological variables that are required to drive a land model. These data are provided as part of the TRENDY protocol for land models that contribute to the Global Carbon Project (Friedlingstein et al., 2022). Three-hourly meteorological data from CanESM5 for the same seven variables are also saved to force CLASSIC offline (i.e., in uncoupled mode). Finally, the GSWP3 forcing data are based on a dynamical downscaling of the 20th century reanalysis (Compo et al., 2011) using a Global Spectral Model (GSM) run at about 50 km resolution and also available at the 6-hourly

temporal resolution. Positive biases in snow albedo and snow water equivalent over the Himalayan region are also simulated in the offline runs when CLASSIC is driven with CanESM5 and CRU-JRA meteorology (Figure 19c–f). The bias is much smaller for snow albedo and of the opposite sign (with a small magnitude) for snow water equivalent when CLASSIC is driven offline with the GSWP3 meteorology. Although other meteorological variables also play some role, the difference in CRU-JRA and GSWP3 runs is largely driven by higher precipitation in CRU-JRA dataset compared to the GSWP3 meteorological dataset

over the Himalayan region (Figure 20i,j), indicating that differences in precipitation in the two meteorological data sets are able to lead to differences in simulated bias in snow albedo.

Figure 20 shows the biases in simulated annual mean temperature and precipitation from CanESM5.0 compared to the CRU-JRA, GSWP3, and GPCP datasets averaged over the 1970–2014 period. Note that the CRU-JRA and GSWP3 are complete meteorological datasets that are available only over land (and are used to drive CLASSIC offline), while GPCP is an

observation-based precipitation dataset and is available for the whole globe. CanESM5.0 shows a large cold bias over the Hi-



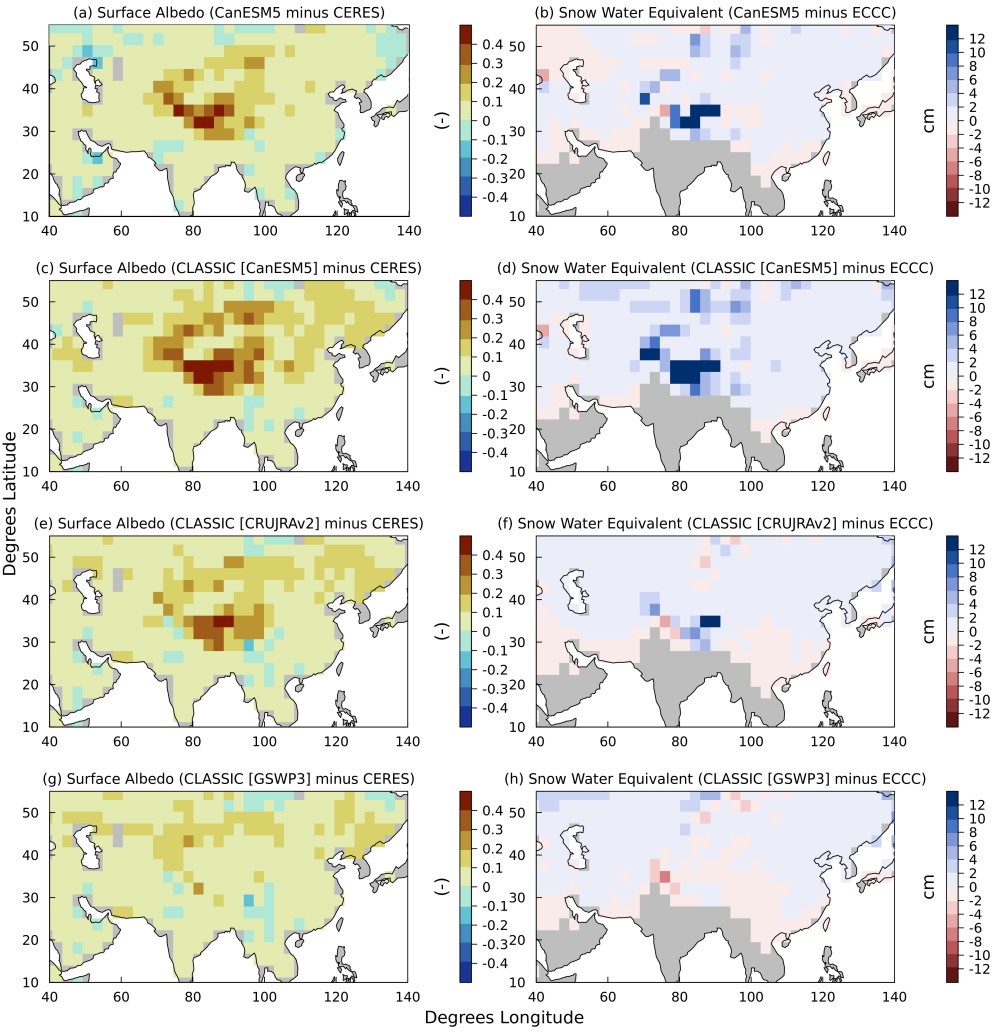

**Figure 19.** Annual-mean bias in simulated surface albedo for shortwave radiation (left column) and snow water equivalent (right column) compared to the CERES and ECCC observation-based products, respectively, over the southeastern Asian region for the 2000–2013 period. Panels (a) and (b) show the biases for CanESM5. Positive values indicate higher model values compared to the observation-based data. The remaining rows show the biases when the CLASSIC land model is driven offline by meteorological data from CanESM5 (panels c and d) and from the observation-based datasets CRU-JRA (panels e and f) and GSWP3 (panels g and h). Note that the largest positive biases in snow water equivalent exceed the colour scale shown (see text for details).





**Figure 20.** Bias in CanESM5 simulated annual-mean surface air temperature and precipitation compared to the CRU-JRA, GSWP3, and GPCP datasets over the southeastern Asian region for the 1970–2014 period. Panels (a) and (b) show the bias in CanESM5 simulated near-surface air temperature compared to the CRU-JRA and GSWP3 datasets, respectively. Panels (c) through (h) show the absolute and relative bias in CanESM5 simulated precipitation compared to the CRU-JRA, GSWP3, and GPCP datasets. Panel (i) and (j) show the absolute and relative precipitation differences between the CRU-JRA and GSWP3 datasets.





malayas region (around -4 to -5°C) compared to both CRU-JRA and GSWP3 datasets (Figure 20a,b) as mentioned earlier, and this cold bias extends into Eastern China. Panels (c) through (h) of Figure 20 show the absolute and relative bias in simulated CanESM5.0 precipitation compared to the CRU-JRA, GSWP3, and GPCP datasets. Overall the bias in CanESM5.0 precipitation is similar compared to all observation-based datasets. The relative bias is largest over India. The coarse T63 spatial

resolution ($\sim 2.8°$) of CanESM5.0 implies that topography is not well resolved and high mountain peaks are not well represented. As a result, higher than observed moisture is advected over the Himalayas by the eastward surface winds in this region (Figure 11 of Swart et al., 2019b), leading to too little precipitation over India and too much over Eastern China. This leads to a warm temperature bias over India (Figure 2) during the JJA monsoon season. In a similar way, CanESM5.0 precipitation is biased low over the Amazonian region and biased high over the Andes mountains (Figure A5) because higher than observed

advected moisture from the Atlantic Ocean makes its way over the Andes, given the westward surface winds in this region, and the low precipitation over the Amazonian region leads to a warm temperature bias there (Figure 2).

The cold bias over the Himalayan region is not unique to CanESM5.0 and is seen in other ESMs. Lalande et al. (2021) analyze temperature and snow biases in CMIP6 models, including CanESM5.0, over the same region and find that the multi-model mean temperature from 26 models is biased cold by 1.9°C associated with a 12% overestimation of snow cover compared

to a satellite-based data set. Lalande et al. (2021) find that although higher than observed precipitation is the cause of the higher snow amounts and consequently higher surface albedo in most models in the region, other factors including snow fraction parameterization also play a role. Hence the Himalayan cold bias can be regarded as a *community-systemic* issue, although it is relatively large in CanESM5.0 ($\sim 4$–5°C; Figure 20a,b) compared to the Lalande et al. (2021) CMIP6 multi-model mean Himalayan cold bias of ($\sim 2°$).

In summary, it appears that positive precipitation bias over the Himalayan region, which is likely caused by the coarsely-resolved topography of the model, combined together with the lack of a snow fraction parameterization that takes into account the effect of topography, leads to a large accumulation of snow as the model spins up under preindustrial forcing. The higher snow amounts and the resulting increase in surface albedo leads to cold temperature bias over the Himalayan region. The next version of CanESM is expected to have a nominal spatial resolution of 1° for its atmosphere and land components, and it is

expected that biases related to unresolved topography will be reduced in magnitude and spatial extent. Work is also currently underway to evaluate an improved snow fraction parameterization in the CLASSIC land model.

## 6   Summary and Discussion

In this paper we have documented efforts within the CCCma, the group leading the development of the Canadian Earth System Model, to more effectively monitor, document and eventually resolve systematic model biases, through the initiation of the

Analysis for Development (A4D) activity. In addition, we have reported the results of the first phase of this activity, which includes two versions of a new CanESM version (CanESM5.1) featuring several improvements compared to CanESM5.0, and a number of "deep-dive" analyses on specific model biases that have provided insights on how these biases can or cannot be reduced in future CanESM versions.





Three categories of model issues were discussed. An example of the first type, a *version-specific* bias (i.e., an issue specific to
a *single* model version) is the occurrence of spurious stratospheric dust spikes and tropospheric dust storms (Section 5.1), as this
was corrected in the model version immediately following CanESM5.0. The high ECS could be considered a version-specific
issue (not necessarily a bias, as the true ECS is unknown), as a substantially lower ECS was obtained through systematic model
tuning in the "p2" version of CanESM5.1. While the low bias in sudden stratospheric warming frequency was not resolved in
CanESM5.1, this bias is arguably version-specific, since we showed that it can be eliminated through retuning of the parameters
in the OGWD parameterization scheme (Section 5.3.3).

A number of *model-systemic* issues were discussed, which are issues that persist across multiple model versions; they are
"stubbon" errors that have proved insensitive to model development efforts. The degradation of ENSO from CanESM2 /
CanCM4 to CanESM5 (Section 5.2.1) may be model-systemic, as it was only modestly improved by retuning in CanESM5.1-
p2. Similarly, the high North Atlantic sea ice bias (Section 5.3.1) and cold winter bias above sea ice (Section 5.3.2) have been
shown to be relatively insensitive to both the CanESM5.1-p2 retuning and to the various sensitivity experiments used to explore
the possible origins of these biases (Section 5.3.1 and Section 5.3.2). Land surface cold biases (Section 5.3.4) have likewise
persisted across multiple model versions, and so can be considered model-systemic.

*Community-systemic* issues are problems that are common to different models developed by different modelling groups. As
noted in Section 5.2.1, excessive westward extension of SST anomalies and unrealistic eastern equatorial Pacific SST seasonal
cycle are both common errors among CMIP models; hence CanESM's model-systemic ENSO biases are also community-
systemic. Likewise, as Section 5.3.4 noted, Himalayan cold bias is a seen in a number of models. Community-systematic
errors are important because their potential influence on multi-model analyses cannot easily be removed by averaging across
a multi-model ensemble, since many models share these same errors, and hence documenting them is important for informing
applications of these models.

As the primary goal of A4D is to better support model development and provide a more objective path towards model ver-
sions with improved properties and behaviour, an essential component of A4D is an ongoing critical review of its effectiveness
in meeting this goal. It is anticipated that the A4D effort will evolve based on the effectiveness of its initial implementation
and due to the implementation of new technologies applied to model development. For example, CCCma has begun a new
objective tuning approach (Hourdin et al., 2017) that uses history matching methods based on Bayesian statistics to determine
suitable values for CanESM's free physical parameters in finalizing model versions (Williamson et al., 2015, 2017). As the
history matching methodology is designed to identify a space of free parameter values, which by design satisfy sets of observed
metrics within uncertainty, it is expected that its implementation will have a significant impact on the future content and role
of the A4D package of automatic diagnostics.

While development of the next major version of CanESM is still in an early stage, it is expected to include substantial
changes compared to CanESM5.0/5.1, including a new dynamical core, a new version of the ocean model that includes a new
sea ice component, and a new ecosystem and land surface model. Such major changes have the potential to drastically change
the model's properties, and ensuring that these properties compare well to observations will present unique challenges. By



increasing focus on analysis that is specifically tailored to support model development we have implemented structures and tools that will help us face these challenges in an effective manner.

*Code and data availability.* The full CanESM5.0 and CanESM5.1 source codes are publicly available at https://gitlab.com/cccma/canesm. The version of the code that was used to produce all the CanESM5.0 simulations submitted to CMIP6, and are described in this paper, is tagged as v5.0.3 and has the associated DOI: https://doi.org/10.5281/zenodo.3251114 (Swart et al., 2019a). The version of the CanESM5.1 code that was used to produce all the simulations described in this paper is tagged as v5.1.6 and has the associated DOI:https://zenodo.org/record/7786802 (Swart et al., 2023). All standard (CMIP-style) CanESM5.0 and CanESM5.1 simulations described in this paper are publicly
available via the Earth System Grid Federation (ESGF). The full CLASSIC code can be downloaded from https://gitlab.com/cccma/classic. The version of CLASSIC code used in this paper is v1.0, and has the associated DOI:https://doi.org/10.5281/zenodo.3522407 (Melton et al., 2019). The input data of the CLASSIC simulations presented here can be downloaded from https://data.isimip.org (GSWP3) and https://crudata.uea.ac.uk/cru/data/hrg/ (CRU-JRA). Data from all bespoke CanESM5.0, CanESM5.1 and CLASSIC experiments described in Section 5.3 are available at https://crd-data-donnees-rdc.ec.gc.ca/ CCCMA/publications/A4D.

**Appendix A: Supplemental Figures**

**Appendix B: Remotely Sensed Observations of Aerosol Optical Depth**

We here describe the remotely sensed observations of aerosol optical depth used for model evaluation in Section 5.1, and provide an overview of their strengths and limitations.

We use monthly mean total AOD retrieved from the Moderate Resolution Imaging Spectroradiometer (MODIS; King et al., 725 2013), Multi-angle Spectral Radiometer (MISR; Diner et al., 1998), and Cloud-Aerosol Lidar with Orthogonal Polarization (CALIOP; Winker et al., 2009) instruments. MODIS and MISR are passive instruments, measuring only during the day, and CALIOP is a lidar sensor which measures during both day and night. There are two MODIS instruments, carried on the Aqua and Terra satellites, with equatorial overpass times of 10:30am and 1:30pm respectively; we consider the products from these two sensors separately. MISR and CALIOP are single instruments. MISR is carried onboard Terra, with one of the MODIS 730 instruments. CALIOP is onboard the Cloud-Aerosol Lidar and Infrared Pathfinder Satellite Observations (CALIPSO) satellite, which lagged Aqua by 1-2 minutes during the time period analyzed here; it has since been relocated to a lower-altitude orbit.

Generally, AOD retrieved from MODIS is higher than that from other satellites, while AOD from MISR and CALIPSO is lower than other satellites, when comparing with other datasets that are available for shorter periods of time (Vogel et al., 2022). The spread between MODIS and MISR/CALIPSO can thus be taken as an approximate estimate of the overall uncertainty in 735 observed AOD. For CALIPSO, we include results for both daytime, cloud-free conditions ("CALIPSO CloudFreeDay") and day-night averaged, all-sky conditions ("CALIPSO AllSkyDayNight").

The retrieval of dust optical depth is more challenging than total optical depth, because it requires additional information on particle size and/or shape in order to distinguish dust optical depth from the contributions of other aerosol types. MODIS and



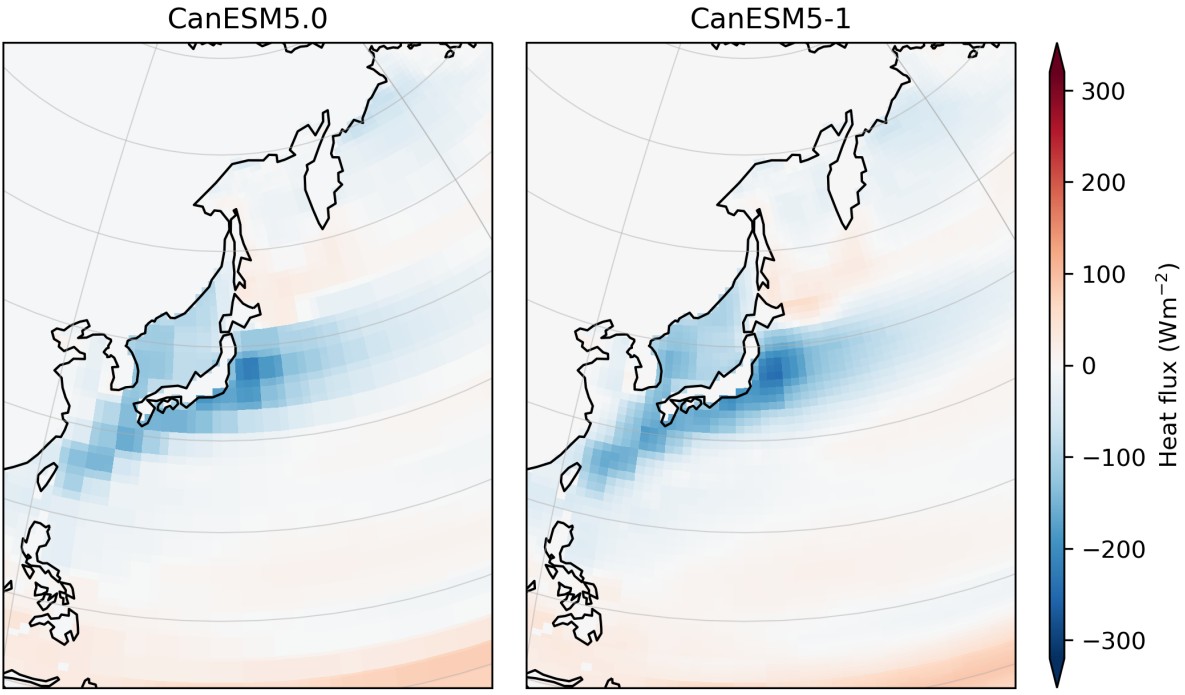

**Figure A1.** Net surface downward heat flux in CanESM5.0 (left) and CanESM5.1 (right) on the native ocean grid, over the vicinity of the Kuroshio Current. The conservative2 remapping method used in CanESM5.1 leads to a somewhat smoother field than seen in CanESM5.0 (conservative). The blockiness in the fields arises due to the much coarse resolution of the atmospheric grid ($3°$), where the fluxes are computed, relative to the ocean grid ($1°$).

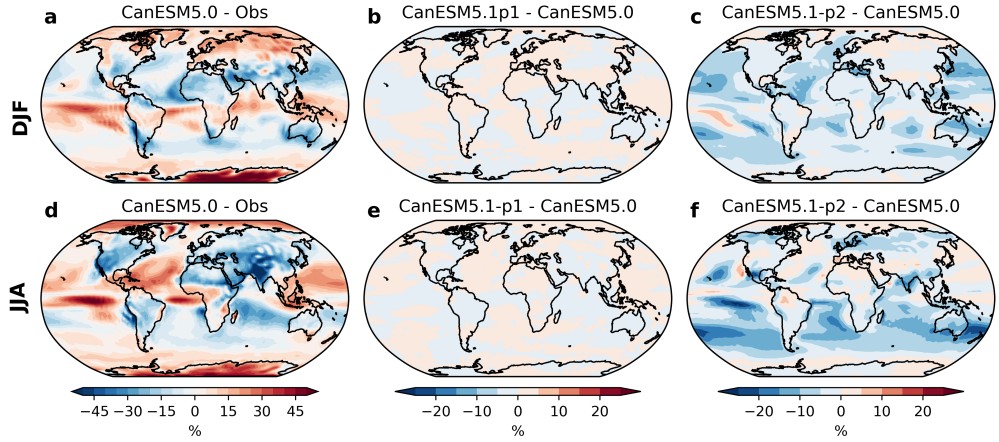

**Figure A2.** As Figure 2, except for cloud fraction and the bias relative to International Satellite Cloud Climatology Project H series (ISCCP-H) satellite-based observations (Young et al., 2018) averaged over 1991–2010

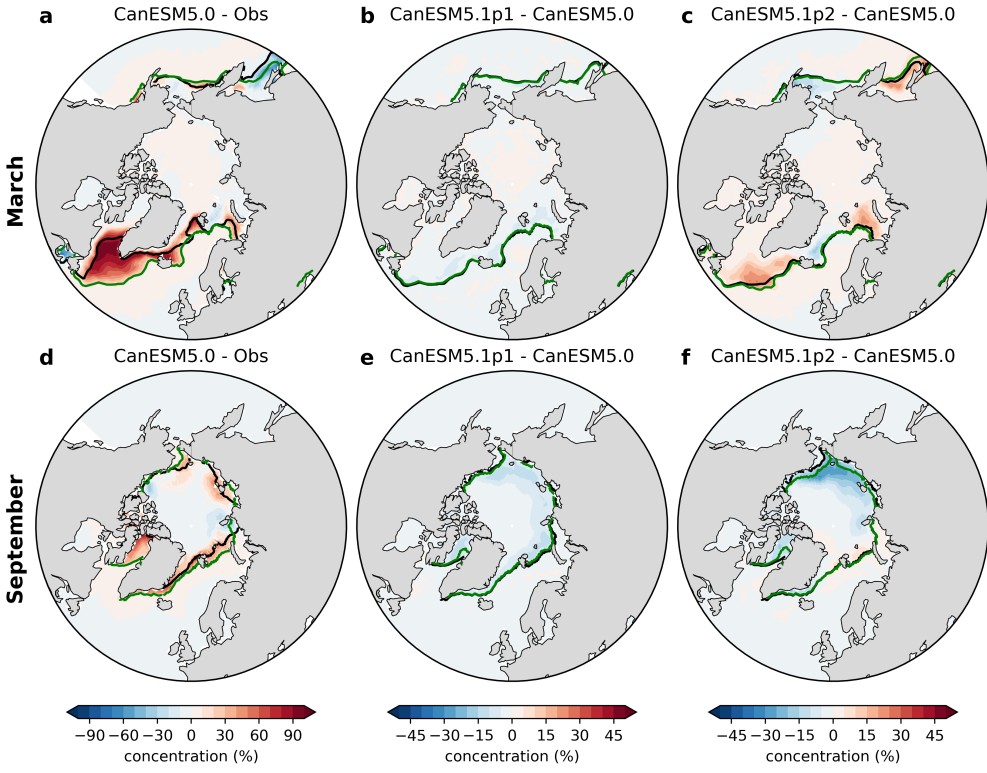

**Figure A3.** As Figure 2, except for Arctic sea ice concentration in March (first row) and September (second row), and sea ice concentration from NSIDC as the observations. The ice edge (15% sea ice concentration contour) is indicated by black for observations in (a,d) and CanESM5.0 in (b,c,e,f) and green for CanESM5.0 in (a,d), CanESM5.1-p1 in (b,e) and CanESM5.1-p2 in (c,f).

MISR do not provide dust products. CALIOP optical depths are published for *dust* and *polluted dust* categories, but these are

separate classifications (the former refers to clean dust only, and the latter refers to dust mixed with urban pollution or biomass burning smoke) and there is no estimate of the total dust optical depth (Omar et al., 2009). Here we use the *dust* classification but emphasize that it is an underestimate of the total contribution from mineral dust. We augment these observations with dust optical depths from Song et al. (2021) (ACROS-MODIS and ACROS-CALIPSO, derived from MODIS and CALIPSO data respectively), Gkikas et al. (2020) (MIDAS, derived from MODIS data), and Voss and Evan (2019) (Voss+2019, derived from

MODIS data).

*Author contributions.* MS co-leads the A4D activity, organized the structure of, coordinated the writing of and edited the manuscript, led the analysis in section 4 and 5.3.3, performed the perturbed gravity wave model simulations for section 5.3.3, led the analysis of the stratospheric warming spikes in section 5.1, and contributed to the writing of several other sections; JA co-leads the A4D activity, coordinated the writing of and edited the manuscript, contributed to the writing of several sections, edited the manuscript, created Figure 1, organized publication

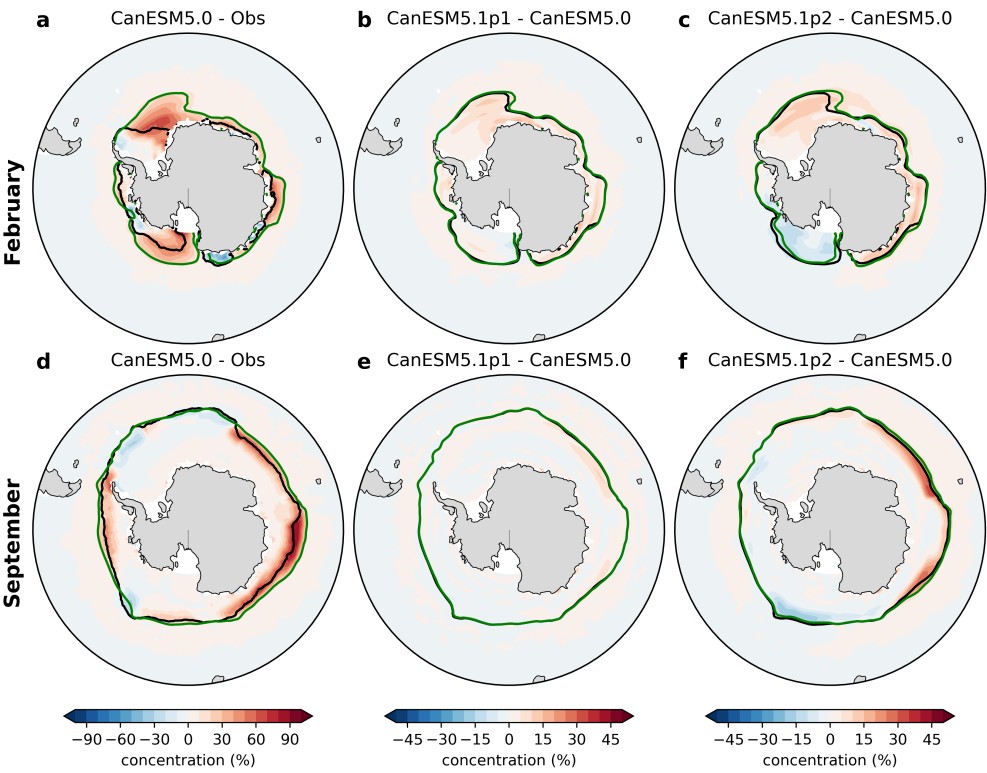

**Figure A4.** As Figure A4, except for Antarctic sea ice in February (first row) and September (second row)

of the datasets used, and contributed to development of the standardized diagnostic software; VA led section 5.3.4; RD led the tropospheric dust analysis in section 5.1 and performed the AMIP simulations for section 5.1 ; NG contributed to sections 5.1 and 5.2.2 and edited the manuscript; VK retuned the dust hybridization parameters in CanESM5.1, performed the systematic tuning of the p2 variant of CanESM5.1, and performed the wind-nudged simulations for section 5.3.1; WM led section 5.2.1; CR leads the development of the standardized diagnostic software, produced the CanESM analysis reports and contributed to section 5.3.1; JoS led section 2; NS leads CanESM development and

led sections 5.3.1 and 5.3.2; JV led the feedback analysis in section 5.2.2; CA contributed to section 3.2; JaC contributed to section 5.2.2; NL contributed to section 5.3.1; WS-L created Figure 11 and provided data for Figure 9; YL created Figure 13; EM created Figure 4; LR contributed to the initial analysis in section 5.1; KS contributed to section 5.1; CrS performed simulations and made figures for section 5.3.4; ClS performed most of the CanESM5.1 simulations; AS contributed to the initial analysis in section 5.3.1; RS-A created Figure 10; LW contributed to the analysis in section 5.3.4; and DY performed simulations for section 5.3.1.

*Competing interests.* The authors declare no competing interests





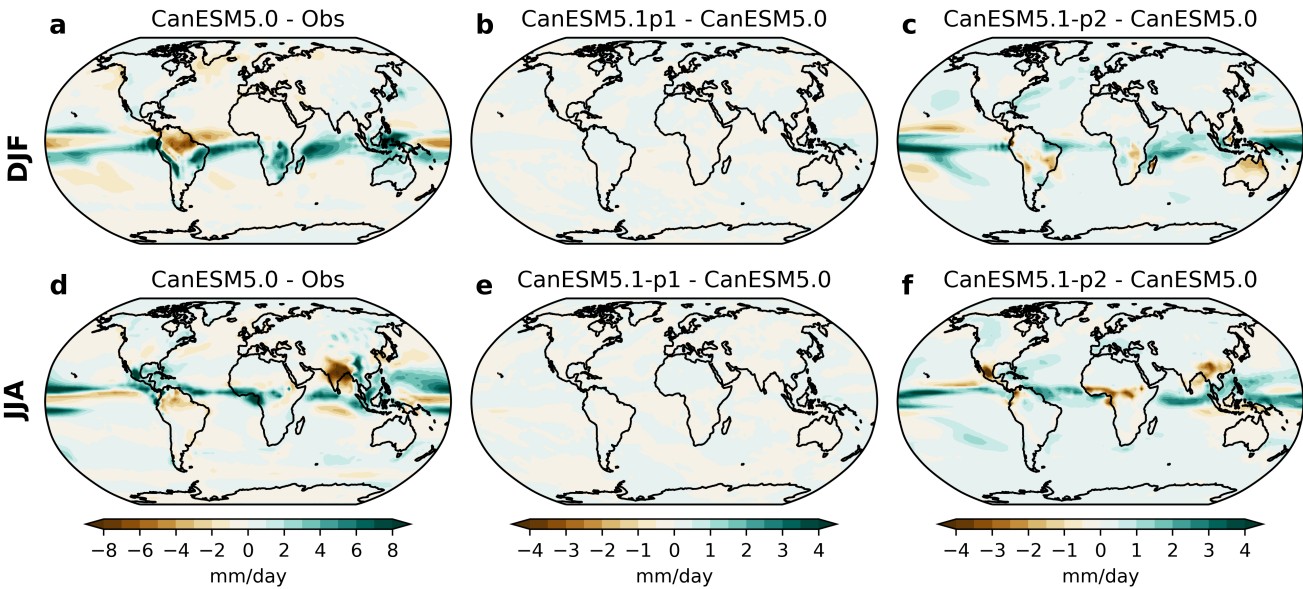

**Figure A5.** As Figure 2, except for precipitation using GPCP as the observations.

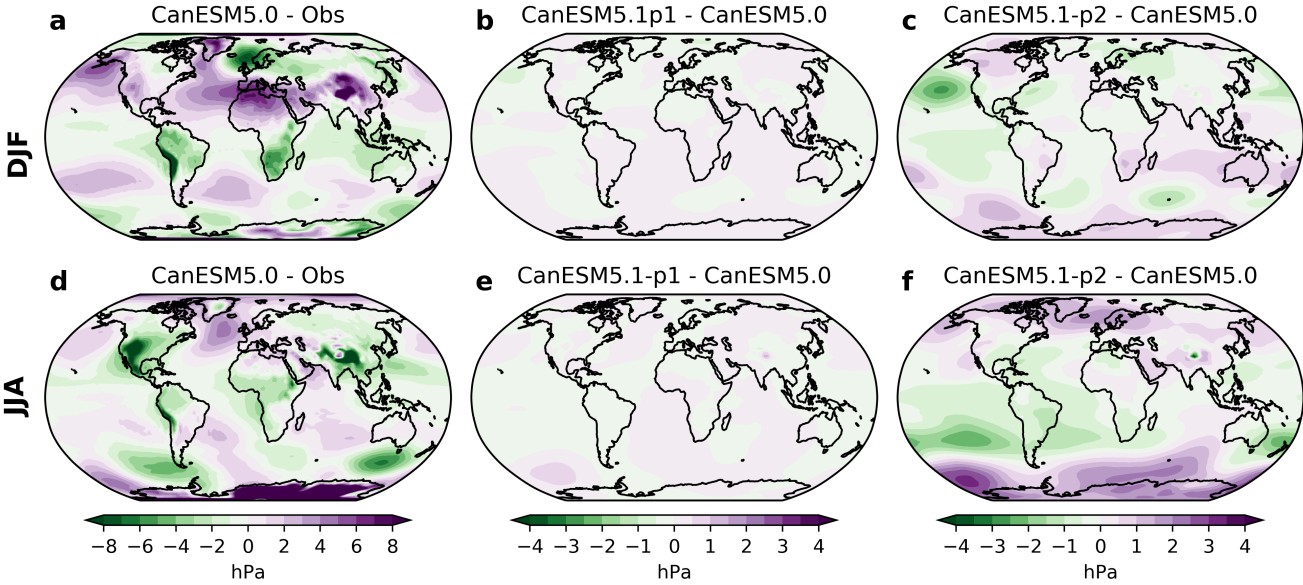

**Figure A6.** As Figure 2, except for sea level pressure.

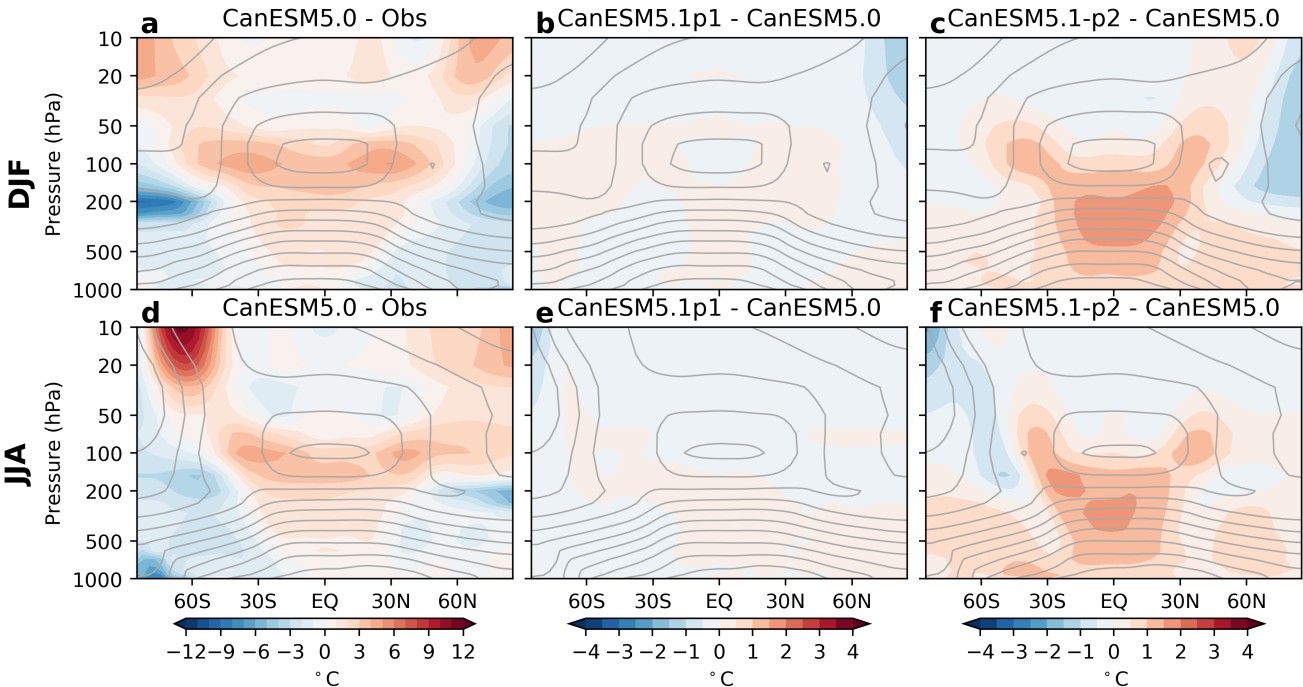

**Figure A7.** As Figure 2, except for zonal mean temperature, with the contours representing the climatological zonal mean temperatures in CanESM5.0 (a,d), CanESM5.1-p1 (b,e) and CanESM5.1-p2 (c,f) (contour interval: 10 °C)

*Acknowledgements.* We thank Yanjun Jiao for technical assistance.



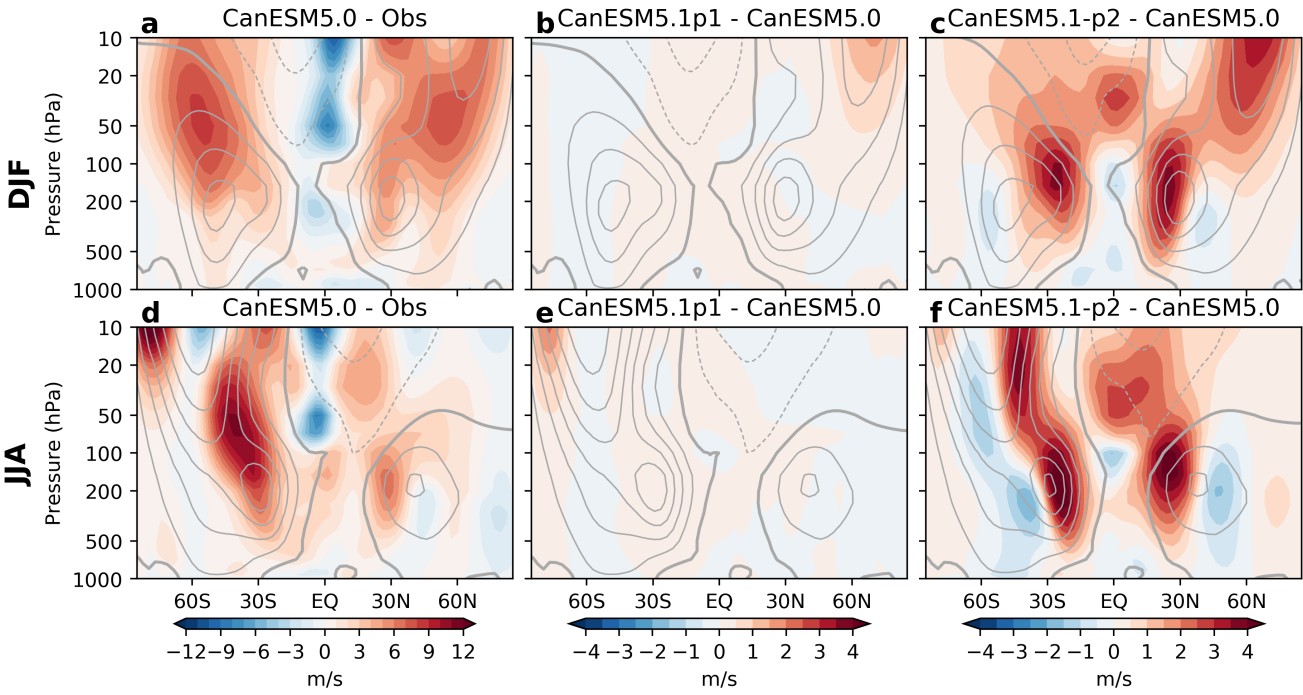

**Figure A8.** As Figure 2, except for zonal mean zonal wind. The contours represent the climatological zonal mean temperatures in CanESM5.0 (a,d), CanESM5.1-p1 (b,e) and CanESM5.1-p2 (c,f). The contour interval is 10 m/s with the thick solid line denoting the 0 m/s contour.

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



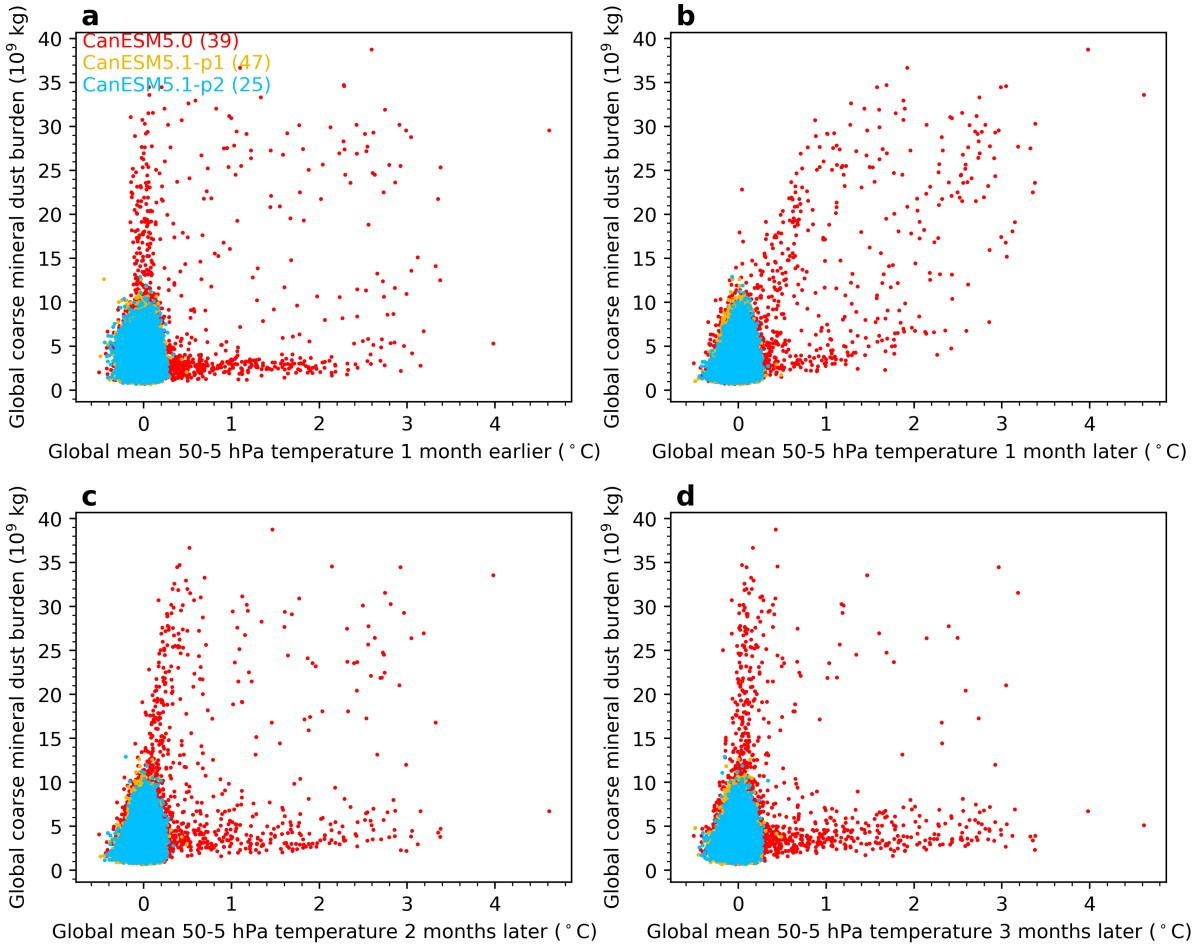

**Figure A9.** Total global coarse mineral dust burden versus unforced global mean lower stratospheric temperature in a) the previous month, b) the next month, c) the second month and d) the third month, for all months and all ensemble members

Canadell, J., Monteiro, P., Costa, M., Cotrim da Cunha, L., Cox, P., Eliseev, A., Henson, S., Ishii, M., Jaccard, S., Koven, C., Lohila, A., Patra, P., Piao, S., Rogelj, J., Syampungani, S., Zaehle, S., and Zickfeld, K.: Global Carbon and other Biogeochemical Cycles and Feedbacks, p. 673–816, Cambridge University Press, Cambridge, United Kingdom and New York, NY, USA, https://doi.org/10.1017/9781009157896.007, 2021.

Charlton-Perez, A. J., Baldwin, M. P., Birner, T., Black, R. X., Butler, A. H., Calvo, N., Davis, N. a., Gerber, E. P., Gillett, N., Hardiman, S., Kim, J., Krüger, K., Lee, Y.-Y., Manzini, E., McDaniel, B. a., Polvani, L., Reichler, T., Shaw, T. a., Sigmond, M., Son, S.-W., Toohey, M., Wilcox, L., Yoden, S., Christiansen, B., Lott, F., Shindell, D., Yukimoto, S., and Watanabe, S.: On the lack of stratospheric dynamical variability in low-top versions of the CMIP5 models, Journal of Geophysical Research: Atmospheres, 118, 2494–2505, https://doi.org/10.1002/jgrd.50125, 2013.



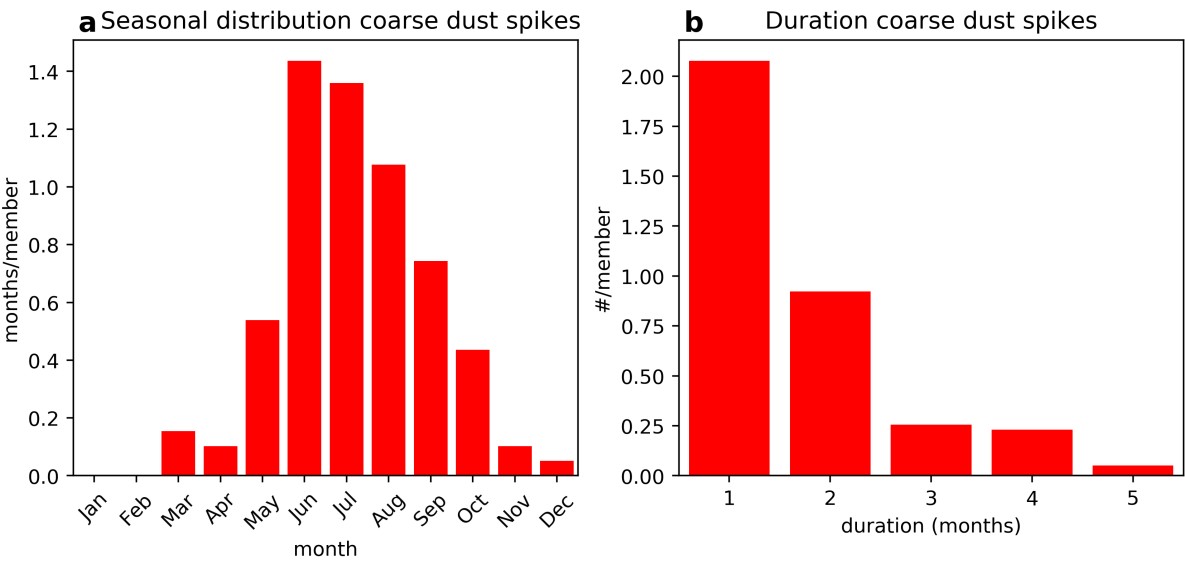

**Figure A10.** a) Seasonal distribution and b) histogram of the duration of the coarse mineral dust spikes with global burden that exceeds $12 \times 10^9 kg$ in CanESM5.0

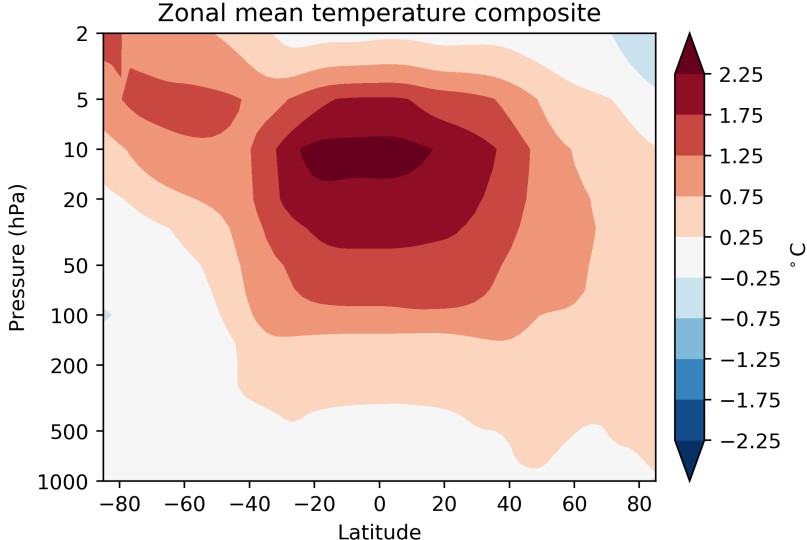

**Figure A11.** Zonal mean temperature anomaly composite of all months in with coarse mineral dust burden exceeding $12 \times 10^9 kg$ in CanESM5.0.



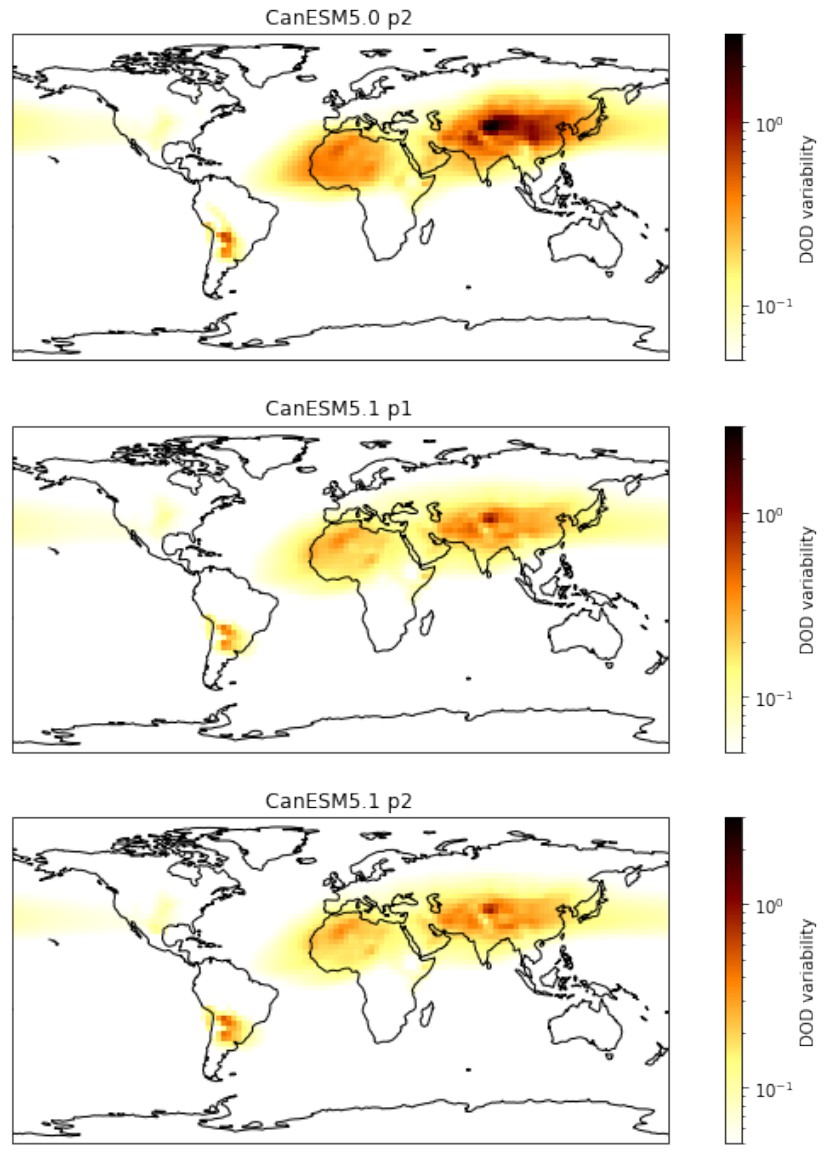

**Figure A12.** Typical variability in dust optical depth in CanESM versions 5.0p2, 5.1p1, and 5.1p2. For each ensemble member, variability is calculated as the standard deviation in dust optical depth over 2007-2014; maps show the ensemble mean of this variability. Note the logarithmic colour scale, and the reduction in variability from CanESM5.0 to CanESM5.1.

**Figure A13.** Surface air temperature anomalies, relative to the control experiment using default CanAM5 values, in the DJF (left column) and JJA (right column) seasons, for the experiments with altered snow conductivity (cons) and snow density (dens) listed in Table 2: experiment 4 (a,b), 3 (c,d), 2 (e,f), and 1 (g,h). Experiments 5 and 6 that alter the ice conductivity are not shown here, but show even smaller anomalies than the other experiments.





Chung, E.-S. and Soden, B. J.: An assessment of direct radiative forcing, radiative adjustments, and radiative feedbacks in coupled ocean–atmosphere models, Journal of Climate, 28, 4152–4170, 2015.

Compo, G. P., Whitaker, J. S., Sardeshmukh, P. D., Matsui, N., Allan, R. J., Yin, X., Gleason, B. E., Vose, R. S., Rutledge, G., Bessemoulin,
P., Brönnimann, S., Brunet, M., Crouthamel, R. I., Grant, A. N., Groisman, P. Y., Jones, P. D., Kruk, M. C., Kruger, A. C., Marshall, G. J.,
        Maugeri, M., Mok, H. Y., Nordli, o., Ross, T. F., Trigo, R. M., Wang, X. L., Woodruff, S. D., and Worley, S. J.: The Twentieth Century
        Reanalysis Project, Quarterly Journal of the Royal Meteorological Society, 137, 1–28, https://doi.org/10.1002/qj.776, 2011.

Diner, D., Beckert, J., Reilly, T., Bruegge, C., Conel, J., Kahn, R., Martonchik, J., Ackerman, T., Davies, R., Gerstl, S., Gordon, H., Muller,
        J.-P., Myneni, R., Sellers, P., Pinty, B., and Verstraete, M.: Multi-Angle Imaging SpectroRadiometer (MISR) Instrument Description and
Experiment Overview, IEEE Trans. Geosci. Remote Sens., 36, https://doi.org/10.1109/36.700992, 1998.

Domeisen, D. I., Butler, A. H., Charlton-Perez, A. J., Ayarzagüena, B., Baldwin, M. P., Dunn-Sigouin, E., Furtado, J. C., Garfinkel, C. I.,
        Hitchcock, P., Karpechko, A. Y., Kim, H., Knight, J., Lang, A. L., Lim, E. P., Marshall, A., Roff, G., Schwartz, C., Simpson, I. R., Son,
        S. W., and Taguchi, M.: The Role of the Stratosphere in Subseasonal to Seasonal Prediction: 2. Predictability Arising From Stratosphere-
        Troposphere Coupling, Journal of Geophysical Research: Atmospheres, 125, 1–20, https://doi.org/10.1029/2019JD030923, 2020.

Dong, Y., Armour, K. C., Zelinka, M. D., Proistosescu, C., Battisti, D. S., Zhou, C., and Andrews, T.: Intermodel spread in the pattern effect
        and its contribution to climate sensitivity in CMIP5 and CMIP6 models, Journal of Climate, 33, 7755–7775, 2020.

Eyring, V., Righi, M., Lauer, A., Evaldsson, M., Wenzel, S., Jones, C., Anav, A., Andrews, O., Cionni, I., Davin, E. L., Deser, C., Ehbrecht, C.,
        Friedlingstein, P., Gleckler, P., Gottschaldt, K.-D., Hagemann, S., Juckes, M., Kindermann, S., Krasting, J., Kunert, D., Levine, R., Loew,
        A., Mäkelä, J., Martin, G., Mason, E., Phillips, A. S., Read, S., Rio, C., Roehrig, R., Senftleben, D., Sterl, A., van Ulft, L. H., Walton,
J., Wang, S., and Williams, K. D.: ESMValTool (v1.0) – a community diagnostic and performance metrics tool for routine evaluation of
        Earth system models in CMIP, Geoscientific Model Development, 9, 1747–1802, https://doi.org/10.5194/gmd-9-1747-2016, publisher:
        Copernicus GmbH, 2016.

Eyring, V., Bock, L., Lauer, A., Righi, M., Schlund, M., Andela, B., Arnone, E., Bellprat, O., Brötz, B., Caron, L.-P., Carvalhais, N., Cionni,
        I., Cortesi, N., Crezee, B., Davin, E. L., Davini, P., Debeire, K., de Mora, L., Deser, C., Docquier, D., Earnshaw, P., Ehbrecht, C., Gier,
B. K., Gonzalez-Reviriego, N., Goodman, P., Hagemann, S., Hardiman, S., Hassler, B., Hunter, A., Kadow, C., Kindermann, S., Koirala,
        S., Koldunov, N., Lejeune, Q., Lembo, V., Lovato, T., Lucarini, V., Massonnet, F., Müller, B., Pandde, A., Pérez-Zanón, N., Phillips, A.,
        Predoi, V., Russell, J., Sellar, A., Serva, F., Stacke, T., Swaminathan, R., Torralba, V., Vegas-Regidor, J., von Hardenberg, J., Weigel,
        K., and Zimmermann, K.: Earth System Model Evaluation Tool (ESMValTool) v2.0 – an extended set of large-scale diagnostics for
        quasi-operational and comprehensive evaluation of Earth system models in CMIP, Geoscientific Model Development, 13, 3383–3438,
https://doi.org/10.5194/gmd-13-3383-2020, publisher: Copernicus GmbH, 2020.

Eyring, V., Gillett, N., Rao, K. A., Barimalala, R., Parrillo, M. B., Bellouin, N., Cassou, C., Durack, P., Kosaka, Y., McGregor, S., Min,
        S., Morgenstern, O., and Sun, Y.: Human Influence on the Climate System, in: Climate Change 2021: The Physical Science Basis.
        Contribution of Working Group I to the Sixth Assessment Report of the Intergovernmental Panel on Climate Change, edited by Masson-
        Delmotte, V., Zhai, P., Pirani, A., Connors, S., Péan, C., Berger, S., Caud, N., Chen, Y., Goldfarb, L., Gomis, M., Huang, M., Leitzell,
K., Lonnoy, E., Matthews, J., Maycock, T., Waterfield, T., Yelekçi, O., Yu, R., and Zhou, B., chap. 3, p. 423–552, Cambridge University
        Press, Cambridge, United Kingdom and New York, NY, USA, https://doi.org/10.1017/9781009157896.005, 2021.

Friedlingstein, P., Jones, M. W., O'Sullivan, M., Andrew, R. M., Bakker, D. C. E., Hauck, J., Le Quéré, C., Peters, G. P., Peters, W., Pongratz,
        J., Sitch, S., Canadell, J. G., Ciais, P., Jackson, R. B., Alin, S. R., Anthoni, P., Bates, N. R., Becker, M., Bellouin, N., Bopp, L., Chau, T.
        T. T., Chevallier, F., Chini, L. P., Cronin, M., Currie, K. I., Decharme, B., Djeutchouang, L. M., Dou, X., Evans, W., Feely, R. A., Feng,





L., Gasser, T., Gilfillan, D., Gkritzalis, T., Grassi, G., Gregor, L., Gruber, N., Gürses, O., Harris, I., Houghton, R. A., Hurtt, G. C., Iida, Y., Ilyina, T., Luijkx, I. T., Jain, A., Jones, S. D., Kato, E., Kennedy, D., Klein Goldewijk, K., Knauer, J., Korsbakken, J. I., Körtzinger, A., Landschützer, P., Lauvset, S. K., Lefèvre, N., Lienert, S., Liu, J., Marland, G., McGuire, P. C., Melton, J. R., Munro, D. R., Nabel, J. E. M. S., Nakaoka, S.-I., Niwa, Y., Ono, T., Pierrot, D., Poulter, B., Rehder, G., Resplandy, L., Robertson, E., Rödenbeck, C., Rosan, T. M., Schwinger, J., Schwingshackl, C., Séférian, R., Sutton, A. J., Sweeney, C., Tanhua, T., Tans, P. P., Tian, H., Tilbrook, B., Tubiello,

F., van der Werf, G. R., Vuichard, N., Wada, C., Wanninkhof, R., Watson, A. J., Willis, D., Wiltshire, A. J., Yuan, W., Yue, C., Yue, X., Zaehle, S., and Zeng, J.: Global Carbon Budget 2021, Earth System Science Data, 14, 1917–2005, https://doi.org/10.5194/essd-14-1917-2022, 2022.

Gastineau, G., Lott, F., Mignot, J., and Hourdin, F.: Alleviation of an Arctic Sea Ice Bias in a Coupled Model Through Modifications in the Subgrid-Scale Orographic Parameterization, Journal of Advances in Modeling Earth Systems, 12, e2020MS002111,
https://doi.org/10.1029/2020MS002111, _eprint: https://onlinelibrary.wiley.com/doi/pdf/10.1029/2020MS002111, 2020.

Gkikas, A., Proestakis, E., Amiridis, V., Kazadzis, S., Tomaso, E. D., Tsekeri, A., Marinou, E., Hatzianastassiou, N., and García-Pando, C. P.: ModIs Dust AeroSol (MIDAS): A global fine resolution dust optical depth dataset, https://doi.org/10.5281/zenodo.4244106, 2020.

Gleckler, P. J., Doutriaux, C., Durack, P. J., Taylor, K. E., Zhang, Y., Williams, D. N., Mason, E., and Servonnat, J.: A more powerful reality test for climate models, Eos, 97, 20–24, 2016.

Gregory, J., Ingram, W., Palmer, M., Jones, G., Stott, P., Thorpe, R., Lowe, J., Johns, T., and Williams, K.: A new method for diagnosing radiative forcing and climate sensitivity, Geophysical research letters, 31, 2004.

Griffies, S. M., Danabasoglu, G., Durack, P. J., Adcroft, A. J., Balaji, V., Böning, C. W., Chassignet, E. P., Curchitser, E., Deshayes, J., Drange, H., Fox-Kemper, B., Gleckler, P. J., Gregory, J. M., Haak, H., Hallberg, R. W., Heimbach, P., Hewitt, H. T., Holland, D. M., Ilyina, T., Jungclaus, J. H., Komuro, Y., Krasting, J. P., Large, W. G., Marsland, S. J., Masina, S., McDougall, T. J., Nurser, A. J. G., Orr,
J. C., Pirani, A., Qiao, F., Stouffer, R. J., Taylor, K. E., Treguier, A. M., Tsujino, H., Uotila, P., Valdivieso, M., Wang, Q., Winton, M., and Yeager, S. G.: OMIP contribution to CMIP6: experimental and diagnostic protocol for the physical component of the Ocean Model Intercomparison Project, Geoscientific Model Development, 9, 3231–3296, https://doi.org/10.5194/gmd-9-3231-2016, 2016.

Guilyardi, E., Bellenger, H., Collins, M., Ferrett, S., Cai, W., and Wittenberg, A.: A first look at ENSO in CMIP5, Clivar Exchanges, 117, 29–32, 2012.

Ham, Y.-G., Kim, J.-H., and Luo, J.-J.: Deep learning for multi-year ENSO forecasts, Nature, 573, 568–572, https://doi.org/10.1038/s41586-019-1559-7, 2019.

Hersbach, H., Bell, B., Berrisford, P., Hirahara, S., Horányi, A., Muñoz-Sabater, J., Nicolas, J., Peubey, C., Radu, R., Schepers, D., Simmons, A., Soci, C., Abdalla, S., Abellan, X., Balsamo, G., Bechtold, P., Biavati, G., Bidlot, J., Bonavita, M., De Chiara, G., Dahlgren, P., Dee, D., Diamantakis, M., Dragani, R., Flemming, J., Forbes, R., Fuentes, M., Geer, A., Haimberger, L., Healy, S., Hogan, R. J.,
Hólm, E., Janisková, M., Keeley, S., Laloyaux, P., Lopez, P., Lupu, C., Radnoti, G., de Rosnay, P., Rozum, I., Vamborg, F., Villaume, S., and Thépaut, J.-N.: The ERA5 global reanalysis, Quarterly Journal of the Royal Meteorological Society, 146, 1999–2049, https://doi.org/10.1002/qj.3803, _eprint: https://onlinelibrary.wiley.com/doi/pdf/10.1002/qj.3803, 2020.

Hitchcock, P., Butler, A., Charlton-Perez, A., Garfinkel, C. I., Stockdale, T., Anstey, J., Mitchell, D., Domeisen, D. I. V., Wu, T., Lu, Y., Mastrangelo, D., Malguzzi, P., Lin, H., Muncaster, R., Merryfield, B., Sigmond, M., Xiang, B., Jia, L., Hyun, Y.-K., Oh, J., Specq, D.,
Simpson, I. R., Richter, J. H., Barton, C., Knight, J., Lim, E.-P., and Hendon, H.: Stratospheric Nudging And Predictable Surface Impacts (SNAPSI): a protocol for investigating the role of stratospheric polar vortex disturbances in subseasonal to seasonal forecasts, Geoscientific Model Development, 15, 5073–5092, https://doi.org/10.5194/gmd-15-5073-2022, 2022.





Hourdin, F., Mauritsen, T., Gettelman, A., Golaz, J.-C., Balaji, V., Duan, Q., Folini, D., Ji, D., Klocke, D., Qian, Y., Rauser, F., Rio, C., Tomassini, L., Watanabe, M., and Williamson, D.: The Art and Science of Climate Model Tuning, Bulletin of the American Meteorological
Society, 98, 589–602, https://doi.org/10.1175/BAMS-D-15-00135.1, publisher: American Meteorological Society Section: Bulletin of the American Meteorological Society, 2017.

Huang, B., Thorne, P. W., Banzon, V. F., Boyer, T., Chepurin, G., Lawrimore, J. H., Menne, M. J., Smith, T. M., Vose, R. S., and Zhang, H.-M.: Extended Reconstructed Sea Surface Temperature, Version 5 (ERSSTv5): Upgrades, Validations, and Intercomparisons, Journal of Climate, 30, 8179–8205, https://doi.org/10.1175/JCLI-D-16-0836.1, publisher: American Meteorological Society Section: Journal of
Climate, 2017.

Jiang, W., Huang, P., Huang, G., and Ying, J.: Origins of the excessive westward extension of ENSO SST simulated in CMIP5 and CMIP6 models, Journal of Climate, 34, 2839–2851, https://doi.org/10.1175/JCLI-D-20-0551.1, 2021.

Jones, C. D., Hickman, J. E., Rumbold, S. T., Walton, J., Lamboll, R. D., Skeie, R. B., Fiedler, S., Forster, P. M., Rogelj, J., Abe, M., Botzet, M., Calvin, K., Cassou, C., Cole, J. N. S., Davini, P., Deushi, M., Dix, M., Fyfe, J. C., Gillett, N. P., Ilyina, T., Kawamiya, M., Kelley,
M., Kharin, S., Koshiro, T., Li, H., Mackallah, C., Müller, W. A., Nabat, P., van Noije, T., Nolan, P., Ohgaito, R., Olivié, D., Oshima, N., Parodi, J., Reerink, T. J., Ren, L., Romanou, A., Séférian, R., Tang, Y., Timmreck, C., Tjiputra, J., Tourigny, E., Tsigaridis, K., Wang, H., Wu, M., Wyser, K., Yang, S., Yang, Y., and Ziehn, T.: The Climate Response to Emissions Reductions Due to COVID-19: Initial Results From CovidMIP, Geophys. Res. Lett., 48, https://doi.org/10.1029/2020GL091883, 2021.

Karpechko, A. Y., Afargan-Gerstman, H., Butler, A. H., Domeisen, D. I. V., Kretschmer, M., Lawrence, Z., Manzini, E., Sigmond, M.,
Simpson, I. R., and Wu, Z.: Northern Hemisphere Stratosphere-Troposphere Circulation Change in CMIP6 Models: 1. Inter-Model Spread and Scenario Sensitivity, Journal of Geophysical Research: Atmospheres, 127, e2022JD036 992, https://doi.org/10.1029/2022JD036992, _eprint: https://onlinelibrary.wiley.com/doi/pdf/10.1029/2022JD036992, 2022.

Kato, S., Loeb, N. G., Rose, F. G., Doelling, D. R., Rutan, D. A., Caldwell, T. E., Yu, L., and Weller, R. A.: Surface Irradiances Consistent with CERES-Derived Top-of-Atmosphere Shortwave and Longwave Irradiances, J. Clim., 26, 2719–2740, 2013.

Kim, J., Son, S.-W., Gerber, E. P., and Park, H.-S.: Defining Sudden Stratospheric Warming in Climate Models: Accounting for Biases in Model Climatologies, Journal of Climate, 30, 5529–5546, https://doi.org/10.1175/JCLI-D-16-0465.1, publisher: American Meteorological Society Section: Journal of Climate, 2017.

King, M. D., Platnick, S., Menzel, W. P., Ackerman, S. A., and Hubanks, P. A.: Spatial and temporal distribution of clouds observed by MODIS onboard the Terra and Aqua satellites, IEEE Trans. Geosci. Remote Sens., 51, https://doi.org/10.1109/TGRS.2012.2227333,
890 2013.

Lalande, M., Ménégoz, M., Krinner, G., Naegeli, K., and Wunderle, S.: Climate change in the High Mountain Asia in CMIP6, Earth System Dynamics, 12, 1061–1098, https://doi.org/10.5194/esd-12-1061-2021, 2021.

Lee, J.-Y., Marotzke, J., Bala, G., Cao, L., Corti, S., Dunne, J., Engelbrecht, F., Fischer, E., Fyfe, J., Jones, C., Maycock, A., Mutemi, J., Ndiaye, O., Panickal, S., and Zhou, T.: Future Global Climate: Scenario-Based Projections and Near-Term Information, p. 553–672,
Cambridge University Press, Cambridge, United Kingdom and New York, NY, USA, https://doi.org/10.1017/9781009157896.006, 2021.

Liang, Y., Gillett, N. P., and Monahan, A. H.: Emergent Constraints on CMIP6 Climate Warming Projections: Contrasting Cloud-and Surface Temperature–Based Constraints, Journal of Climate, 35, 1809–1824, 2022.

Liao, H., Wang, C., and Song, Z.: ENSO phase-locking biases from the CMIP5 to CMIP6 models and a possible explanation, Deep Sea Research Part II: Topical Studies in Oceanography, 189-190, 104 943, https://doi.org/10.1016/J.DSR2.2021.104943, 2021.





Lin, H., Merryfield, W. J., Muncaster, R., Smith, G. C., Markovic, M., Dupont, F., Roy, F., Lemieux, J.-F., Dirkson, A., Kharin, V. V., Lee, W.-S., Charron, M., and Erfani, A.: The Canadian Seasonal to Interannual Prediction System Version 2 (CanSIPSv2), Weather and Forecasting, 35, 1317–1343, https://doi.org/10.1175/WAF-D-19-0259.1, 2020.

Liu, M., Ren, H.-L., Zhang, R., Ineson, S., and Wang, R.: ENSO phase-locking behavior in climate models: From CMIP5 to CMIP6, Environmental Research Communications, 3, 31 004, https://doi.org/10.1088/2515-7620/abf295, 2021.

Lott, F.: Alleviation of Stationary Biases in a GCM through a Mountain Drag Parameterization Scheme and a Simple Representation of Mountain Lift Forces, Monthly Weather Review, 127, 788–801, https://doi.org/10.1175/1520-0493(1999)127<0788:AOSBIA>2.0.CO;2, publisher: American Meteorological Society Section: Monthly Weather Review, 1999.

Meier, W. N., Fetterer, F., Windnagel, and Stewart, J. S.: NOAA/NSIDC Climate Data Record of Passive Microwave Sea Ice Concentration, Version 4, https://doi.org/10.7265/efmz-2t65, 2021.

Melton, J. R., Arora, V., Wisernig-Cojoc, E., Seiler, C., Fortier, M., Chan, E., and Teckentrup, L.: The Canadian Land Surface Scheme including Biogeochemical Cycles, https://doi.org/10.5281/zenodo.3522407, language: eng, 2019.

Melton, J. R., Arora, V. K., Wisernig-Cojoc, E., Seiler, C., Fortier, M., Chan, E., and Teckentrup, L.: CLASSIC v1.0: the open-source community successor to the Canadian Land Surface Scheme (CLASS) and the Canadian Terrestrial Ecosystem Model (CTEM) – Part 1: Model framework and site-level performance, Geoscientific Model Development, 13, 2825–2850, https://doi.org/10.5194/gmd-13-2825-915 2020, 2020.

Merryfield, W. J., Lee, W.-S., Boer, G. J., Kharin, V. V., Scinocca, J. F., Flato, G. M., Ajayamohan, R. S., Fyfe, J. C., Tang, Y., and Polavarapu, S.: The Canadian Seasonal to Interannual Prediction System. Part I: Models and Initialization, Monthly Weather Review, 141, 2910–2945, https://doi.org/10.1175/MWR-D-12-00216.1, publisher: American Meteorological Society Section: Monthly Weather Review, 2013.

Morice, C. P., Kennedy, J. J., Rayner, N. A., Winn, J. P., Hogan, E., Killick, R. E., Dunn, R. J. H., Osborn, T. J., Jones, 920 P. D., and Simpson, I. R.: An Updated Assessment of Near-Surface Temperature Change From 1850: The HadCRUT5 Data Set, Journal of Geophysical Research: Atmospheres, 126, e2019JD032 361, https://doi.org/10.1029/2019JD032361, _eprint: https://onlinelibrary.wiley.com/doi/pdf/10.1029/2019JD032361, 2021.

Mudryk, L.: Historical gridded snow water equivalent and snow cover fraction over Canada from remote sensing and land surface models, available at http://climate-scenarios.canada.ca/?page=blended-snow-data (last accessed Oct 2022), http://climate-scenarios.canada. 925 ca/?page=blended-snow-data, 2020.

Neelin, J. D., Jin, F.-F., and Syu, H.-H.: Variations in ENSO phase locking, Journal of Climate, 13, 2570–2590, 2000.

Notz, D. and SIMIP Community: Arctic Sea Ice in CMIP6, Geophysical Research Letters, 47, e2019GL086 749, https://doi.org/10.1029/2019GL086749, _eprint: https://agupubs.onlinelibrary.wiley.com/doi/pdf/10.1029/2019GL086749, 2020.

Omar, A. H., Winker, D. M., Vaughan, M. A., Hu, Y., Trepte, C. R., Ferrare, R. A., Lee, K.-P., Hostetler, C. A., Kittaka, C., Rogers, R. R., 930 Kuehn, R. E., and Liu, Z.: The CALIPSO Automated Aerosol Classification and Lidar Ratio Selection Algorithm, J. Atmospheric Ocean. Technol., 26, https://doi.org/10.1175/2009JTECHA1231.1, 2009.

Planton, Y. Y., Guilyardi, E., Wittenberg, A. T., Lee, J., Gleckler, P. J., Bayr, T., McGregor, S., McPhaden, M. J., Power, S., Roehrig, R., Vialard, J., , and Voldoire, A.: Evaluating climate models with the CLIVAR 2020 ENSO Metrics Package, Bulletin of the American Meteorological Society, 102, E193 – E217, https://doi.org/10.1175/BAMS-D-19-0337.1, 2021.

Platnick, S., King, M., and Hubanks, P.: MODIS Atmosphere L3 Monthly Product. NASA MODIS Adaptive Processing System, Goddard Space Flight Center, https://doi.org/10.5067/MODIS/MOD08_M3.061, 2017.





Polvani, L. M., Sun, L., Butler, A. H., Richter, J. H., and Deser, C.: Distinguishing Stratospheric Sudden Warmings from ENSO as Key Drivers of Wintertime Climate Variability over the North Atlantic and Eurasia, Journal of Climate, 30, 1959–1969, https://doi.org/10.1175/JCLI-D-16-0277.1, publisher: American Meteorological Society Section: Journal of Climate, 2017.

Qaddouri, A. and Lee, V.: The Canadian Global Environmental Multiscale model on the Yin-Yang grid system, Quarterly Journal of the Royal Meteorological Society, 137, 1913–1926, https://doi.org/10.1002/qj.873, _eprint: https://onlinelibrary.wiley.com/doi/pdf/10.1002/qj.873, 2011.

Reynolds, R. W., Smith, T. M., Liu, C., Chelton, D. B., Casey, K. S., and Schlax, M. G.: Daily high-resolution-blended analyses for sea surface temperature, Journal of Climate, 20, 31 004, https://doi.org/10.1088/2515-7620/abf295, 2007.

Roach, L. A., Dörr, J., Holmes, C. R., Massonnet, F., Blockley, E. W., Notz, D., Rackow, T., Raphael, M. N., O'Farrell, S. P., Bailey, D. A., and Bitz, C. M.: Antarctic Sea Ice Area in CMIP6, Geophysical Research Letters, 47, 1–10, https://doi.org/10.1029/2019GL086729, 2020.

Santer, B. D., Po-Chedley, S., Mears, C., Fyfe, J. C., Gillett, N., Fu, Q., Painter, J. F., Solomon, S., Steiner, A. K., Wentz, F. J., Zelinka, M. D., and Zou, C.-Z.: Using Climate Model Simulations to Constrain Observations, Journal of Climate, pp. 1–59, https://doi.org/10.1175/JCLI-D-20-0768.1, 2021.

Scinocca, J. F. and McFarlane, N. A.: The parametrization of drag induced by stratified flow over anisotropic orography, Quarterly Journal of the Royal Meteorological Society, 126, 2353–2393, http://onlinelibrary.wiley.com/doi/10.1002/qj.49712656802/abstract, 2000.

Shell, K. M., Kiehl, J. T., and Shields, C. A.: Using the radiative kernel technique to calculate climate feedbacks in NCAR's Community Atmospheric Model, Journal of Climate, 21, 2269–2282, 2008.

Sigmond, M. and Scinocca, J. F.: The Influence of the Basic State on the Northern Hemisphere Circulation Response to Climate Change, Jour-
nal of Climate, 23, 1434–1446, https://doi.org/10.1175/2009JCLI3167.1, publisher: American Meteorological Society Section: Journal of Climate, 2010.

Sigmond, M., Scinocca, J. F., Kharin, V. V., and Shepherd, T. G.: Enhanced seasonal forecast skill following stratospheric sudden warmings, Nature Geoscience, 6, 98–102, https://doi.org/10.1038/ngeo1698, 2013.

Sigmond, M., Fyfe, J. C., Saenko, O. A., and Swart, N. C.: Ongoing AMOC and related sea-level and temperature changes after achieving
the Paris targets, Nature Climate Change, 10, 672–677, https://doi.org/10.1038/s41558-020-0786-0, 2020.

Simpson, I. R., Hitchcock, P., Seager, R., Wu, Y., and Callaghan, P.: The Downward Influence of Uncertainty in the Northern Hemisphere Stratospheric Polar Vortex Response to Climate Change, Journal of Climate, 31, 6371–6391, https://doi.org/10.1175/JCLI-D-18-0041.1, publisher: American Meteorological Society Section: Journal of Climate, 2018.

Soden, B. J., Held, I. M., Colman, R., Shell, K. M., Kiehl, J. T., and Shields, C. A.: Quantifying climate feedbacks using radiative kernels,
Journal of Climate, 21, 3504–3520, 2008.

Song, Q., Zhang, Z., Yu, H., Ginoux, P., and Shen, J.: Global Dust Optical Depth Climatology Derived from CALIOP and MODIS Aerosol Retrievals on Decadal Timescales: Regional and Interannual Variability, Atmospheric Chem. Phys., 21, https://doi.org/10.5194/acp-21-13369-2021, 2021.

Song, Z., Liu, H., and Chen, X.: Eastern equatorial Pacific SST seasonal cycle in global climate models: From CMIP5 to CMIP6, Acta
Oceanologica Sinica, 39, 50–60, https://doi.org/10.1007/s13131-020-1623-z, 2020.

Swart, N. C., Cole, J., Kharin, S., Lazare, M., Scinocca, J., Gillett, N., Anstey, J., Arora, V., Christian, J., Hanna, S., Jiao, Y., Lee, W., Majaess, F., Saenko, O., Seiler, C., Seinen, C., Shao, A., Solheim, L., Salzen, K. v., Yang, D., and Winter, B.: The Canadian Earth System Model (CanESM) - v5.0.3, https://doi.org/10.5281/zenodo.3251114, 2019a.





Swart, N. C., Cole, J. N. S., Kharin, V. V., Lazare, M., Scinocca, J. F., Gillett, N. P., Anstey, J., Arora, V., Christian, J. R., Hanna, S., Jiao, Y., Lee, W. G., Majaess, F., Saenko, O. A., Seiler, C., Seinen, C., Shao, A., Sigmond, M., Solheim, L., von Salzen, K., Yang, D., and Winter, B.: The Canadian Earth System Model version 5 (CanESM5.0.3), Geoscientific Model Development, 12, 4823–4873, https://doi.org/10.5194/gmd-12-4823-2019, 2019b.

Swart, N. C., Cole, J., Kharin, S., Lazare, M., Scinocca, J., Gillett, N., Anstey, J., Arora, V., Christian, J., Hanna, S., Jiao, Y., Lee, W., Majaess, F., Saenko, O., Seiler, C., Seinen, C., Shao, A., Solheim, L., Salzen, K. v., Yang, D., and Winter, B.: The Canadian Earth System Model (CanESM) - v5.1.6, https://doi.org/10.5281/zenodo.7786802, 2023.

Swenson, S. C. and Lawrence, D. M.: A new fractional snow-covered area parameterization for the Community Land Model and its effect on the surface energy balance, Journal of Geophysical Research: Atmospheres, 117, https://doi.org/10.1029/2012JD018178, 2012.

Taylor, K. E.: Summarizing multiple aspects of model performance in a single diagram, Journal of Geophysical Research: Atmospheres, 106, 7183–7192, https://doi.org/10.1029/2000JD900719, 2001.

Torrence, C. and Compo, G. P.: A practical guide to wavelet analysis, Bulletin of the American Meteorological society, 79, 61–78, 1998.

Verseghy, D., McFarlane, N., and Lazare, M.: CLASS—A Canadian land surface scheme for GCMs, II. Vegetation model and coupled runs, International journal of climatology, 13, 347–370, 1993.

Virgin, J. G., Fletcher, C. G., Cole, J. N. S., von Salzen, K., and Mitovski, T.: Cloud Feedbacks from CanESM2 to CanESM5.0 and their influence on climate sensitivity, Geoscientific Model Development, 14, 5355–5372, https://doi.org/10.5194/gmd-14-5355-2021, publisher: Copernicus GmbH, 2021.

Vogel, A., Alessa, G., Scheele, R., Weber, L., Dubovik, O., North, P., and Fiedler, S.: Uncertainty in Aerosol Optical Depth From Modern Aerosol-Climate Models, Reanalyses, and Satellite Products, J. Geophys. Res. Atmospheres, 127, https://doi.org/10.1029/2021JD035483, 2022.

von Salzen, K., Scinocca, J. F., McFarlane, N. A., Li, J., Cole, J. N. S., Plummer, D., Verseghy, D., Reader, M. C., Ma, X., Lazare, M., and Solheim, L.: The Canadian Fourth Generation Atmospheric Global Climate Model (CanAM4). Part I: Representation of Physical Processes, Atmosphere-Ocean, 51, 104–125, https://doi.org/10.1080/07055900.2012.755610, 2013.

Voss, K. K. and Evan, A. T.: Dust aerosol optical depth, https://doi.org/10.1594/PANGAEA.909140, supplement to: Voss, KK; Evan, AT (2019): A new satellite-based global climatology of dust aerosol optical depth. Journal of Applied Meteorology and Climatology, https://doi.org/10.1175/JAMC-D-19-0194.1, 2019.

Wang, W. and McPhaden, M. J.: The Surface-Layer Heat Balance in the Equatorial Pacific Ocean. Part I: Mean Seasonal Cycle, Journal of Physical Oceanography, 29, 1812 – 1831, https://doi.org/10.1175/1520-0485(1999)029<1812:TSLHBI>2.0.CO;2, 1999.

Williamson, D., Blaker, A. T., Hampton, C., and Salter, J.: Identifying and removing structural biases in climate models with history matching, Climate Dynamics, 45, 1299–1324, https://doi.org/10.1007/s00382-014-2378-z, 2015.

Williamson, D. B., Blaker, A. T., and Sinha, B.: Tuning without over-tuning: parametric uncertainty quantification for the NEMO ocean model, Geoscientific Model Development, 10, 1789–1816, https://doi.org/10.5194/gmd-10-1789-2017, publisher: Copernicus GmbH, 2017.

Winker, D. M., Vaughan, M. A., Omar, A., Hu, Y., Powell, K. A., Liu, Z., Hunt, W. H., and Young, S. A.: Overview of the CALIPSO Mission and CALIOP Data Processing Algorithms, J. Atmospheric Ocean. Technol., 26, https://doi.org/10.1175/2009JTECHA1281.1, 2009.

Young, A. H., Knapp, K. R., Inamdar, A., Hankins, W., and Rossow, W. B.: The International Satellite Cloud Climatology Project H-Series climate data record product, Earth System Science Data, 10, 583–593, https://doi.org/10.5194/essd-10-583-2018, publisher: Copernicus GmbH, 2018.





Zelinka, M. D., Myers, T. A., McCoy, D. T., Po-Chedley, S., Caldwell, P. M., Ceppi, P., Klein, S. A., and Taylor, K. E.: Causes of Higher Climate Sensitivity in CMIP6 Models, Geophysical Research Letters, 47, e2019GL085 782, https://doi.org/10.1029/2019GL085782, _eprint: https://onlinelibrary.wiley.com/doi/pdf/10.1029/2019GL085782, 2020.

1015 Zhai, C., Jiang, J. H., and Su, H.: Long-term cloud change imprinted in seasonal cloud variation: More evidence of high climate sensitivity, Geophysical Research Letters, 42, 8729–8737, 2015.

Zhang, J. and Rothrock, D. A.: Modeling Global Sea Ice with a Thickness and Enthalpy Distribution Model in Generalized Curvilinear Coordinates, Monthly Weather Review, 131, 845–861, https://doi.org/10.1175/1520-0493(2003)131<0845:MGSIWA>2.0.CO;2, publisher: American Meteorological Society Section: Monthly Weather Review, 2003.

1020 Zhao, A., Ryder, C. L., and Wilcox, L. J.: How Well Do the CMIP6 Models Simulate Dust Aerosols?, Atmospheric Chem. Phys., 22, https://doi.org/10.5194/acp-22-2095-2022, 2022.