# Peer review of "Improvements in the Canadian Earth System Model (CanESM) through systematic model analysis: CanESM5.0 and CanESM5.1"

_Geoscientific Model Development, 2023_

## Author Comment (AC1)

**Response to Reviewer 1**

The authors document ongoing model analysis efforts for the Canadian Earth System Model (CanESM). Upgrades from CanESM5.0 to CanESM5.1 (p1) are described, featuring an extensive coverage of one tuning fix related to the hybridization of advective atmospheric tracers which got the model rid of spurious unrealistic spikes in simulated stratospheric temperatures (I have to stress that Section 5.1 particularly reads like a detective novel set in an ESM context, and that's a compliment). The authors then present CanESM5.1 p2, an alternate version of CanESM5.1 p1 specifically retuned for better representing ENSO variability. Finally, the authors address less successful attempts in correcting other prominent biases (e.g., overestimation of North Atlantic sea ice, cold bias over the Himalaya) providing further development perspectives inferred from preliminary experimentations (mostly based on offline Earth system model component simulations), which also provide valuable insight to the community.

Yet what makes this paper stands out to my eyes lies in these bias corrections being set in the framework of the "Analysis for development" (A4D) internal effort. This provides a refreshingly transparent and assuming outlook on model development strategy, which is extremely valuable to the worldwide modelling community. The two successful bias corrections listed above can thus be perceived as two flagship applications of the A4D procedure.

The manuscript is of high quality and clearly falls within the scope of GMD. It represents a significant amount of technical work which tends to often be overlooked in literature. As said above, the earnestness and transparency with which it approaches difficult challenges in model developments is a big plus. However, I think that clarifying some specific points would help the reader and hopefully improve the manuscript. Therefore, I recommend its publication after addressing the comments listed below.

**Thank you for going through the manuscript so carefully and for providing such thoughtful and positive comments!**
* * *
Specific comments

1) The three-fold typology in model issues and the classification of encountered biases in the conclusion is one of the manuscript's strengths. Nevertheless, while I kind of

understand where the categories are coming from and acknowledge that there is no perfect way to define them, I think that they could be worded in a sounder manner with little to no impact on the manuscript. I'm OK with the definition of community-specific issues as related to physical phenomena that are universally hard to represent, and thus translate into issues in virtually all models. Then, I'd distinguish remaining issues as model-specific or *configuration*-specific, instead of version-specific. "Model versions" is somehow vague, and strictly speaking some model version changes could yield major impacts on the dynamical core and resulting physics (e.g., relaxing the hydrostaticity, or changing the vertical coordinate -- which are big model changes, I admit). Configuration-specific issues would be those that can be addressed by changing anything but the model source code (resolution, new tuning, input data). Model-specific issues would then be issues that call for model developments, which would include parameterization updates/changes, or dynamical core updates. Finally, regardless of the way issue categories are defined, classifying issues is not straightforward, and boundaries in-between categories are porous. The manuscript's conclusion kind of infers it when discussing where to categorize the discussed biases, but I think that insisting on this classification exercise not being rocket-science would make sense as classes are introduced in Section 2.

**We thank the reviewer for identifying the issues related to the three categories of model biases that were introduced in Section 2. What we specifically mean by model version and our rationale for introducing these categories were not sufficiently clear and this impacted the discussion in Section 6. The utility of categorizing biases/issues in the way we have chosen was to provide an initial indication of where the sources of the biases potentially reside. We have modified the discussion surrounding their introduction in Section 2 to clarify this point and have modified their discussion in the conclusions to be more consistent with this intended purpose.**

**New text to section 2 is added (Lines 49-72)**

> **The focus of model development is to produce new model versions for use in applications (such as CMIP). Here, "model version" is synonymous with model finalization, whereby all properties of a model are decided and held fixed (e.g., model resolution, dynamical core, formulation of physical parameterizations, assignment of free parameters, etc.). Such model versions are assigned a version number, or ID, for reference (the naming convention for CanESM versions is described in Section 3). A primary goal of model development is to**

bring about improvement in the model properties and behaviour with each succeeding version. To accomplish this, model development efforts generally focus on systematic errors, or "issues", that have been found with the properties and behaviour of recent and predecessor model versions. Model issues are generally first identified as a simple bias in some quantity relative to observations, but ideally they would ultimately be connected to, and understood in terms of, the specific representation of physical processes or dynamical mechanisms in the model. While there are potentially many ways to characterize model issues, we have found it useful to broadly categorize them into three types:

> *Version-specific* issues are unique to a particular model version; by definition they are due to changes made in that version relative to its predecessor.

> *Model-systemic* issues are shared across multiple versions of a model; such issues have proved relatively insensitive to recent development efforts.

> *Community-systemic* issues are systematic errors shared by multiple diverse climate models (e.g., across CMIP ESMs), and are typically due to a community-wide issue related to the absence, or manner of treatment, of one or more physical processes.

Changes to Section 6 include the following paragraphs (Lines **693-717**):

To aid in the analysis of these biases, three broad categories of model issues were employed to help identify their sources: *version-specific*, *model-systemic*, and *community-systemic* (Section 2). The majority of biases found in CanESM5.0/5.1 appear to be *version-specific* issues (i.e., particular to this latest major model version). These include the occurrence of spurious "spikes" in stratospheric temperature and dust, and tropospheric dust storms (Section 5.1, an issue that was resolved in CanESM5.1); degraded ENSO in relative to its predecessor CanESM2/CanCM4 (Section 5.2.1, a bias that was reduced in CanESM5.1-p2); a high ECS (Section 5.2.2, reduced by about 30% in CanESM5.1-p2); excessive North Atlantic sea ice during late winter (Section 5.3.1); and too few sudden stratospheric warmings (Section 5.3.3). *Model-systemic* issues include the cold winter bias above sea ice (Section

**5.3.2) and the Himalayan cold bias (Section 5.3.4).** *Community-systemic* **biases – i.e., biases in CanESM that resemble those seen in multiple CMIP models – include the excessive westward extension of SST anomalies (Section 5.2.1); and the Himalayan cold bias (Section 5.3.4).** *Community-systematic* **errors are important to identify because their potential influence on multi-model analyses cannot easily be removed by averaging across a multi-model ensemble (since many models share these same errors) and hence documenting them is important for informing users of the output of these models.**

**In general, the nature of any model bias can be further understood in terms of it being parametric or structural. Parametric issues are dependent on parameter values primarily associated with physical parameterizations, while structural ones are dependent on model formulation (essentially, everything else). For** *version-specific* **biases, the question of whether they are dependent on model tuning is highly relevant.** *Model-systemic* **biases suggest structural origins but parametric explanations might also be relevant if a particular tuning approach has impacted multiple model versions. For** *community-systemic* **biases, it is almost certainly the case that their origins are structural. The p2 variant of CanESM5.1 (Section 3.2.1) was an attempt to gain some insight into the parametric nature of biases present in the p1 variant of CanESM5.1 (and in CanESM5.0, given its strong similarity to CanESM5.1-p1). Ideally, one would span all parameter values to gauge such sensitivity. At the CCCma we have begun using a more objective tuning approach (e.g. Hourdin et al., 2017) that uses history matching methods based on Bayesian statistics to determine suitable values for CanESM's free physical parameters in finalizing model versions (Williamson et al., 2015, 2017). The application of this approach to parameter tuning spans all possible values of selected free parameters and so should be a valuable tool to identify the nature of model biases as either parametric or structural.**

In addition, references to these types of issues have been added to Line 258 (Section 5.1), Lines 348, 351 and 352 (Section 5.2.1), Line 424 (Section 5.2.2), Line 458 (Section 5.3.1), Line 505 (Section 5.3.2), Line 552 (Section 5.3.3), and Lines 617 and 676 (Section 5.3.4).

2) There are a few different model versions being used and I found the indexing and their interdependency confusing at times.

2a. L. 26: "CanESM5" is mentioned here, while only "CanESM5.0" had previously been mentioned. My understanding is that "CanESM5" refers to both CanESM5.0 and CanESM5.1. Is that right? If so, it'd probably be best to define what exactly is meant by "CanESM5", and to use it once CanESM5.1 has been introduced (if it is included in CanESM5, that is).

**We apologize for the confusion. In this instance, CanESM5, refers to CanESM5.0 only. We have corrected this here, and elsewhere in the paper where "CanESM5" was inadvertently used (instead of referring to CanESM5.0 or 5.1 specifically).**

2b. L. 99: the "p" index for the patch version is quite confusion-prone, since the "p" letter is also used for distinguishing other things (CMIP physics-related ensemble) which are important to the rest of the manuscript. Since the patch version is not used anywhere else in the manuscript (unless I misunderstood), lines 99 to 103 can be removed from the paper, for the sake of clarity. IMO every GMD reader already knows that CanESM5.0 is more different from CanESM2 than CanESM5.1 is from CanESM5.0. And they can read and understand the manuscript without knowing about patch versions.

**We apologize for the confusion. We have removed use of the "p" label for the patch version (in the revised manuscript, "p" only refers to the physics variant) and have clarified the naming convention by rewording the sentence (Line 114-116):**

> **Swart et al. (2019b) described the model characteristics and climatological properties of CanESM5.0, which was also referred to as "CanESM5" in that study, and has the precise version name CanESM5.0.3. CCCma uses a three digit naming convention for our models: Major.Minor.Patch. CCCma uses a three digit naming convention for our models: Major.Minor.Patch.**

**While the reviewer is correct that the patch version is not referred to anywhere else in the manuscript, we think it is valuable to report the full naming convention (Major.Minor.Patch) for completeness, and because this accords with a commonly used software versioning convention (Semantic Versioning) that potentially could be relevant to anyone accessing the publicly available model source code.**

The distinction between Patch and physics variant is further clarified by adding **(Lines 119-121):**

> **Further changes to model configuration, such as adjustment of tunable parameters, can also be represented by the physics variant label, as described further below[1]**
>
> > **[1] The physics variant label identifies alternate model configurations that have physically meaningful differences, and hence potential impact on the simulated climate. It is completely unrelated to the patch version, which denotes technical changes that do not impact the simulated climate**

2c. L. 118: for the sake of clarity, I would say right there two variants of CanESM5.1 have been implemented, and that the common traits between both these variants, which are new compared to CanESM5.0, are the bullet points below. Actually, I'd create subsubsections for the CanESM5.1 three bullet points (common to p1 and p2), and then another one for presenting the two variants, so that the distinction between both levels is clearer.

**We agree that it is important that the distinctions between the multiple model versions discussed in the paper are made as clear as possible. The opening of Section 3.2 now reads (Lines 138-141):**

> **CanESM5.1 is a new version of CanESM, for which two model versions labelled "p1" and "p2" have been released. The distinctions between the p1 and p2 variants are described below (Section 3.2.1). It is important to note that the differences between the p1 and p2 variants of CanESM5.1 are completely independent from, and unrelated to, the differences between the p1 and p2 variants of CanESM5.0. Figure 2 summarizes the evolution from CanESM5.0 to CanESM5.1, listing the key model changes for each version and its variants.**

**This is followed by the list of changes in 5.1 compared to 5.0. Distinctions between CanESM5.1 p1 and p2 variants are now described in a separate subsubsection (3.2.1) as suggested by the reviewer.**

**In addition, we have added a new Figure (Figure 2) that summarizes the evolution from CanESM5.0 to CanESM5.1, listing the key model changes for each version and its variants.**

2d. Please specify whether CanESM5.1 p2 is built on top of CanESM5.1 p1, or is it just targeted at different applications/diagnostics? I think the latter, right?

**Correct, it is the latter. The two model versions use the same code and differ only in their tuning parameters. This has been clarified by adding a sentence to the beginning of Subsection 3.2.1 (Line 160-161):**

> **The source code of both p1 and p2 CanESM5.1 variants is essentially identical, with the two model versions differing only in the tuning of parameterizations in the atmospheric component of the model (as indicated in Figure 2).**

2e. Ideally, it would be very nice to have a diagram (e.g. Venn and/or arrow-connected boxes) to help the reader differentiate the different model versions and their relationships: is X a successor or an alternate version of Y, etc.

**Thank you for this excellent suggestion. In response to the reviewer's comment we have added a new figure (Figure 2), which summarizes the main differences and relationships between the different model versions.**

3) Bare ground fraction (L. 286 on): I don't understand the discussion. The first sentence of the paragraph says that there was an interpolation error, and then there's a discussion (and runs performed) to decide whether the correction should be applied or not? If this is an interpolation error, shouldn't it just be corrected, or am I missing something? If the "wrong" bare ground fraction yields better results, is it still reasonable to keep it? Are we getting on overtuning grounds? If so, it's worth being explicitly mentioned.

**Thank you for this comment. The intention of the sensitivity experiments was not to inform the decision whether or not the interpolation error should be corrected. We agree with the reviewer that any error should be corrected. Instead, the sensitivity experiments served to quantify the impact of that error on already existing simulations, as now explicitly stated in the revised version (Lines 325-326):**

> **While this error will be corrected in future versions of CanESM, we here quantify the impact of this error on already existing simulations.**

4) L. 436 – 474: consider putting the details in appendix and leave in the main body just the three bullet points around L. 432, and there comes no definitive conclusions / solutions from this. It's still valuable information for modellers, and I can feel the sweat and tears, but this part is a bit too lengthy and unconclusive yet to be worth a main body spot to my external eyes. Also, L. 471 – L. 474 (up until "considered") read a bit like a funding application (and the bulk of it is already in the manuscript's conclusion). Please consider removing it or keeping it for a next paper where these things are actually presented.

**Following the reviewer's suggestion, we have moved L. 436-474 to Appendix C. We have left L. 471-474 in the paper as they provide potentially useful ideas on how to resolve the issue.**
* * *
Technical comments

- L. 46: I wouldn't use the word "specific" to describe the biases dealt with in section 5, as these biases are present in both CanESM5.0 and CanESM5.1, but also in other ESM models (as the manuscript rightfully later says so). "Persisting biases", maybe?

**We changed 'specific' to 'various' (Line 46)**

- L. 94 "a new"

**Corrected**

- L. 104: please provide a reference for CanESM2

**References for CanESM2 have been added (Line 124)**

- L. 119: if CanESM5.0 p1 and p2 are the same as CanESM5 p1 and p2 as per Swart et al. (2019), please specify it here. If not, explain

**Yes, they are. Text has been added to the previous Section (3.1) to clarify that "CanESM5" in Swart et al. 2019b refers to CanESM5.0 (Line 114-115). This was stated in a footnote in the submitted version, but we have removed the footnote and made this information more prominent by stating it at the beginning of Section 3.1. In addition we have added the following sentence (Line 131-132):**

**Two CanESM5.0 model variants, labelled by the physics variant labels "p1" and "p2", have been released, as described in Swart et al. (2019b).**

- L. 124 – 127: papers are meant to be read by human beings, not parsed by a namelist-reading program. Please refer to the method as "second-order conservative". It could also be worth explaining that on top of the fields themselves, second-order methods remap their spatial gradients in a conservative way (which fits the desired results). And potentially cite Jones 1999 (https://doi.org/10.1175/1520-0493(1999)127%3C2204:FASOCR%3E2.0.CO;2 ).

**We have rewritten this bullet point to read (Line 148-150):**

> **– An improved remapping of atmospheric heat fluxes that are passed to the ocean grid within the coupler, by changing to a second-order conservative scheme (Jones, 1999) that preserves a smooth derivative across grid cells[2] . This helped to reduce the nonphysical "blocky" pattern in the heat fluxes on the ocean grid (see Figure A1).**

> > **[2]Specifically, the remapping was changed from the Earth System Modeling Framework (ESMF) *conservative* routine in the p2 version of CanESM5.0, to the *conservative2* option in CanESM5.1. For further details we refer to the ESMF reference manual ( https://earthsystemmodeling.org/docs/release/latest/ ESMF_refdoc.pdf).**

- L. 128 – 135: this represents a tremendous amount of extremely useful and ungrateful work – congratulations. Did you notice any impact on performance (speed, memory requirements, etc.)? Also, does "same bit pattern" mean "bit identical"? If so, just say the code F90-izing is bit-identical (and the array transformation isn't, but climatologically equivalent).

**The changes did not have any major impact on model performance. Regarding the bit patterns, we have modified the text to clarify the terminology (Line 156-158):**

> **The syntax changes are bit-identical (i.e., they preserved the bit pattern of the model). The changes to array structure are not bit-identical but had no statistically discernible impact on the model climate.**

- Fig. 2 and others in appendix: I'm fine with using ERA5 as a reference, but labelling it as "Obs" in figure subtitles is taking it a bit too far.

**We use different observation-based products throughout the paper to compare the model with the real world. For simplicity, and following other papers including Swart et al (2019b), we use the abbreviation 'Obs' in the labelling of many of the panels. In all cases where this label is used, the specific observation-based product is clearly specified in the caption of the Figures. In Figures A6-A8 in the revised manuscript we have added explicit mention of ERA5 as the observational dataset (previously these figure captions only referred to Fig. 2 for this information).**

- L. 169: "…, as \*supported\* by panels…"

**Wording changed following the reviewer's suggestion**

- L. 182: please make these reports accessible permanently, e.g. by uploading them to Zenodo, or by adding them as supplementary material to the paper.

**We have made these reports available at [https://doi.org/10.5281/zenodo.8196308](https://doi.org/10.5281/zenodo.8196308), which is now noted in the revised manuscript (Line 218)**

- L. 189: include a citation (e.g. https://www.pnas.org/doi/full/10.1073/pnas.1906556116 ) after "observations".

**Citation added**

- L. 189 "that that"

**Corrected**

- L. 189 – 191: it definitely is noteworthy that one ensemble member has positive February trend, however suggesting this hints at internal variability as a driver of observed positive Antarctic sea-ice trend feels like a bit of a stretch.

**We have removed the suggestion that internal variability is a driver of the observed positive Antarctic sea-ice trend. Also, upon closer investigation we found that more than one CanESM5.1 ensemble member simulate a positive Antarctic sea ice trend, which is now reflected in the new text (Line 225-227):**

> **We note though, that as a result of internal variability in CanESM5.1-p1, 3 of 47 ensemble members (6%) simulate a positive Antarctic sea ice trend in February (for CanESM5.1-p2: 1 of 25 members, 4%), suggesting that CanESM5.1 is consistent with observations.**

- L. 196: "such as the run"

**Changed to 'future projections such as the projection that follows the SSP5-8.5 scenario' (Line 231-232)**

- L. 288: "This issue is investigated by comparing two atmosphere-only simulations (with and without bare ground fraction correction), in which the atmosphere is nudged to reanalysis so that the observed meteorological conditions, which have a large impact on dust, are well reproduced"

**Wording changed following the reviewer's suggestion (Line 326-328)**

- Fig. 11: I think that here "seasonal cycle" refers to monthly means – annual means, right? If so, it'd be worth specifying it in the figure caption. Also, please specify how the ensemble members were picked (presumably randomly).

**We have changed the caption of this Figure (now Figure 12) to:**

> **Mean seasonal cycle (i.e., the monthly minus annual means) of equatorial Pacific SST during 1950-2014 for a) ERSSTv5, b) CanESM2, c) CanESM5.0p-2, d) CanESM5.1-p1 and e) CanESM5.1-p2. Each model version is represented by the mean of the first 10 ensemble members.**

- L. 348: to me "gradients" are local properties, by definition. I'd describe Diff_CE as a "large-scale zonal variations"

**Changed "zonal gradient" to "large-scale zonal differences" since the metric is a simple difference between two regions. (Line 387)**

- L. 360: is/was CanOM4 an in-house ocean model, or adapted from another model as CanNEMO is from NEMO? Please provide a bit more detail and fitting citations (including the NEMO book for CanNEMO).

**We have added details on CanOM4 in footnote #6 (page 21):**

> **CanOM4 was derived from the NCAR Climate System Model ocean component (Gent et al., 1998), with significant modifications and additions to physical parameterizations as summarized in Arora et al (2011) and Merryfield et al (2013)**

**In addition, we have added a reference to the NEMO book (Line 400).**

- L. 363: a complex interplay of both oceanic and atmospheric processes

**Corrected**

- L. 366: isn't CanESM5.1 p1's climate sensitivity virtually the same as CanESM5.0? If so, please rephrase, e.g., "CanESM5.1 p2 was successfully tuned to reduce (-20%) the overestimated climate sensitivity of CanESM5.0 and 5.1 p1".

**Yes, that is correct. We changed the wording to (Line 408-409)**

> **A second metric that was used to tune the p2 version of CanESM5.1 was historical warming, which was unrealistically high in CanESM5.0 (Swart et al., 2019b) and is about 20% weaker than in CanESM5.0 and the p1 version of CanESM5.1 (Section 4.2).**

**Note that the real climate sensitivity is unknown, and hence tuning efforts were targeted at historical warming (which is known), as opposed to climate sensitivity.**

- L. 413: I think replacing "air-sea interactions" with "sea-surface buoyancy loss" would be more focused. Please consider citing literature, e.g. https://link.springer.com/article/10.1007/s00382-019-04802-4

**Agreed. Wording changed following the reviewer's suggestion, and reference to Kostov et al (2019) added (Line 461-462)**

- L. 410: since you're talking about deep convection, could you specify the choice of convection parameterization in CanNEMO? Enhanced diffusivity?

**Yes, the NEMO enhanced diffusivity scheme is used for convection**

- L. 418: please specify *ocean* vertical diffusivity.

**Modified following the reviewer's suggestion (Line 467)**

- L. 425 – 426: which bulk formulae, which runoff observations, and which SSS?

**These details are provided in Griffies et al. (2016), as cited. It is the CORE forcing, based on the NCEP1 reanalysis, the CORE bulk formula, Dai and Trenberth based runoff, and SSS for World Ocean Atlas.**

- L. 430: it could also be that the CanESM forcings have been obtained from coupled runs, so that they have the imprint of the ocean surface biases (the same way observed SAT have an imprint of sea-ice presence or lack thereof).

**Agreed, thank you for pointing this out. We have clarified this by changing the text to (Line 477-482):**

> **Under CanESM5.0 forcing, the ocean-only model reproduces the biases seen in the coupled model (Figure 16b). While it would be tempting to attribute the ocean and sea ice biases to CanESM5.0 forcings, it is important to realize that the CanESM5.0 forcings themselves have an imprint of the ocean surface biases in CanESM5.0 and hence that inherent ocean surface biases cannot be excluded as a possible cause. In conclusion, while these ocean-only experiments are instructive, they lack coupled feedbacks, and do not provide definitive evidence about the cause of the biases, or their solutions.**

- L. 477: ocean physics tuning or adjusments?

**Changed to 'ocean physics tuning' (Line 490)**

- L. 481: for sea-ice covered ocean cells, are SSTs the sea-ice surface temperature, or the temperature of the liquid ocean? If the latter, then SSTs can't get below freezing point anyway, which may explain the lack of signal.

**SSTs are the liquid ocean temperature and cannot drop below freezing, but they can be above freezing (and hence can induce melt). The point here is just that this is not a likely source of the bias.**

- Table 2: please briefly provide references for the empirical function.

**We changed the wording of the caption of this table (now Table 3):**

> **snow conductivity is a function of snow density (this scheme was used in CanESM2; McFarlane et al 1992).**

- L. 484: "This reasoning…": or that this is a community-systemtic issue?

**Thank you for the interesting suggestion. Using an adapted version of Lisa Bock's ESMValTool recipe, we have investigated if the cold winter bias above sea ice is a common model bias. The figure below shows the CMIP6 multi-model near surface temperature bias in DJF (left) and JJA (right). While there is certainly a cold bias in the Arctic in DJF in CMIP6, there is not a robust cold bias in JJA over Antarctic sea ice (at least, not to the extent we see it in CanESM). This suggests that the winter cold bias above sea ice is CanESM specific, and not a common bias among CMIP6 models.**

[Figure]

- L. 515: missing period.

**Corrected**

- L. 521: lonely )

**Corrected**

- Table 3: why capital W

**Corrected**

- L. 590: it may be worth talking about "overtuning" here (and a reference would be nice).

**Thank you for this suggestion. We have changed the text to (Line 607-608):**

**This highlights intrinsic difficulties often encountered in tuning efforts: the improvement of one aspect of the circulation can often lead to the deterioration of another, and focussing on only one aspect of the climate may lead to 'overtuning'.**

- L. 591: please remove sentence starting with "Physically" – it generically can be said about modelling in general, so not very engaging.

**Removed 'Physically'**

- L. 622: please first introduce the datasets (with adequate citations), and then explain that they've been tested as model forcing. The reader doesn't know what these acronyms mean when they reach line 622.

**To clarify the nature of the forcing datasets used to drive offline CLASSIC simulations, we have separated the description of the offline simulations and their forcing datasets, now its own paragraph (Line 637-647), from description of the results (Fig. 20c-h), which is now in the following paragraph (Line 648-655). The beginning of this paragraph indicates that the datasets have been well tested as CLASSIC forcings ("Offline simulations of CLASSIC are routinely performed…"). References for the two observation-based forcing datasets (CRU-JRA and GSWP3), and their descriptions, are now placed immediately after the sentence saying that these datasets are used to drive CLASSIC offline simulations, and we have also added the full names of these datasets (i.e., defined the CRU-JRA and GSWP3 acronyms at their first mention in the text); see Line 641-647.**

- L. 625: required to drive CLASSIC, right?

**Correct. We have added clarification of this by referring directly to CLASSIC at the end of the sentence (Line 642-643):**

**provides 6-hourly values of seven meteorological variables that are required to drive a land model such as CLASSIC.**

- L. 638: please rephrase – not sure what "complete" means here (and seems counter-intuitive)

Here "complete" referred to the multi-variable nature of these datasets, as required to drive CLASSIC offline simulations. To avoid confusion we have deleted "complete" and rephrased the sentence slightly (Line **657-659**):

> **Note that the CRU-JRA and GSWP3 are meteorological datasets that are available only over land (and are used to drive CLASSIC offline, as described above), while GPCP is an observation-based precipitation dataset and is available for the whole globe.**

- L. 646: "As a result" is a bit fast here, especially as GMD is specialized on this. I think (?) that the authors are thinking of reduced blocking. Please provide more detail, and potentially some references ( e.g. https://journals.ametsoc.org/view/journals/atsc/66/2/2008jas2689.1.xml )

**Rephrased as "This is consistent with higher than observed moisture being advected over the Himalayas by the eastward surface winds in this region…" (Line 665-666)**

- L. 664 missing space

**Fixed.**

- L. 694: informing users of the model?

**Changed to "informing users of the output of these models" (Line 704)**

- L. 705: if the choices of these new model components have been made (e.g., I suspect sea ice is SI3), it would be worth explicitly specifying them.

**Final choices of these model components have not been made yet.**

- Acknowledgement: please acknowledge external data used in the study, e.g. https://confluence.ecmwf.int/display/CKB/How+to+acknowledge+and+cite+a+Climate+Data+Store+%28CDS%29+catalogue+entry+and+the+data+published+as+part+of+it for ERA5.

**References to external data have been included in the main body of the paper, following a recent (2022) similar model evaluation paper in GMD ( https://gmd.copernicus.org/articles/15/2973/2022/)**

**Response to Reviewer 2**

**General comments,**

The manuscript "Improvements in the Canadian Earrth System Model (CanESM) through systematic model analysis: CanESM5.0 and CanESM5.1" details the evaluation of two versions of CanESM with a focus on the improvements and distortions on the global climate and specific geographic locations. The document has the objective of highlighting the benefits of analyzing the model through what the authors call the "Analysis for Development" or A4D. This idea has a lot of potential for model development and understanding processes governing the Climate System. And to highlight this in a publication is a merit that needs to be acknowledged to the authors.

As I mentioned, the intention has its merits, but there are aspects of the manuscript that makes it difficult to find a clear story. Instead, the actual state of the manuscript gives the impression of a collection of reports rather than a story to tell to the scientific community. In the following lines, it is listed the points that the authors can improve.

**Thank you for thorough assessment and helpful comments! Indeed, it was our purpose to highlight how the A4D activity has supported our model development activities with some illustrative reports on investigations into persistent model biases. Thank you for your suggestions on how to improve the coherence of the paper.**

1. The document makes it not clear if the versions and patches of the CanESM5 model were proposed as a result of the A4D initiative or if the versions and patches were decided before, and the A4D only analyzed the outputs. For example, it is not clear if the choices in the parameters that changed between patch 1 (p1) and patch 2 (p2) are the result of the A4D initiative. So, what is the role of the A4D initiative? Were the changes between CanESM5.0 to CanESM5.1 also a product of the A4D initiative?

   **Thank you for pointing this out. In the revised version of the paper, we better highlight the progress that was made in the context of the A4D activity:**

   - **We changed the last sentence of Section 2, to make clear that not all changes that went into CanESM5.1 were the result of the A4D activity (Line 109-111):**

> **The rest of the paper documents results from the first phase of the A4D activity. This includes substantial contributions to the development of a new and improved version of CanESM, CanESM5.1, and a number of deep-dive analyses that provide insight on the origin and possible elimination of systematic model biases in CanESM.**

- **We now highlight in the new Figure 2 which model improvements were the results of the A4D activity**
- **We now explicitly mention that all sensitivity experiments presented in Section 5.3 are the result of the A4D activity (Line 184).**

2. Aside from the versions and patches, the document presents numerous experiments in which only the atmosphere module or the ocean module were used, and all this universe makes it difficult to put in context the advantages of the A4D initiative. An idea could be to make a sketch showing the "genealogy" of the experiments and patches described in the text. And in this sketch, the authors can highlight the versions, patches, and experiments suggested by the A4D initiative.

**All sensitivity experiments presented in this paper have been done in the context of the A4D activity (see also our response to the previous comment). We have added a new section (3.3) in which all experiments are described, with subsection 3.3.2 and the new Table 2 describing the details of the sensitivity experiments, including the model version they were based on.**

3. Moreover, it is not clear if all only-atmosphere or only-ocean experiments will contribute to building a new patch or version of the model. For example, the section about dust tunning is interesting because an error is tracked, fixed, and implemented in the new version of CanESM 5.1. However, the last paragraph in section 5.1 (lines 287-298) is unclear about the reason for using the atmospheric-only simulations, which are not referred to by a name, and what the document is gaining from them. A similar example is the conclusion for the OGWD parameter, which has a different effect between sudden stratospheric warming (SSW) events and the neck wind regions. When I arrived at this part, I asked, "Okay, what are the following steps?"

The atmosphere-only and ocean-only simulations in Section 5 were meant to provide insight into the causes of biases in CanESM5.0/CanESM5.1. While in many cases we have not found a solution to fixing the biases, the insight gained from these sensitivity experiments will all feed into the development of future CanESM versions.

In the revised manuscript, we have clarified the goal of the atmosphere-only simulations in Section 5.1 **(Line 325-330):**

> **While this error will be corrected in future versions on CanESM, we here quantify the impact of this error on already existing simulations. This is done by comparing two atmosphere-only simulations (with and without bare ground fraction correction), in which the atmosphere is nudged to reanalysis so that the observed meteorological conditions, which have a large impact on dust, are well reproduced. We use atmosphere-only simulations with SSTs and sea ice following observations to exclude the possible impact of SST and sea ice biases on our results.**

As already mentioned in the original version of the paper, the OGWD tuning was performed in atmosphere-only mode for efficiency reasons ("to speed up the process"; see Line **570-571**). The original version of the paper also suggested that a next step to simultaneously reduce the bias in both the basic state (including neck region winds) and SSW biases is to implement a new orographic lift component (**Line 609-613**), which is being planned for a future model version.

4. I think that the manuscript will benefit if it is presented first with changes in the global Climate System (historical climate, global dust, ENSO, climate sensitivity) and then shows the regional impacts (Dust in east China, sea-ice area for different months and places, Himalayas'cold bias). And then, it is more obvious to highlight what changed across versions and what is still unchanged. For example, according to the results, fixing the "hybridization" problem corrected the stratospheric temperature spike but with little changes on the global scale (temperature, precipitation, climate sensitivity).

**Thank you for this suggestion, but we believe that the current structure is an effective way to organize and communicate the differences across different model versions. An investigation of the main physical processes and**

**characteristics that are different between CanESM5.0-p2 and CanESM5.1 (dust and stratospheric temperatures) are all collected and discussed in one section (Section 5.1), whereas the main physical processes/characteristics that changed between CanESM5.1-p1 and CanESM5.1-p2 are collected and discussed in another section (Section 5.2). Section 5.3 summarizes investigations of other CanESM5.0/5.1 biases that have not been resolved yet. We believe that reorganization of the paper according to the scale of the process (from global to regional), would make it harder for the reader to find the main differences between the different model versions.**

5. I also suggest adding a summary of the characteristics of the CanESM5 model in terms of resolution (horizontal and vertical), the most important schemes used, and The period of run.

   **We have added information about the horizontal and vertical resolution of the atmospheric and ocean models to Section 3.1 (Line 127-128). Deciding what would be the most important parameterization schemes used in a comprehensive Earth System Model is subjective as it depends on a reader's particular interests, and Swart et al. 2019b already provides a comprehensive overview of the parameterization schemes used in CanESM5.0. The period of run depends on the particular experiment being conducted, and Section 1 already indicates the wide range of experiments conducted (indicated by MIP participation in Figure 1) and total number of simulation years.**

-Line 7: I do not agree with the "substantial improvements". Yes, there are specific improvements, but there are still biases in the representation of global scale temperature, precipitation, seasonality, etc.

**We changed the wording to 'important improvements' (Line 7)**

-Line 17-19: While I like the statement, I find some caveats in this. What do you mean by more reliable climate change projections or high-quality models? Are high-quality models the ones that produce a similar pattern of any climate variable, even if processes are not represented correctly? From my point of view, a climate model has the objective of representing the processes governing the climate inside a framework of the assumptions they are built. Thus, more parameterization schemes (statistical approaches) are used fewer processes are explicitly represented. I think this phrase is referring to the fact that

tunning the model to represent historical climate gives more confidence in the climate projections.

**The questions that the reviewer raises are interesting, but outside of the scope of what we want to discuss in this introductory statement. To address the reviewer's concerns we have removed "effective" and "reliable" since these are subjective terms and would require knowledge about the actual future. By "high-quality" we had intended "physically realistic". Accordingly we have rewritten the two sentences beginning Section 1 as follows (Line 17-19):**

> **Efforts to adapt to and mitigate future climate change rely on climate change projections, which can be provided by climate models. It is therefore important to develop physically realistic climate models in order to provide credible and user relevant output.**

-Line 36: What do you mean to be "particularly good"? Pattern, variability?

**This statement refers to Figure 3.42a of IPCC AR6 (Eyring et al. 2021), which compares the RMSD with respect to observational datasets of 16 atmospheric variables across the CMIP6 multi-model ensemble. The CanESM5 column (10th from left) is mostly blue, indicating better than the median model performance. This is also highlighted in the text of IPCC AR6 (Sec. 3.8.2.1): "Several high climate-sensitivity models (Section 7.5; Meehl et al., 2020), in particular CanESM5, CESM2, CESM2-WACCM, HadGEM3-GC31-LL, and UKESM1-0-LL, score well against the benchmarks." We have clarified the reference to Eyring et al. 2021 to read: "(Eyring et al., 2021, Figure 3.42a)" (Line 36-37)**

-Line 108-109: Is this line suggesting that having more ensembles sacrificing the resolution is indeed good? One can argue that if all the ensembles point out in the wrong direction, there is not any advantage to this.

**This sentence is simply stating that a computationally cheaper model allows for larger ensemble sizes. It is not making a value judgement (good/bad) about this approach.**

-Line 186: The historical Antarctic sea ice trend In Figure 4a is only for September. So, is it enough to use one month to state that the historical sea ice trend is very close to observations?

**Thank you for pointing this out. We have changed the wording to specify that these statements apply to (Arctic) sea ice trends in September only (Line 221-222)**

-Line 198: What is GSAT? it is not specified in the main text.

**GSAT is the Global mean Surface Air Temperature, which is now specified in the revised text.**

-Line 329: From 10b-d, the mean spectrum in CanESM5 is moving to higher frequencies along with the new patches. Do you have an explanation for this?

**First please note that during the review process a bug in our code was discovered. The power spectra for the models was erroneously based on the non-detrended nino3.4 index, as opposed to the detrended nino3.4 index for the observations (ERSSTv5). This is corrected in the revised manuscript, which now shows a much closer correspondence of the model with the observations (Figure 11 in the new manuscript).**

**As the reviewer notes, the power spectrum has a slight shift to lower periods from CanESM5.0-p2 to CanESM5.1-p1 possibly resulting from sampling error in the finite ensemble. There is a larger shift from CanESM5.1-p1 to CanESM5.1-p2, which might be an emergent property of the tuning of ENSO amplitude and ENSO seasonality from CanESM5.1-p1 to CanESM5.1-p2, however, the exact reason for this shift is unclear to us.**

-Lines 359-362: I like this phrase. Do you have any mechanism that could potentially affect the pattern of SST in the Pacific?

**Because there are so many differences between CanOM4 and CanNEMO, this would be very difficult to answer without performing a large array of sensitivity studies. The modified text below, together with the added references for CanOM4 and CanNEMO, should make this clear to readers (Line 399-403):**

> **This strongly suggests that the change from CanOM4 in CanESM2 to CanNEMO (Madec and the NEMO team, 2012) in CanESM5.0/5.1, which involves many differences in the grid configuration, numerics and physical parameterizations, may be the main underlying reason for the equatorial Pacific seasonal cycle changes between CanESM2 and CanESM5.0/5.1. The**

**relative insensitivity to atmospheric model differences in the CanESM5 versions described here aligns with this view.**

-Line 377: What is IRF? This is not explained in the main text.

**Thank you for pointing that out. The revised version now includes an expanded explanation of the adjusted cloud radiative effects (CRE) and the instantaneous radiative forcing (IRF) (Line 418-421)**

**The adjusted CRE method assumes that clouds dampen the instantaneous radiative forcing (IRF, the radiative forcing associated with a particular forcing agent assuming that all other variables are held fixed) by 16% (Soden et al, 2008). Hence, the total sky IRF is calculated by multiplying the clear sky IRF with a globally uniform proportionality constant of 1/1.16. Then, taking the difference between the two yields the portion of the IRF from clouds.**

-Line 394: I did not fully understand your argument about moisture. Is it because the water vapor feedback is higher in patch 2 (p2) than in patch 1 (p1)?

**The sentence refers to the difference between CanESM5.0 and CanESM5.1-p1 (not between CanESM5.1-p1 and CanESM5.1-p2), and changes in moisture would result in changes in clouds. To make this more clear we have changed the text to (Line 438-440):**

**The BCS of CanESM5.1-p1 is slightly lower than that of CanESM5.0, which may be due to the fact that the retuning of the hybridization parameters from CanESM5.0 to CanESM5.1 described in Section 3.2 also affects moisture and hence clouds.**

-Line 395-399: The small reduction in BCS is enough to explain your reduction in climate sensitivity.

**(We assume that this is a question rather than a statement). The reduction in BCS is quite small, and the reduction in climate sensitivity is larger than might be expected based on the relationship between BCS and climate sensitivity across CMIP6 models. Therefore, we state in the paper that "the lower BCS contributes modestly to CanESM5.1-p2's lower climate sensitivity". We do not claim that the lower BCS explains the full reduction in climate sensitivity.**

-Line 402-403: Which type of parameterization in Table 1 do you refer to? What about the role of shallow clouds, as explained by Vogel et al. (2022)?

**Our tuning runs show that MBLC and BCA are most sensitive to ap_facacc (a parameter in the cloud microphysics scheme), ap_scale_scmbf and ap_weight (parameters in the deep convection scheme). For brevity we decided not to add these details to the paper. The results presented in Vogel et al. (2022) strengthen our results, as now described in the revised version of the paper (Line 444-445):**

> **This result is further strengthened by the findings of Vogel et al (2022b), who find that many models overestimate shortwave cloud feedbacks compared to observations.**

-Lines 407-408: What is the connection between excessive sea-ice and salinity? I would have thought that too much sea-ice indicates less fresh water in the ocean and, as a consequence, more salinity.

**Our understanding is that low salinities lead to enhanced stratification of the water column, which prevents convection and vertical mixing of heat, and this is a driver of the excess sea-ice concentration. The ultimate reasons for the low salinity bias are not perfectly understood, but could result from biases in freshwater transport into the ocean, i.e. precipitation and runoff minus evaporation. In terms of excess sea-ice leading to higher ocean salinity as the reviewer states, this might be the case in regions where sea-ice originally forms, and hence where brine rejection occurs. However, in other regions, into which sea-ice is transported (and melts) excess sea-ice would be associated with a low or negative salinity bias. Therefore, excess transport of sea-ice into this region might also play a role in the low salinity bias.**

-Lines 471-477: In all the discussion, there is no mention of the air-sea-ice processes that the model could misrepresent. Do you have any concrete idea? What about the ice module? Is everything okay with that module?

**The ice module is the standard LIM2 sea-ice module, which has been used extensively in the community. Without doubt LIM2 is of limited complexity, and is now somewhat outdated. Nonetheless, similar biases occur even with more advanced ice models (e.g. SI3 in NEMO4) in the CanESM system. We do note that "new approaches**

to sea-ice thermodynamics and coupling are being considered" (Line 486-487). As discussed in Swart et al. (2019b), there are compromises made in the coupling of sea-ice thermodynamics, in order to have model stability and equal representation of processes (e.g. black carbon) across land and sea-ice cover.

-Lines 555-561: It was not clear if changing G(v) or Fcrit led to an increase in SSW events. In CanAM simulations, an increase in G(v) increases SSW events, but this logic does not apply in the CanESM5 simulations. Could you comment on this? Is it because G(v) and Fcrit have a different relationship with OGWD?

An increase in both G(v) and Fcrit are (for different physical reasons) associated with an increased orographic gravity wave flux at the surface, leading to increased deposition of orographic gravity waves in the free atmosphere, decreasing the strength of the polar vortex and increasing the number of SSWs. This explains the increased SSW frequency with increasing G(v) in our CanAM simulations (as we did not perturb Fcrit in CanAM-G[1,2,3]). Regarding the change between CanESM2 and CanESM5, it appears that the lowering of Fcrit outweighs the increase in G(v), leading to a net decrease of the number of SSWs in CanESM5 compared to CanESM2.

-Line 664: It is stated that the lack of realistic topography could be the reason for the cold bias in the Himalayas. Having stated this, do you think that using a resolution of 1° is enough to solve this problem?

The CMIP6 ensemble shows common biases in this region despite many of these models having horizontal resolutions close to 1 degree (Lalande et al. 2021, cited in the paper). This suggests that 1 degree resolution is insufficient to remove this bias. However since CanESM5's Himalaya cold bias is larger than the CMIP6 multi-model mean bias, and CanESM5 has relatively coarse horizontal resolution compared to other CMIP6 models, as noted in Section 5.3.4 we anticipate some reduction of the bias will occur when we increase the resolution. An increased spatial resolution helps with a better representation of topography, which has two effects. First, a better topography is expected to yield more realistic moisture advection patterns. Second, a better representation of topography also allows to model snow deposition and melt more realistically.

---

## Author Response (AR2)

**Responses**

Thank you for your revised manuscript that answers quite exhaustively the remarks of the 2 reviewers. However, I would like you to still consider the following points that I still have (of course based on the two reviews) :

**Thank you for going through our responses so carefully, and for raising the additional issues. We have addressed these as detailed below.**

Reviewer 2, remark 1 : I think that the functioning and impact of what you call the A4D initiative is still not clearly described in the text. After reading the paper, I don't really know what the A4D is in practice. In section 2, the paper details that A4D is a set of best practices in model development (e.g. regular group meetings, systematic documentation and tracking of model issues through Gitlab, engagement of all group members through group-wide acceptance of new ESM versions, etc.) but it seems that it also includes a common set of automatic diagnostics (see the 2nd objective described in section 2). ….

If you want to put emphasis on the A4D initiative, you should better describe in Section 2 not only the objectives but also the practical implementation in terms of procedures and software. Possibly you could make a reference to an internal report describing the different aspects of the initiative, if such a report exists.

**Thank you for this insightful comment. We have reorganized and rewritten parts of Section 2 to better describe the practical implementation of the A4D initiative. Specifically, the objectives are now followed by the phrase 'To this end', after which the actions taken to meet the objectives are described in detail.**

Finally, on line 109, it is written that the "rest of the paper documents results from the first phase of the A4D activity"; why the first phase? What would be the next phases?

**We have removed 'the first phase' from lines 109 and 689, as the definition of a phase is subjective and not further discussed in the paper.**

Also the A4D software or "automatic package" (see line 95) must be made available on Zenodo like the CanESM sources.

**We have added the A4D standardized diagnostics software package to Zenodo, which is now bundled with the reports. In addition, we have included a link to the Zenodo archive in the code and data availability section.**

L 440-442 : As a reply to Reviewer 2, you write "The atmospheric retuning in CanESM5.1-p2 resulted in an even lower BCS and hence a slightly smaller high bias relative to observations, which contributes modestly to CanESM5.1- p2's lower climate sensitivity." If I understand well, you are not sure of the exact relation between BCS and climate sensitivity. Therefore I suggest to add a "probably" in that sentence: "... , which probably contributes modestly to CanESM5.1- p2's lower climate sensitivity."

**Rephrased as:**

> **Given the multi-model relationship between BCS and EffCS evident from Figure 14c and other studies (Brient et al., 2016; Liang et al., 2022), it is likely that this slight BCS reduction contributes modestly to CanESM5.1-p2's lower climate sensitivity.**

L455: I agree with Reviewer 2 that excessive sea-ice and fresh bias in SSS is counter intuitive. Please add some of the explanation you provided to Reviewer 2 in the final version of your manuscript.

**We have added the following footnote to line 455:**

> [7]**While a high sea ice bias is often associated with a high salinity bias in regions where sea ice forms, the low salinity bias simulated in the North Atlantic is associated with enhanced stratification, preventing convection and vertical mixing of heat, possibly driving the high bias in sea ice**

L 156: I think "The syntax changes are bit-identical (i.e., they preserved the bit pattern of the model)." should be "The syntax changes provide bit-identical results (i.e., they preserved the bit pattern of the model results).

**You are correct, we have corrected this.**

L480: The sentence "and hence inherent ocean surface biases cannot be excluded as a possible cause" needs to be rephrased ; the inherent ocean surface biases are the cause of what? Of the ocean and sea ice biases? Please rephrase or simply remove

"and hence that inherent ocean surface biases cannot be excluded as a possible cause".

**We have rephrased this; the new text now reads (line 478-480):**

> **While it would be tempting to attribute the ocean and sea ice biases to CanESM5.0 forcings, it is important to realize that the CanESM5.0 forcings themselves have an imprint of the ocean surface biases in CanESM5.0. Therefore, we cannot exclude the possibility that the ocean surface biases are due to deficiencies in the ocean model.**

L561: I am not sure why you added an hyphen in "CanESM2 simulated a too-weak polar vortex"

**Rephrased to:**

> **… to correct for the weak bias in the polar vortex simulated by CanESM2**

Table 3: Please replace "therodynamics" with "thermodynamics"

**Corrected, thank you.**

---

## Author Response (AR3)

**Responses**

Thank you for your careful corrections. I still have only one minor point. I think "high sea ice bias" in your new sentence "While a high sea ice bias is often associated with a high salinity bias in regions where sea ice forms, the low salinity bias simulated in the North Atlantic is associated with enhanced stratification, preventing convection and vertical mixing of heat, possibly driving the high bias in sea ice", should be better qualified as "high sea ice volume bias" or should be replaced by "excessive sea ice" . Note also that the period (full stop) punctuation mark is missing at the end of the sentence.

**Thank you for raising this issue. We have corrected the text to:**

> **While excessive sea ice is often associated with a high salinity bias in regions where sea ice forms, the low salinity bias simulated in the North Atlantic is associated with enhanced stratification, preventing convection and vertical mixing of heat, possibly causing the excessive sea ice.**